# Molecular insights into the regulation of GNPTαβ by LYSET

Xi Yang [1,2], Balraj Doray [3,5], Danielle Henn[1,5], Varsha Venkatarangan[1,5], Benjamin C. Jennings [3], Zhongzheng Dong [1], Jiaxuan Liang[1], Weichao Zhang [1], Bokai Zhang [1], Linchen Yu[1], Liang Chen [1], Stuart Kornfeld [3,4] & Ming Li [1] ✉

In vertebrates, newly synthesized lysosomal enzymes traffic to lysosomes through the mannose-6-phosphate (M6P) pathway. The Golgi membrane protein LYSET was recently discovered to regulate lysosome biogenesis by controlling the level of GlcNAc-1-phosphotransferase (GNPT). However, its working mechanism remained unclear. In this study, we demonstrate that LYSET is a two-transmembrane protein essential for GNPT stability, cleavage by Site-1 Protease (S1P), and enzymatic activity. We reconcile conflicting models by showing that LYSET enhances GNPT cleavage and prevents its mislocalization to lysosomes for degradation. We further establish that LYSET achieves this by interacting with GOLPH3 and retromer complexes to anchor the LYSET-GNPT complex at the Golgi. Alanine mutagenesis identified an $F^4XXR^7$ motif in LYSET's N-tail for GOLPH3 binding. The retromer further promotes Golgi retention by binding to the C-terminal of LYSET and recycling it from endolysosomes. Together, our findings reveal LYSET's multifaceted role in stabilizing GNPT, retaining it at the Golgi, and ensuring the fidelity of the M6P pathway, thereby providing insights into its molecular function.

In vertebrates, most lysosomal enzymes are delivered to the organelle lumen through the mannose-6-phosphate (M6P) pathway. At the cis-Golgi, GlcNAc-1-phosphotransferase (GNPT) transfers GlcNAc-1 phosphate from UDP-GlcNAc to specific mannose residues on the high mannose glycan chains of lysosomal enzymes, following which an uncovering enzyme (UCE) removes GlcNAc, forming an M6P monoester. At the trans-Golgi, M6P receptors (M6PRs) recognize the M6P modification and selectively bind to these enzymes. M6PRs traffic from the trans-Golgi network (TGN) to the endosomes, releasing the lysosomal enzymes due to increasing acidity. These enzymes further reach the lysosome through endomembrane trafficking, while the M6PRs are recycled back to the TGN through the retromer machinery[1,2]. Disruption of the M6P pathway results in the mistargeting of most lysosomal enzymes and a severe lysosome storage disease, Mucolipidosis type II (MLII)[3].

TMEM251/LYSET/GCAF is a recently identified gene essential for the M6P pathway and the proper trafficking of lysosomal enzymes[4–7]. Patients with a loss-of-function mutation in this gene die in childhood or early adulthood. They present symptoms like those seen in MLII, such as severe skeletal dysplasia, coarsened facial features, short stature, and protruding abdomen[8]. Intriguingly, research shows that cells deficient in LYSET are refractory to several viral infections[7]. This gene is also critical for the propagation of certain tumors under nutrient-poor conditions[6]. Therefore, LYSET is crucial for human health and a potential drug target for treating cancers and viral infections.

LYSET's role within the M6P pathway is uncertain, specifically its effect on GNPT. In early September 2022, we and two other groups published simultaneously the discovery of LYSET and its connection to the M6P pathway. The consensus is that LYSET interacts with GNPT at the Golgi, which is essential for cellular GNPT activity. GNPT is an

[1]Department of Molecular, Cellular, and Developmental Biology, University of Michigan, Ann Arbor, MI, USA. [2]Department of Biological Sciences, Knoebel Institute for Healthy Aging, University of Denver, Denver, CO, USA. [3]Department of Internal Medicine, Washington University School of Medicine, St. Louis, MO, USA. [4]Deceased: Stuart Kornfeld. [5]These authors contributed equally: Xi Yang, Balraj Doray, Danielle Henn, Varsha Venkatarangan. ✉e-mail: mlium@umich.edu

α2β2γ2 hexamer encoded by two genes, *GNPTAB* (encoding α,β subunits) and *GNPTG* (encoding γ subunit). Site-1-Protease (S1P) cleaves the α/β precursor upon its arrival at the Golgi, activating GNPT. We observed that the deletion of *LYSET* resulted in the loss of cleaved β subunit and the accumulation of some uncleaved precursor[5]. Therefore, we proposed that LYSET is critical for GNPT cleavage and activation of its enzymatic activity. However, our model did not explain the reduction of total GNPT protein levels following *LYSET* deletion. An alternative model proposed that the processing of GNPT is unaffected following LYSET knockout; instead, the LYSET-GNPT interaction is necessary for retaining cleaved GNPT at the Golgi. The transmembrane helices of GNPT contain multiple hydrophilic residues that could destabilize the enzyme in the membrane. Without LYSET, cleaved GNPT is trafficked to the lysosome and degraded by lysosomal proteases[6,7,9,10]. However, this model seems counterintuitive since most luminal enzymes fail to reach the lysosome in *LYSET* knockout cells, thereby abolishing their proteolytic activity. Both models account for the absence of cleaved GNPT observed following *LYSET* knockout yet occur through different mechanisms.

In this study, we aimed to resolve this uncertainty and elucidate the molecular details of LYSET function. First, we confirmed that LYSET is essential for the M6P pathway and lysosomal digestion. LYSET interacts with GNPT to regulate its cleavage efficiency, enzymatic activity, and protein stability. Deleting *LYSET* does result in the mislocalization of GNPT to the lysosome and its degradation. Through topology analysis, we determined that LYSET is a 2-transmembrane protein with both termini facing the cytosol. Importantly, in addition to the N-terminal cytosolic tail of GNPT interacting with COPI, the cytosolic tails of LYSET associate with the COPI adaptor GOLPH3 and retromer to maintain the LYSET-GNPT complex at the Golgi. Thus, we propose that LYSET actively contributes to maintaining GNPT Golgi localization by recruiting recycling machinery, including both coatomer and retromer.

## Results

### Reevaluating the relationship between LYSET and GNPTαβ in the M6P pathway

Because of the differences between the models, we felt it was necessary to reexamine the relationship between LYSET and GNPTαβ carefully. Using a GNPTαβ−3HA knock-in strain in HEK293T background[5], we confirmed a loss of the cleaved β subunit upon *LYSET* knockout, accompanied by the accumulation of a higher molecular weight band in the knockout cells (Fig. 1A, B). The higher molecular weight band is attributed to the full-length GNPTαβ being further glycosylated (Fig. 1A, red arrow). There was a ~72% reduction in total GNPTαβ protein levels (cleaved + uncleaved). Importantly, this reduction was not due to decreased transcription, as q-PCR results indicated comparable levels of *GNPTAB* mRNA between WT and *LYSET* knockout cells (Fig. 1C). Additionally, using an antibody recognizing the α subunit, we observed a parallel abolishment of the cleaved α subunit and a reduction in total GNPTαβ levels post *LYSET* knockout (Fig. 1D, E). It's worth noting that the α subunit, after cleavage by S1P, develops more complex-type glycans and migrates similarly to uncleaved GNPTαβ[11]. Following deglycosylation by PNGase F treatment, it migrates faster than the uncleaved GNPTαβ (Fig. 1D, PNGase F-treated panel).

Extending our analysis beyond HEK293T cells, we generated *LYSET* and *GNPTAB* knockouts in the melanoma SKMEL30 cell line, known for its heightened expression of LYSET and GNPTαβ[7]. In line with HEK293T cell results, *LYSET* deletion significantly reduced GNPTαβ levels (~70% reduction) and nearly abolished the cleaved α subunit (Fig. 1F, G). Immunostaining using the α subunit antibody revealed that *LYSET* deletion led to a loss of most endogenous GNPTαβ Golgi signal (Fig. 1H). Instead, we observed a dot-like artifact within the Golgi cisternae of cells lacking GNPTαβ, most likely generated by the

antibody utilized. We did not, however, observe GNPTαβ mislocalization to lysosomes.

Without activated GNPTαβ, one should expect no M6P modifications on lysosomal enzymes. Indeed, using a single-chain antibody against M6P (scFv M6P)[12], we observed no differences between samples from *LYSET* deletion cells, *GNPTAB* deletion cells, or double deletion cells (Fig. S1A). The minor bands observed in knockout lanes likely represent non-specific binding by the antibody[13]. Without M6P modification, luminal enzymes, represented by CTSC and CTSD, cannot traffic to lysosomes and mature, resulting in a significant fraction being secreted into the media as unprocessed proenzymes (Fig. S1B). Enzyme assays of lysosomal hydrolases following CI-MPR-Sepharose pull-down supported these findings, as phosphorylated β-hexosaminidase (β-Hex), α-galactosidase (α-Gal), and β-galactosidase (β-Gal) activity were detected only in whole cell lysates from WT cells (Fig. S1C). Finally, the lack of functional hydrolases was also verified by the accumulation of undigested lysosomal substrates, represented by LAPTM4A and LC3B-II (Fig. S1D).

In summary, using two distinct cell lines (HEK293T and SKMEL30) and antibodies targeting both α and β subunits (after 3HA tagging), our findings consistently support that LYSET is essential for the processing and stability of endogenous GNPTαβ. In the absence of LYSET, the cleaved α and β subunits are undetectable, and M6P modifications on lysosomal enzymes are absent, leading to trafficking defects of luminal enzymes and the accumulation of undigested substrates.

### The multifaceted role of LYSET in GNPTαβ stability, cleavage, and enzymatic activity

The absence of cleaved α and β subunits could be due to either a lack of S1P-dependent GNPTαβ cleavage (processing model)[5] or rapid trafficking of the cleaved α and β subunits to the lysosome for degradation (trafficking model)[6,7]. However, the processing model cannot explain the ~70% loss of total GNPTαβ.

A key experiment to differentiate these models involves using a pulse-chase to track the fate of newly synthesized GNPTαβ in the absence of LYSET. To this end, we expressed GNPTαβ-V5 under a tetracycline-inducible (Tet-on) promoter and established stable cell lines in both WT and LYSET knockout backgrounds. Following a 2-h doxycycline induction, we added cycloheximide to halt protein synthesis and monitored the fate of newly synthesized GNPTαβ-V5 at hourly intervals. In WT cells, the majority of GNPTαβ-V5 was cleaved into the β subunit. Total GNPTαβ-V5 (uncleaved + cleaved) remained relatively stable over a 5-hour chase (Fig. 2A, B). In contrast, in LYSET knockout cells, significantly less β subunit (Fig. 2A, B) and the accumulation of a higher molecular weight band (Fig. 2A, red arrow) were observed. Total GNPTαβ-V5 also decreased after 5 h (Fig. 2A). These results indicate that 1) the processing is nearly abolished, and 2) even uncleaved GNPTαβ-V5 is degraded in the absence of LYSET. They are inconsistent with the hypothesis that LYSET has no impact on GNPTαβ cleavage, as suggested by the trafficking model, but they do support the notion that LYSET serves as an anchor to maintain GNPTαβ at the Golgi, regardless of cleavage.

The processing model proposes that LYSET acts as a bridge between S1P and GNPTαβ to enhance the cleavage of the latter. So, we asked if increasing the amount of S1P would bypass the dependence on LYSET to cleave GNPTαβ. To this end, we expressed GNPTαβ at low levels (400 ng) in *LYSET* knockout cells with or without overexpression of S1P and measured GNPT activity. The presence of the additional S1P clearly increased GNPT activity, nearly doubling its value (from 296 to 544 pmol/hr/mg protein, Fig. 2C, D, compare 2 & 3). However, low-level co-expression of LYSET with endogenous S1P had an even greater impact on GNPT activity (3.4 fold increase), whereas the greatest stimulation (almost 5 fold) occurred when both LYSET and S1P were co-expressed (Fig. 2C, D). These data further support that, unlike the

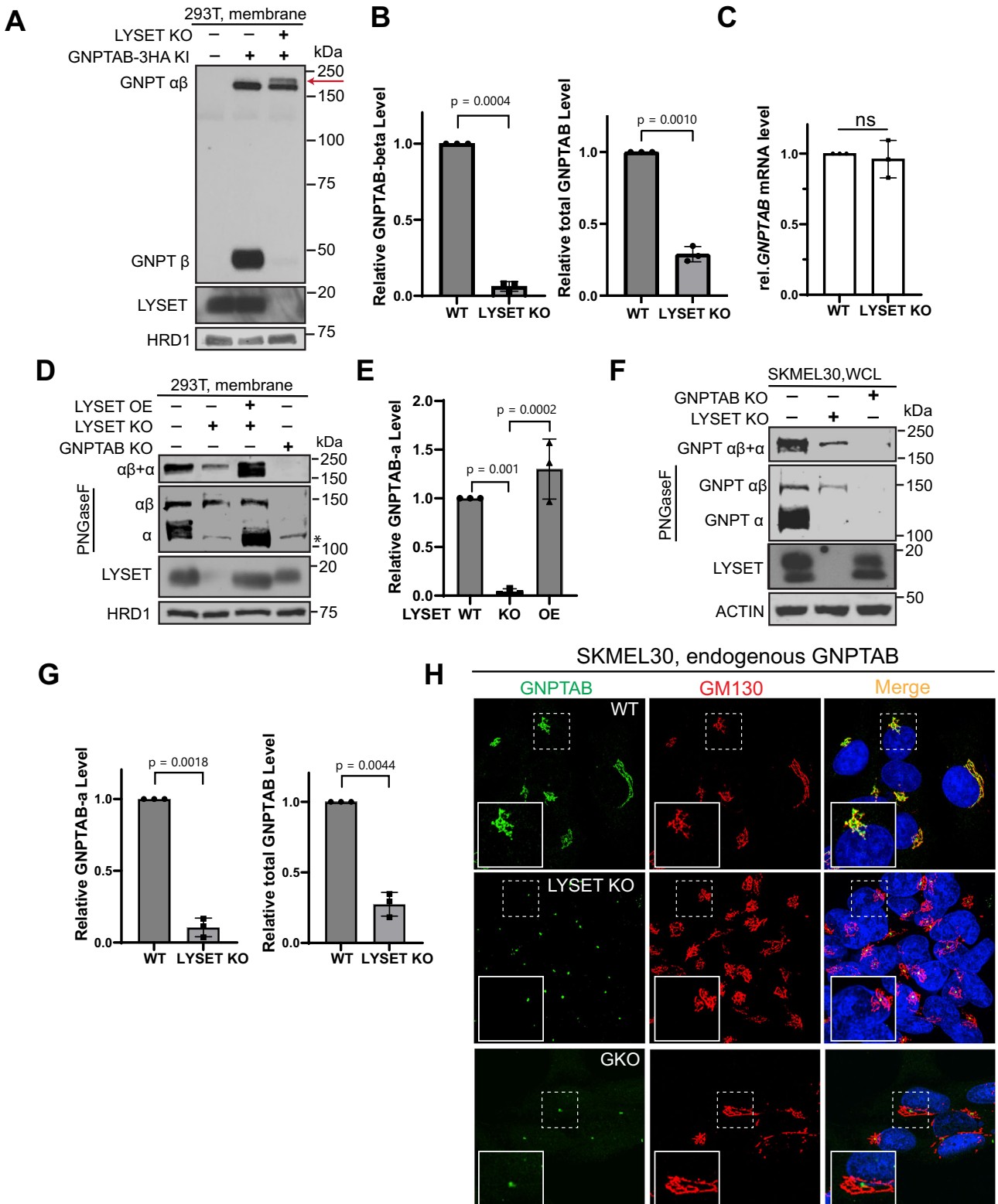

assumption that LYSET has no impact on GNPTαβ cleavage, LYSET works synergistically with S1P to enhance its cleavage/activation efficiency.

We next asked if co-expression of LYSET could stabilize the GNPTαβ protein and increase its PTase activity. Keeping the amount of *GNPTAB* cDNA constant (400 ng), while increasing the amount of *LYSET* cDNA, resulted in rising levels of PTase activity in a dose-dependent manner. Cellular levels of phosphorylated endogenous β-

Hex also increased, but were quickly saturated (Fig. 2E). Further, the amount of the processed β subunit also increased in a LYSET dose-dependent manner (Fig. 2F). These results provide direct evidence that LYSET is critical for stabilizing GNPTαβ and enhancing its PTase activity within a cell.

Lastly, we asked if LYSET is required for GNPTαβ activity after its cleavage/activation. To answer this, we built a GNPTαβ construct that can bypass S1P-dependent processing by replacing its recognition

**Fig. 1 | LYSET is critical for the stability of GNPTαβ. A** Immunoblot showing the full-length and β subunit of endogenous GNPTαβ−3HA in HEK293T WT and *LYSET* KO cells. The red arrow indicates an accumulation of a higher molecular weight modification of GNPTαβ−3HA in *LYSET* KO cells. **B** Quantification of (**A**). Data is presented as mean values +/- standard deviation. N = 3 biological replicates. Statistical analysis was performed using a two-tailed unpaired t-test with Welch's correction. **C** qPCR analysis comparing *GNPT*AB mRNA levels between HEK293T WT and *LYSET* KO cells. Data is presented as mean values +/- standard deviation. N = 3 biological replicates. Statistical analysis was performed using a two-tailed unpaired t-test with Welch's correction. ns no statistical significance. **D** Immunoblot showing the α subunit of endogenous GNPTαβ in the membrane fraction of HEK293T WT, *GNPTAB* KO, *LYSET* KO, and LYSET overexpressing (OE) cells. The LYSET overexpression lysates were diluted to prevent signal over-saturation. The asterisk denotes a non-specific band. **E** Quantification of (**D**). Data is presented as mean values +/- standard deviation. N = 3 biological replicates. Statistical analysis was performed using one-way ANOVA with Dunnett's multiple comparison test. **F** Immunoblot showing the α subunit of endogenous GNPTαβ in SKMEL30 WT, *GNPTAB* KO, and *LYSET* KO cells. WCL: whole cell lysate. **G** Quantification of (**F**). Data is presented as mean values +/- standard deviation. N = 3 biological replicates. Statistical analysis was performed using a two-tailed unpaired t-test with Welch's correction. **H** Immunostaining images showing the localization of endogenous GNPTαβ in SKMEL30 WT, *LYSET* KO, and *GNPTAB* KO cells. The dot-like structure seen near the Golgi in *LYSET* KO and *GNPTAB* KO cells is an antibody staining artifact. Scale bar: 10 μm. Source data are provided as a Source Data file.

sequence with that of Furin[14]. Furin is a calcium-dependent serine endoprotease that can efficiently cleave precursor proteins at the Golgi[15], allowing efficient GNPTαβ cleavage in the absence of LYSET (Fig. 2G, lanes 4 and 5). As an alternative approach to activating GNPTαβ, we directly deleted its auto-inhibition motif and S1P cleavage site (amino acids 820-928)[14]. As shown in Fig. 2G, when stably overexpressed, both constructs can rescue the mutant phenotypes of *GNPTAB* knockout cells but not those of *LYSET* knockout cells, indicating that LYSET is required for the GNPTαβ activity in vivo even after its cleavage.

In summary, we propose that LYSET is necessary to keep GNPTαβ in the Golgi, where it is essential for the stability of both cleaved and uncleaved GNPTαβ and consequently facilitates S1P-mediated cleavage efficiency and activated GNPTαβ enzyme activity.

## Both cleaved and uncleaved GNPTαβ undergo lysosomal degradation without LYSET

What causes the reduction of GNPTαβ after *LYSET* knockout? In Fig. 1H, we cannot determine if endogenous GNPTαβ is mislocalized to the lysosome. To investigate further, we performed a longer exposure of the endogenous GNPTαβ−3HA immunoblot and detected two prominent bands between 15 and 20 kDa in *LYSET* KO cells (Fig. 3A), which likely represent degradation products of GNPTαβ−3HA (Red arrow, Fig. 3B). With the longer exposure, we also observed a minor band corresponding to the β subunit (Fig. 3A).

Since proteasomal degradation typically yields small peptides ranging from 6-10 residues[16,17], the proteasome is unlikely to generate the observed degradation products. Therefore, we explored the possibility of lysosomal degradation. To investigate, we isolated lysosomes using magnetic beads after incubating cells with dextran-coated magnetite (DexoMAG®), which is endocytosed and delivered to the lysosome (Fig. 3C). In lysosomes from WT cells, the β subunit was clearly detected, while levels of the uncleaved GNPTαβ precursor were negligible (Fig. 3D, lanes 4 vs. 5). Conversely, lysosomes from LYSET KO cells contained dramatically increased amounts of both cleaved and uncleaved GNPTαβ−3HA, accompanied by several prominent degradation intermediates (Fig. 3D, lanes 7 vs. 8). To answer if S1P cleavage is required for the mislocalization of GNPTαβ−3HA, we treated cells with S1P inhibitor, PF-429242, for 19 h before isolating lysosomes. As shown in Fig. 4E, even though the cleaved β subunit is much reduced after PF-429242 treatment, uncleaved GNPTαβ−3HA still was mislocalized to the lysosome, and the total level of lysosomal GNPTαβ −3HA was largely unchanged (lanes 8 vs. 10). These results support that endogenous GNPTαβ−3HA is mislocalized to the lysosome without LYSET, regardless of its cleavage state.

We also treated cells with BafA1 as an alternative method to confirm lysosomal degradation. As shown in Fig. 3F, 16 h of BafA1 treatment significantly reduced the prominent degradation products between 15 and 20 kDa and partially stabilized the β subunit. We then investigated whether BafA1 treatment would reveal lysosomal mislocalization of GNPTαβ. Surprisingly, immunostaining with the α

antibody showed that the stabilized GNPTαβ was localized to the Golgi rather than the lysosome (Fig. 3G), suggesting that BafA1 stabilizes GNPTαβ by preventing its exit from the Golgi rather than by inhibiting lysosomal degradation. This result is consistent with that found by Richards et al. and Pechincha et al.[6,7].

In conclusion, our findings independently verified that *LYSET* knockout leads to the mislocalization and degradation of GNPTαβ in the lysosome, regardless of its cleavage state. Interestingly, as noted before, BafA1 treatment prevents GNPTαβ exit from the Golgi rather than solely blocking lysosomal degradation.

## LYSET has two transmembrane helices with both termini facing the cytosol

To understand how LYSET retains GNPTαβ in the Golgi, we first need to establish LYSET 's membrane topology. AlphaFold predicted LYSET to be a three-transmembrane (TM) helices-containing protein[18], while other prediction tools, such as TOPCONS and OCTOPUS, suggested it had two-TM helices[19,20]. A recent publication proposed that LYSET has two transmembrane domains with both N- and C-termini facing the lumen[8]. However, no experimental evidence was presented.

Four reasonable topology models may exist for LYSET, as depicted in Fig. 4A. We first determined if the loop between TM helices 1 and 2 is located in the cytoplasm, as was previously proposed[8]. A native [51]EGT[53] sequence within the loop was mutated to create an NxS/T *N*-linked glycosylation site (E51N, Fig. 4B) with no detrimental effect on the function of LYSET in stimulating GNPTαβ activity (Fig. S2A). The E51N mutant migrated more slowly than WT LYSET, suggesting it was glycosylated. To confirm this, cell lysate was treated with either Endo H, which had minimal effect, or PNGase F, which collapsed the slower-migrating band to the size of WT, confirming that the E51N mutant had acquired complex N-linked glycans (Fig. 4C). This result demonstrates that the loop region located between residues 28 and 68 is in the lumen of the Golgi, ruling out topology models 1 and 3.

The C-terminal tail faces the cytosol in model 2, while in model 4, it faces the lumen (Fig. 4A). To distinguish between these possibilities, we employed a proteinase K sensitivity assay in which cells partially permeabilized with digitonin were treated with proteinase K, which can then only digest cytosolic proteins, but not those in organelle lumens (Fig. 4D).

For this assay, the C-terminus of LYSET was tagged with GFP, which does not disrupt the normal function of LYSET (Fig. S2B, C). As controls, we used Man1A1-GFP, a Golgi membrane protein with its C-terminal in the lumen (proteinase K resistant), and CTNS-GFP, a lysosome membrane protein with the C-terminus facing the cytosol (proteinase K sensitive). Over a 2-min time period, Man1A1-GFP remained fluorescent, while both CTNS-GFP and LYSET-GFP were quenched (Fig. 4E, F). This result shows that the C-terminal tail of LYSET is located in the cytoplasm, eliminating model 4. Therefore, LYSET contains two transmembrane helices, with both its N and C termini facing the cytosol (i.e., topology model 2).

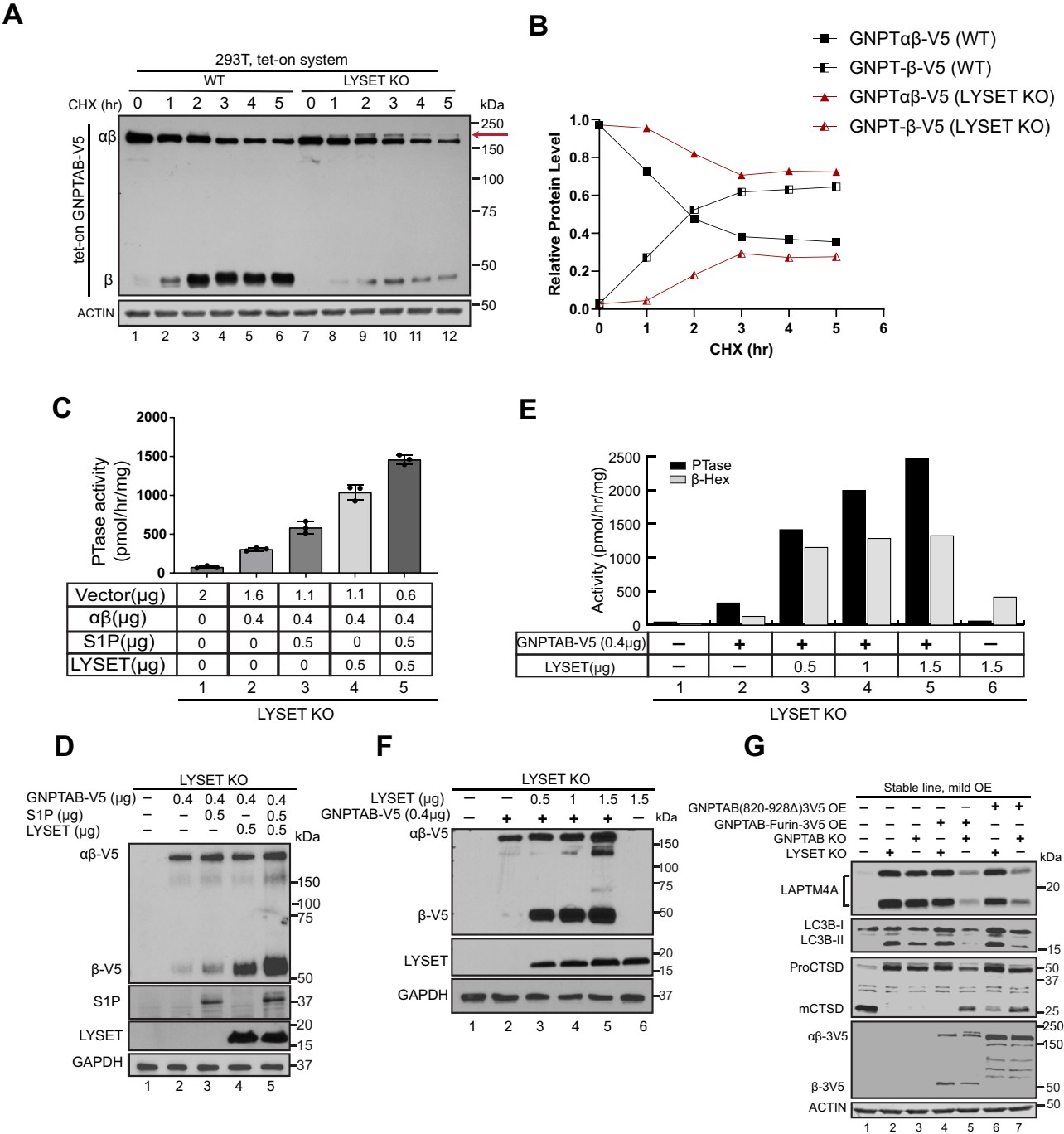

**Fig. 2 | LYSET is essential for the processing efficiency and enzyme activity of GNPTαβ. A** Representative pulse-chase experiment comparing the processing and protein stability of newly synthesized GNPTαβ in WT and *LYSET* KO cells. The experiment was independently repeated three times with similar results. The red arrow indicates a higher molecular weight modification of GNPTαβ-V5. **B** The relative protein level of the full-length GNPTαβ−3V5 and the cleaved GNPTβ−3V5 in WT and LYSET KO cells after 5 h of CHX treatment for (**A**). **C** LYSET and S1P work synergistically to restore the PTase activity of GNPTαβ in *LYSET* KO cells. Data is presented as mean values +/- standard deviation. N = 3 biological replicates.

**D** Immunoblot showing of (**C**). **E** Overexpression of LYSET in the presence of 0.4 μg of GNPTαβ plasmid enhances PTase and β-Hex activities in a dose-dependent manner. Values represent the average of two assays from two independent transfections. **F** Under the same experimental conditions as in (**C**), LYSET overexpression increases the protein stability and processing efficiency of GNPTαβ. **G** Immunoblot showing that two mutants bypassing cleavage, GNPTαβ-Furin-3V5 and GNPTαβ(820-928)deletion-3V5, rescue *GNPTαβ* KO, but not *LYSET* KO cells. Source data are provided as a Source Data file.

## Alanine scan of LYSET reveals residues critical for its function and localization

Next, we performed a comprehensive alanine scanning analysis to identify critical residues for LYSET function. We systematically mutated each amino acid of LYSET to alanine, grouping around five residues at a time. Because the anti-LYSET antibody only recognizes the

C-terminal tail, mutations closer to the C-terminus (residues 102 to 131) cannot be detected by the LYSET antibody. To address this, either an N-terminal 3xFLAG tag or a 3xHA tag was incorporated for immunoblotting and immunostaining.

In total, 31 mutants covering the entire more common, short-isoform of LYSET were expressed in *LYSET* KO HEK293T cells under the

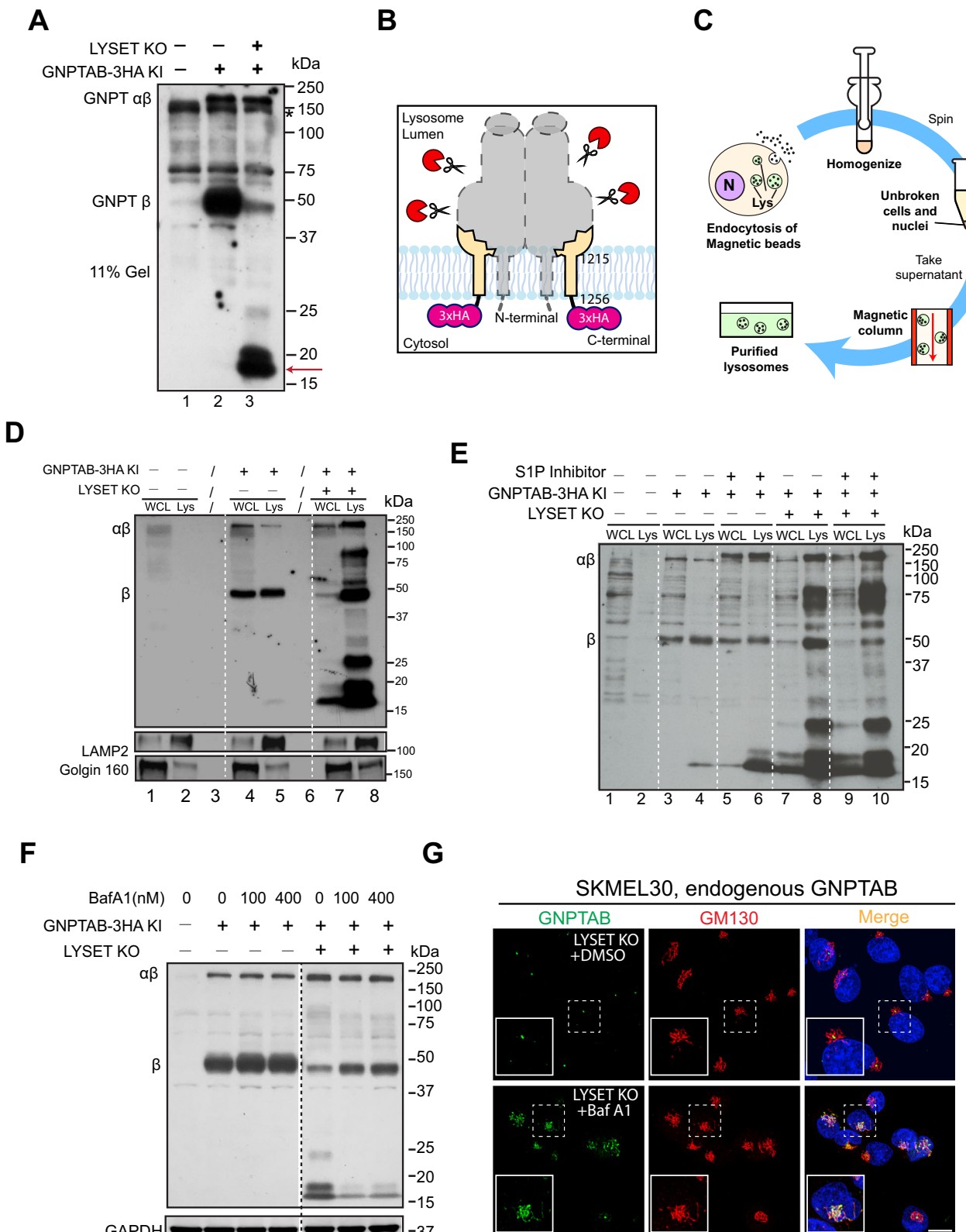

**Fig. 3 | Both processed and unprocessed GNPTαβ are degraded by lysosomes in _LYSET_ KO cells. A** Immunoblot showing the accumulation of endogenous GNPTαβ degradation products between 15 and 20 kDa (red arrow). **B** Model illustrating the degradation products observed in (**A**), likely representing the last TM helix with the 3HA tag. **C** Schematic representation of the magnetic isolation process of lysosomes. **D** Immunoblot comparing the levels of endogenous GNPTαβ−3HA in the whole cell lysate (WCL) and lysosomal fractions (Lys) in WT and _LYSET_ KO cells.

**E** Immunoblot showing the effect of S1P inhibitor (50 μM for 19 h) on the levels of endogenous GNPTαβ−3HA in WCL and lysosomal fractions in WT and _LYSET_ KO cells. **F** Protein levels of GNPTαβ in WT and _LYSET_ KO cells following 16 h of 100 nm BafA1 treatment. **G** Localization of endogenous GNPTαβ in SKMEL30 _LYSET_ KO cells after 16 h of 100 nM BafA1 treatment. Scale bar: 10 μm. Source data are provided as a Source Data file.

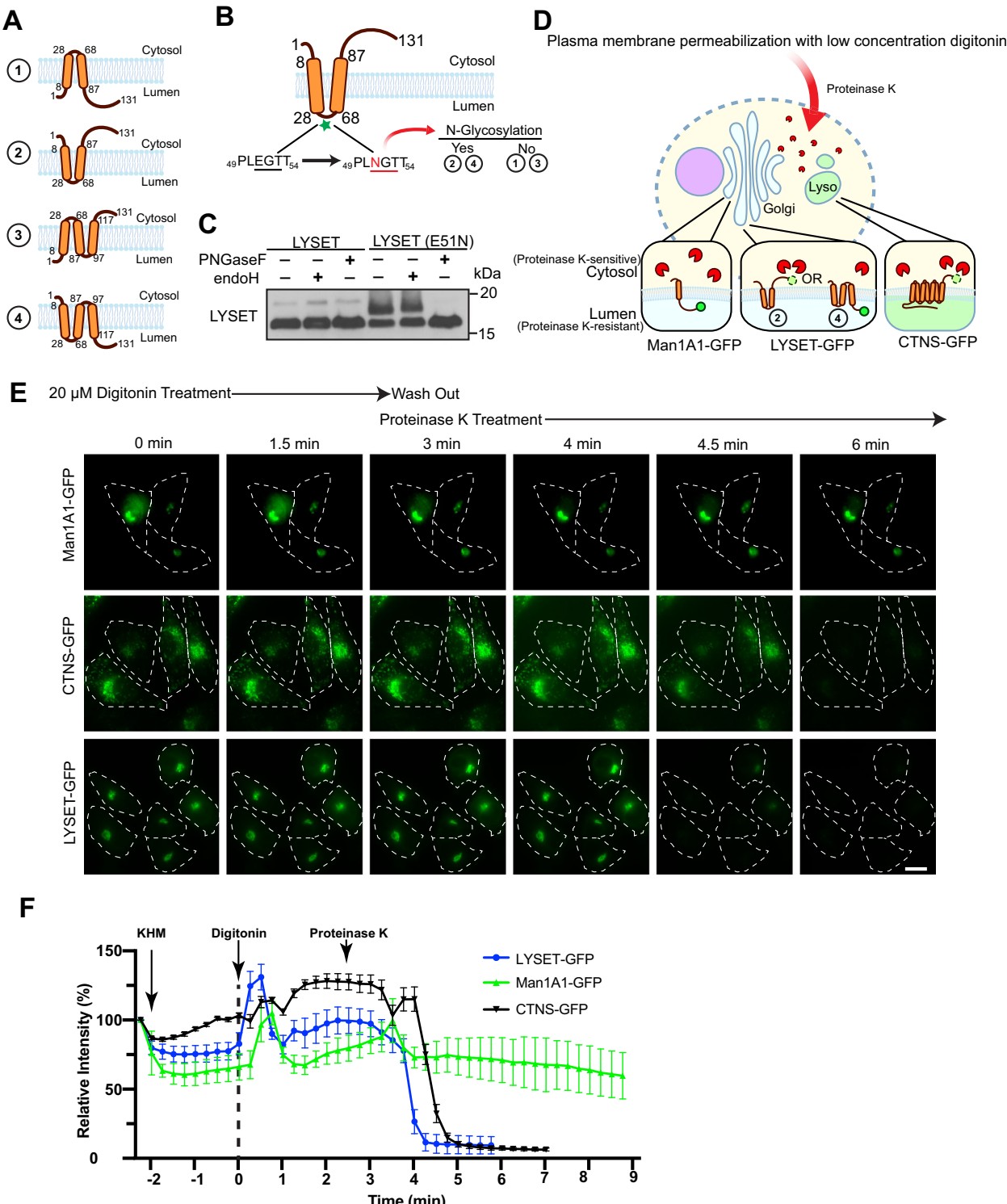

**Fig. 4 | LYSET is a two-transmembrane helix protein with both termini in the cytosol. A** Schematic representation of the four possible membrane topologies of LYSET. **B** Cartoon illustrating the E51N mutation. Glycosylation of this mutant would support either topology 2 or 4 shown in (**A**). **C** Immunoblot showing the expression of LYSET and LYSET (E51N) before and after treatment with either endoH or PNGaseF. **D** Model depicting the limited proteinase K assay. Man1A1-GFP serves as the luminal control, while CTNS-GFP acts as the cytosolic control. **E** The limited proteinase K assay of cells expressing Man1A1-GFP (top panel), CTNS-GFP (middle panel), and LYSET-GFP (bottom panel). Cells were washed with KHM buffer, then treated with 20 μM digitonin for 3 min, followed by 50 μg/ml Proteinase K. **F** Quantification of GFP signal from (**E**). Cells were washed with KHM buffer for 2 min before the addition of digitonin. Scale bar: 10 μm. Source data are provided as a Source Data file.

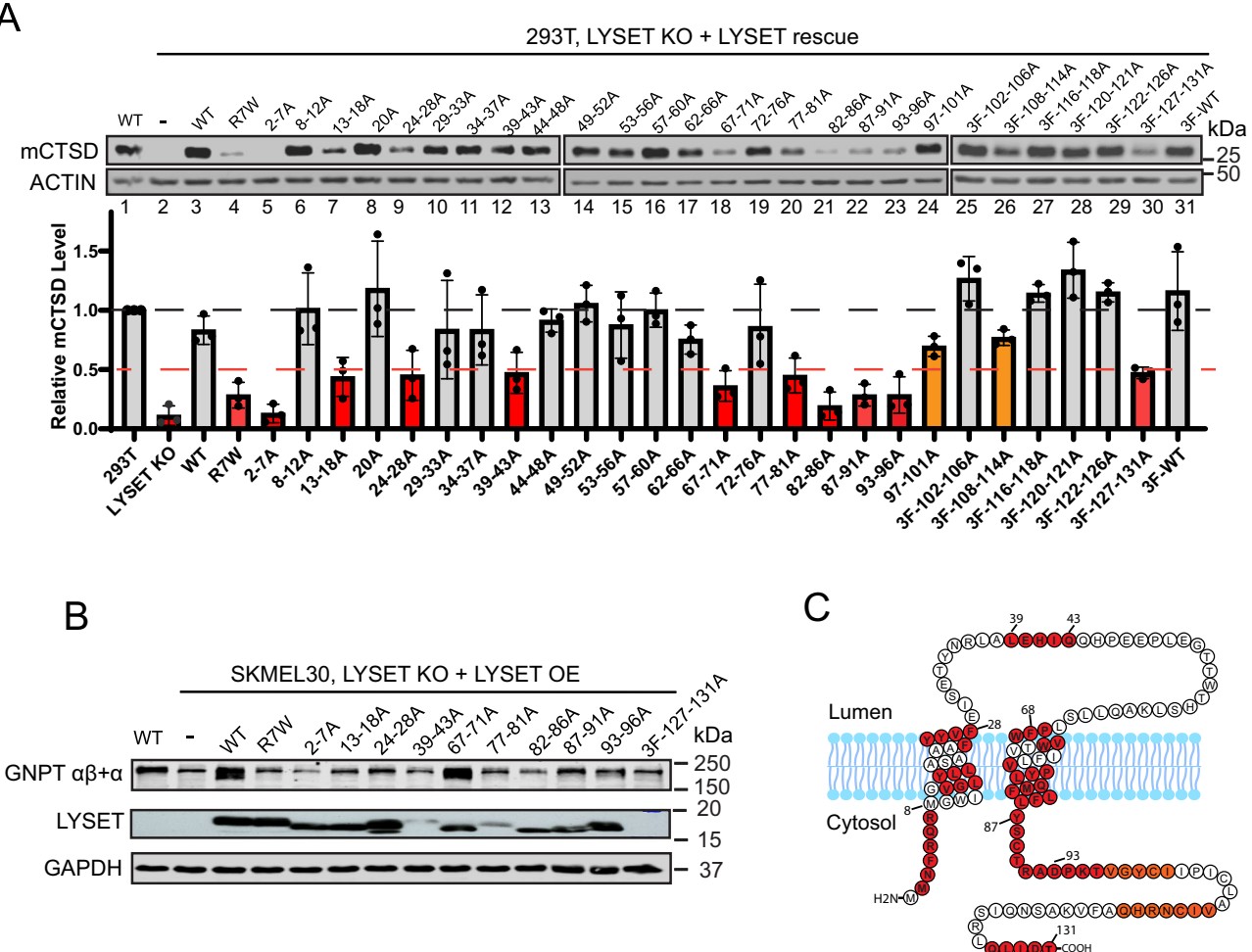

**Fig. 5 | Alanine scan of LYSET reveals residues critical for its function.**
**A** Immunoblot analysis of mCTSD levels in HEK293T *LYSET* KO cells complemented with various LYSET mutants. Mutants with mCTSD levels below 50% of WT levels are marked in red, indicating a loss of function for LYSET. Two additional mutants in the C-terminal, mildly affecting mCTSD levels, were marked in orange. Data is presented as mean values +/- standard deviation. **B** Immunoblot showing the protein levels of GNPTαβ in SKMEL30 cells expressing different LYSET mutants. **C** Topology map of LYSET (based on the model predicted by TOPCON), highlighting residues critical for its function and localization in color. Source data are provided as a Source Data file.

Tet-on promoter. This promoter, when uninduced, maintains a low level of protein expression (leaky expression), preventing potential overexpression artifacts. To assess the functionality of LYSET mutants, we utilized mature CTSD levels as a readout (Fig. 5A). The deletion of LYSET resulted in the abolishment of mCTSD (Fig. 5A, compare lanes 1 to 2). Reintroducing LYSET mutants restored CTSD maturation to varying degrees. A disease mutant, R7W, served as a positive control in this screen[8]. A 50% or less cutoff of WT mCTSD levels identified ten defective mutants with loss of LYSET function (highlighted in red, bottom panel, Fig. 5A). Additionally, two regions in the C-terminal of LYSET showed defective CTSD maturation (highlighted in orange, Fig. 5A, C), although mCTSD levels remained above the cutoff level.

The subcellular localization of these mutants were determined by stably overexpressing them in SKMEL30 *LYSET* knockout cells and immunostaining. Under these conditions, nine out of ten mutants localized to the Golgi and co-localized with GM130 (Fig. S3A, B). Only LYSET(93-96 A) lost its Golgi localization and was trapped in the ER (Fig. S3A–C).

We also examined whether these mutations led to the loss of GNPTαβ. While all other mutations led to a significant reduction in GNPTαβ level, LYSET(67-71 A) rescued it to the same extent as WT (Fig. 5B). This suggests that mutations in this region do not disrupt

LYSET's function of anchoring GNPTαβ in the Golgi, but potentially impair some other function of this protein in the M6P pathway.

In summarizing these results, we plotted the critical regions in the topology model (Fig. 5C). It indicates that the two TM helices (13-18, 24-28, 67-71, 77-81, 82-86) and the immediate cytosolic regions next to them (2-7, 87-91, 93-96) are crucial for LYSET function. Among them, residues 93-96 are particularly vital for the ER exit. Lastly, a small region within the lumenal loop (39-43) and the very C-terminus (127-131) also play a critical role in LYSET function.

### The N-terminal tail of LYSET interacts with GOLPH3 and facilitates its Golgi localization

We next sought to dissect the molecular mechanism that holds the LYSET-GNPTαβ complex in the Golgi. The N-terminal cytoplasmic tail (N-tail) of GNPTαβ is known to bind directly to the β/δ and γ/ζ subunits of COPI, and this interaction is critical for the Golgi localization of GNPTαβ[21]. Patient mutations disrupting this interaction result in the enzyme's mislocalization to the lysosome, leading to Mucolipidosis III[21]. However, the mislocalization of GNPTαβ observed upon LYSET depletion implies that the GNPTαβ-COPI interaction alone is insufficient to retain the enzyme at the cis-Golgi, underscoring the additional contribution of LYSET. Then, how does LYSET perform this role?

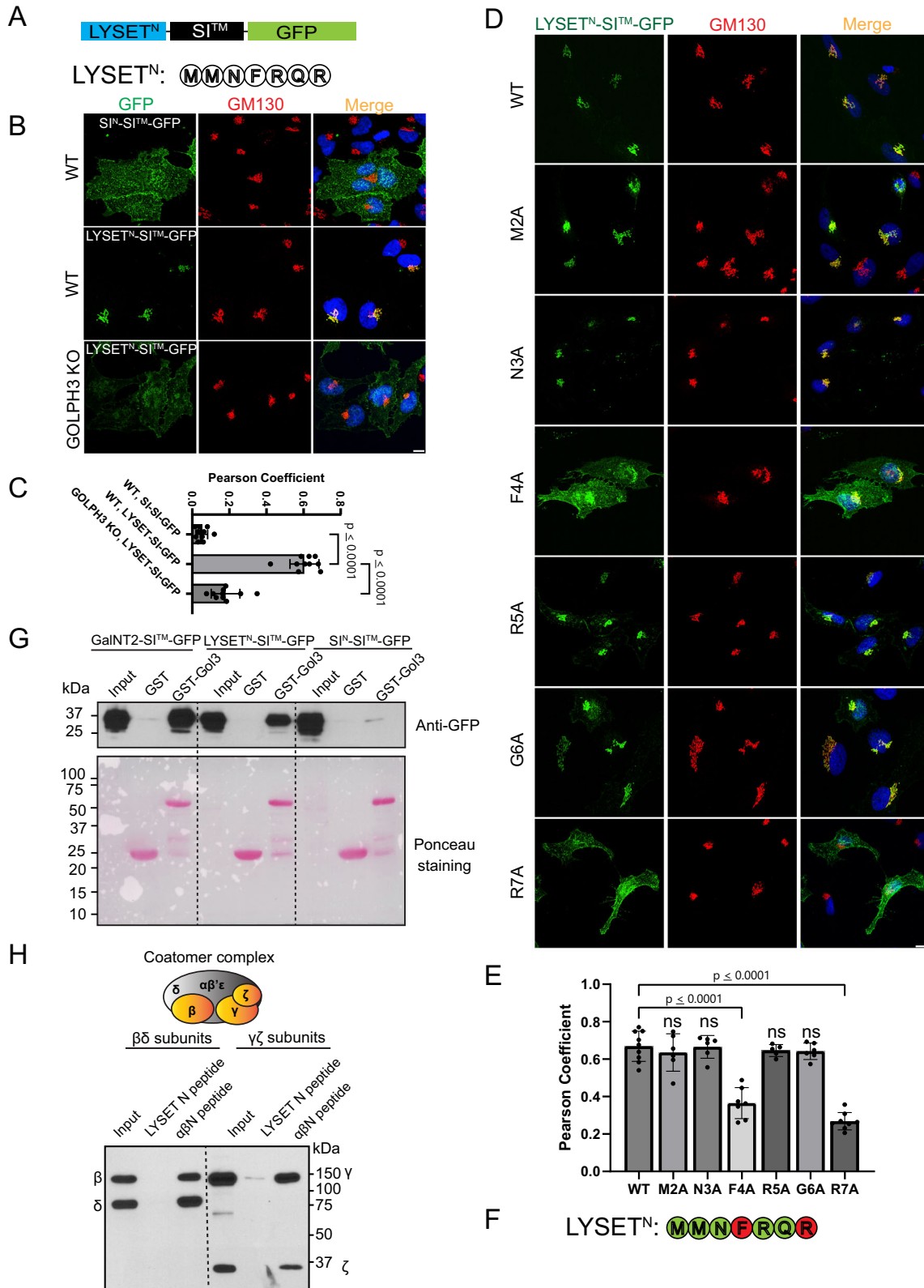

We first determined if the N-terminal cytoplasmic tail of LYSET was sufficient to redirect a GFP reporter from the plasma membrane to the Golgi. For this assay, we utilized a construct, SI$^N$-SI$^{TM}$-GFP, encoding the N-tail and transmembrane segment of the plasma membrane protein, sucrase isomaltase (SI), fused to GFP. We then switched its N-tail with the first seven amino acids of the short isoform of LYSET to generate LYSET$^N$-SI$^{TM}$-GFP (Fig. 6A). We and others have shown that

while SI$^N$-SI$^{TM}$-GFP localizes to the plasma membrane, replacement of its N-tail with the N-tails from several other Golgi glycosyltransferases, including GNPTαβ, restricts the chimera to the Golgi[21,22]. Immuno-fluorescence microscopy of WT HeLa cells transfected with LYSET$^N$-SI$^{TM}$-GFP shows tight colocalization with the Golgi marker, GM130, whereas SI$^N$-SI$^{TM}$-GFP localized predominantly on the plasma membrane (Fig. 6B, C). This result shows that the N-tail of LYSET has

**Fig. 6 | The N-terminal F$^4$XXR$^7$ motif of LYSET contributes to its Golgi localization by interacting with GOLPH3. A** Schematic representation of the chimera. The N-terminal tail of LYSET is fused with the transmembrane helix of sucrase isomaltase (SI), followed by GFP. **B** Colocalization of SI$^N$-SI$^{TM}$-GFP and LYSET$^N$-SI$^{TM}$-GFP with GM130 in HeLa WT and *GOLPH3* KO cells. **C** Pearson coefficient analysis of (**B**) between GFP and the Golgi marker GM130. Data is presented as mean values +/- standard deviation. Each dot represents one cell. Statistical analysis was performed using one-way ANOVA with Tukey's multiple comparison test. **D** Alanine mutagenesis scan to identify critical residues required for Golgi localization. **E** Pearson coefficient analysis of (**D**) between GFP and the Golgi marker GM130. Data is presented as mean values +/- standard deviation. Each dot represents one cell. Statistical analysis was performed using one-way ANOVA with Dunnett's multiple comparison test, comparing the N-terminal mutants with WT cells. ns – no statistical significance. **F** The F$^4$XXR$^7$ motif is critical for the Golgi localization of LYSET. **G** Pull-down assays demonstrating the interaction between the LYSET-N chimera and GST-GOLPH3. GalNT2-SI$^{TM}$-GFP and SI$^N$-SI$^{TM}$-GFP served as positive and negative controls, respectively. **H** Pull-down assays using synthesized LYSET N-terminal peptide showed no interaction with the COPI β/δ or γ/ζ subcomplexes. The N-terminal peptide of GNPTαβ (αβN peptide) was used as a positive control. Scale bar: 10 μm. Source data are provided as a Source Data file.

sufficient information to retain the chimera in the Golgi. Further investigation via an alanine mutagenesis scan revealed that residues F4 and R7 are critical for the Golgi localization (Fig. 6D–F). Importantly, the R7W mutation in humans is known to cause dysostosis multiplex, also known as Mucolipidosis type V[5,8].

To further test the importance of the F$^4$XXR$^7$ motif, we tagged full-length LYSET mutants (F4A, R5A, or R7A) with a 10xGCN4 tag[23] (Fig. S4A) and expressed them under the leaky expression of a Tet-on promoter. Both F4A and R7A mutations failed to rescue the LYSET KO phenotype in HEK293T cells, as indicated by the lack of CTSD maturation (Fig. S4B, C). Moreover, when moderately induced by a 100 ng/ml doxycycline, these two mutations resulted in reduced GM130-LYSET colocalization and mislocalization of some LYSET to the plasma membrane in SKMEL30 cells (Fig. S4D-E). In contrast, LYSET-T(R5A) retained the ability to rescue mCTSD (Fig. S4B-C) and was still localized to the Golgi, although an increased ER signal was observed (Fig. S4D, E).

The adapter protein GOLPH3 has multiple recognition motifs, one of which is an LXXR motif. The F$^4$XXR$^7$ motif at the N-terminal of LYSET is similar to the LXXR motif, it suggesting that its interaction with COPI might be mediated by GOLPH3[22,24–26]. The Munro group has shown that a GST-GOLPH3 fusion protein efficiently bound GalNT2-SI$^{TM}$-GFP that carries the N-tail of the Golgi enzyme *N*-acetylgalactosaminyltransferase-2[22]. We prepared GST-GOLPH3 and performed the binding assays using GalNT2-SI$^{TM}$-GFP and SI$^N$-SI$^{TM}$-GFP as positive and negative controls, respectively. As shown in Fig. 6G, LYSET$^N$-SI$^{TM}$-GFP bound well to GST-GOLPH3, while there was no binding of GST-GOLPH3 to SI$^N$-SI$^{TM}$-GFP as previously reported[22].

We also tested whether the coatomer complex directly interacts with the N-terminus of LYSET. To this end, we synthesized a biotinylated peptide corresponding to the first 7 amino acids (N-tail) of LYSET, immobilized it on streptavidin beads, and performed pull-down assays with Sf9 insect cell lysates expressing the β/δ or γ/ζ subcomplexes of COPI[21]. While the GNPTαβ N-tail (positive control) robustly bound both β/δ and γ/ζ dimers, no binding was detected with the LYSET N-tail (Fig. 6H).

To confirm GOLPH3's role in vivo, we deleted *GOLPH3* in HeLa cells, which caused LYSET$^N$-SI$^{TM}$-GFP to lose most of its Golgi localization (Fig. 6B, C). U2OS cells lacking GOLPH3/3 L also had reduced colocalization of induced LYSET-10xGCN4 with GM130 (Fig. S5A, B). Furthermore, immunostaining of SKMEL30 cells confirmed diminished Golgi signals for GNPTαβ in GOLPH3-deficient cells (Fig. S5C–E), underscoring the importance of LYSET for maintaining GNPTαβ at the Golgi.

Finally, knocking down GOLPH3 disrupted lysosomal enzyme trafficking in HeLa cells, as evidenced by impaired CTSC and CTSD maturation (Fig. S5F-G).

In summary, our findings demonstrate that LYSET's N-tail mediates Golgi localization through its GOLPH3 interaction.

## The retromer complex recycles LYSET from the endo-lysosome to the Golgi

Another critical molecular machinery for maintaining membrane proteins at the Golgi is the retromer complex[27–30]. Retromers actively retrieve cargo proteins, such as CI-MPR and sortilin, from endosomes and transport them back to the trans-Golgi network, preventing their degradation in the lysosome. To assess whether the retromer complex also contributes to the Golgi localization of LYSET, we deleted *VPS35*, a key component of the VPS26/29/35 retromer subcomplex, in HEK293T cells. Deletion of *VPS35* resulted in a significant reduction in total endogenous GNPTαβ protein levels, while LYSET levels remained largely unchanged (Fig. 7A, B). Lysosome purification from two independent knockout colonies revealed that significant fractions of GNPTαβ and LYSET were mislocalized to the lysosome and partially degraded (Fig. 7C, lanes 6, 8 vs. 4). Furthermore, we observed a decrease in VPS26, another essential component of the retromer complex, indicating that the stability of retromer subunits is dependent on an intact retromer subcomplex (Fig. 7A). Similar observations have been reported before[31].

We then utilized the SKMEL30 cell line to investigate the localization of endogenous LYSET and GNPTαβ through immunostaining. As shown in Fig. 7D, E, deletion of *VPS35* led to the mislocalization of endogenous LYSET from the Golgi to punctate structures. Moreover, the Golgi localization of GNPTαβ was markedly reduced (Fig. 7D, E). These results underscore the critical role of the retromer complex in maintaining both proteins at the Golgi.

We then explored whether overexpression of LYSET could restore the Golgi localization of GNPTαβ in *VPS35* deletion cells. Indeed, upon overexpression, a substantial amount of LYSET localized to the Golgi, leading to the restoration of GNPTαβ Golgi localization (Fig. 7D, E). These findings suggest that the loss of GNPTαβ Golgi localization following *VPS35* deletion is attributable to the depletion of LYSET from the Golgi.

Since the retromer is required to maintain LYSET at the Golgi, our next goal was to identify the retromer-binding motif within LYSET. We first asked whether the cytosolic C-terminal region of LYSET contains sufficient information for Golgi localization. To test this, we generated a chimera by replacing the cytosolic tail of the plasma membrane protein interleukin-2 receptor alpha chain (IL-2Rα) with the C-terminal tail of LYSET (amino acids 88–131) (Fig. 8A)[32]. In WT cells, this chimera displayed increased Golgi localization, indicating that the LYSET C-terminal tail contains a functional Golgi-targeting signal (Fig. 8B, C). However, in VPS35 knockout cells, this Golgi localization was lost, and the chimera redistributed into punctate structures (Fig. 8B, C). These results suggest that the C-terminal tail of LYSET confers Golgi localization in a retromer-dependent manner.

To further map the motif responsible, we performed an alanine scanning mutagenesis of the C-terminal region (residues 87–131) in the LYSET-10xGCN4 construct and analyzed subcellular localization in SKMEL30 cells. Of these, we observed decreased Golgi localization and increased punctate signal for mutations clustered in two regions: residues 97 and 99, and residues 108–110. Notably, these regions partially reduced mCTSD levels in earlier experiments, though not below our predefined threshold (Fig. 5A, C – orange).

To test whether these motifs function cooperatively, we combined the mutations. Constructs bearing either the 97 A,99 A or 108–110 A mutations alone showed only a slight increase in

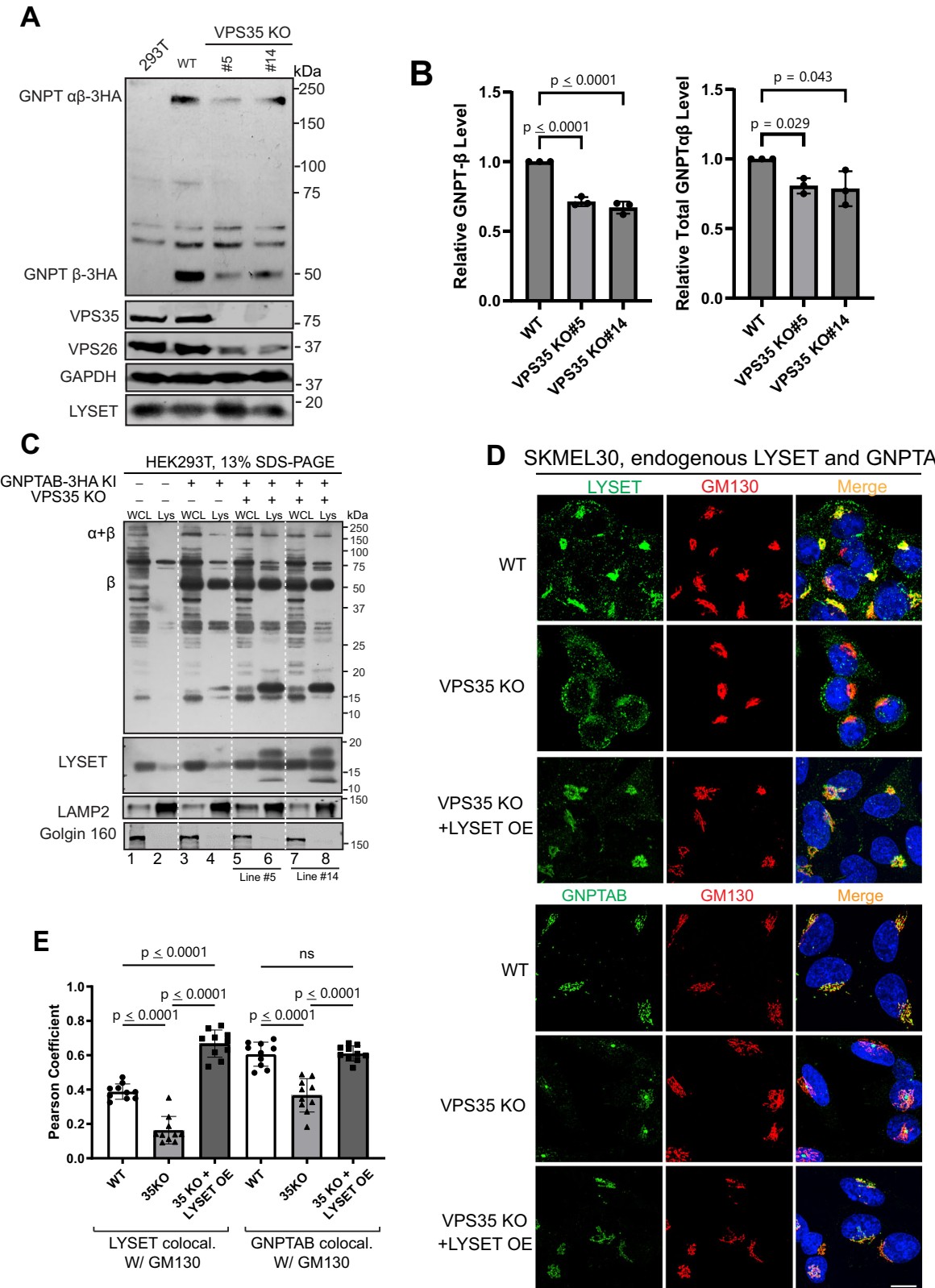

**Fig. 7 | The retromer complex recycles LYSET from endo-lysosomes to the Golgi. A** Immunoblot showing endogenous GNPTαβ−3HA protein levels of two independent *VPS35* KO colonies. **B** Quantification of (**A**). Data is presented as mean values +/- standard deviation. N = 3 biological replicates. Statistical analysis was performed using one-way ANOVA with Dunnett's multiple comparison test. **C** Lysosome isolation demonstrating that a fraction of GNPTαβ−3HA and LYSET is mislocalized to the lysosome and degraded in *VPS35* KO cells. **D** Localization of endogenous LYSET and GNPTαβ in SKMEL30 WT, *VPS35* KO, and *VPS35* KO + *LYSET* overexpression cells. **E** Pearson coefficient analysis of (**D**) between LYSET and the Golgi marker GM130. Each dot represents one cell. Data is presented as mean values +/- standard deviation. Statistical analysis was performed using one-way ANOVA with Tukey's multiple comparison test comparing VPS35 KO cells and VPS35 KO cells with LYSET overexpression to WT cells. ns no statistical significance. Scale bar: 10 μm. Source data are provided as a Source Data file.

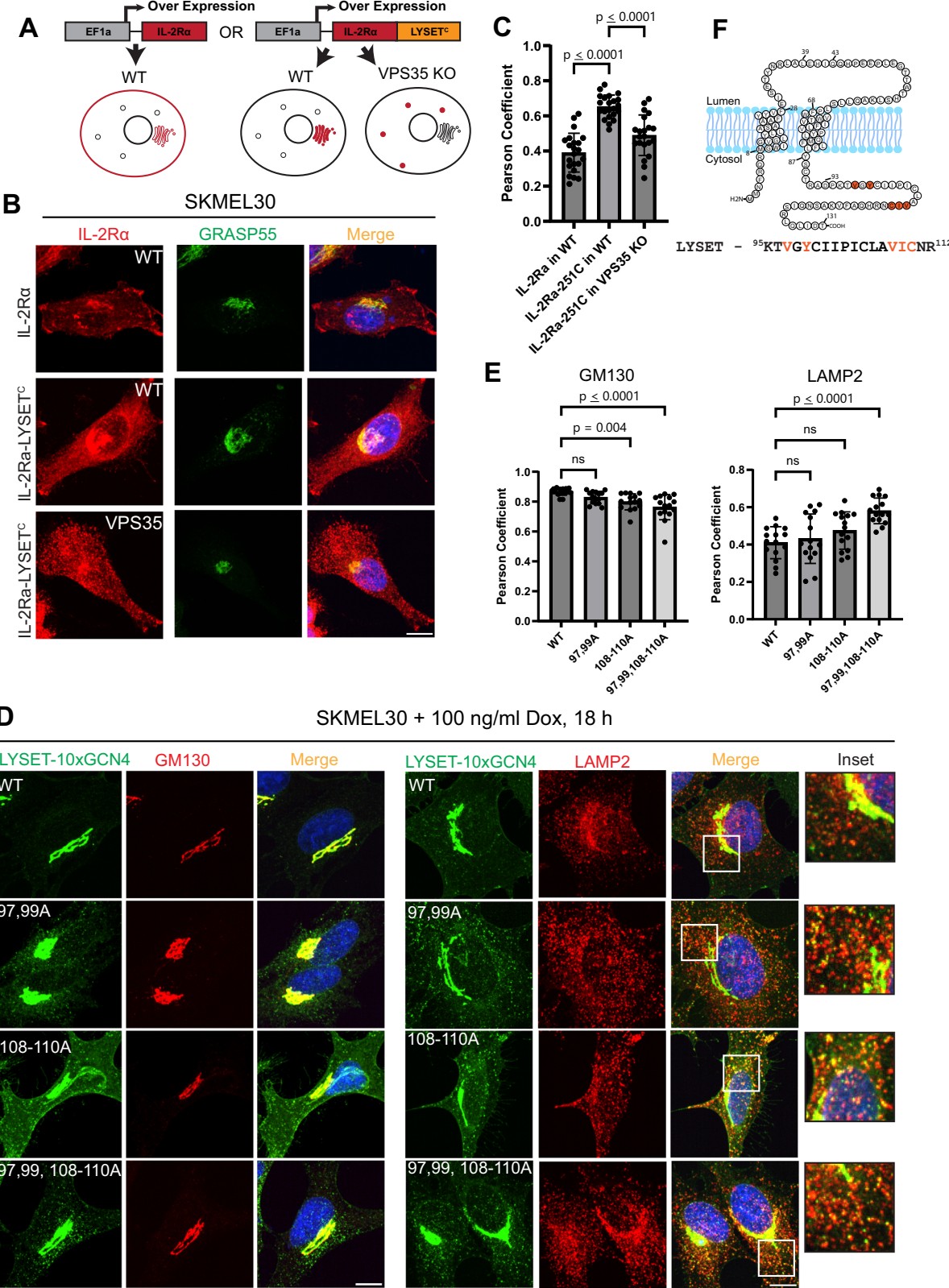

LAMP2 colocalization, without a significant reduction in Golgi signal (Fig. 8D, E). However, the combined mutant, LYSET-10xGCN4(97,99,108–110 A), showed significantly reduced colocalization with the Golgi marker and markedly increased colocalization with LAMP2 (Fig. 8D, E), suggesting additive disruption of Golgi retention. These findings imply that retromer binds cooperatively to both motifs for efficient LYSET recycling to the Golgi (Fig. 8F).

Bringing together our findings from alanine scanning, GOLPH3 interaction, and retromer dependence, we propose a model in which LYSET actively maintains GNPTαβ localization and function at the Golgi (Fig. 9). First, LYSET forms a stable complex with GNPTαβ. Second, the N-termini of both LYSET and GNPTαβ interact with COPI machinery, either directly(GNPTαβ) or via GOLPH3 (LYSET), promoting retrograde recycling from the trans-Golgi to the cis-Golgi. Third,

**Fig. 8 | Critical C-terminal residues required for LYSET interaction with the retromer complex. A** Schematic representation of the Interleukin-2 receptor alpha chain (IL-2Rα)−LYSET chimera. **B** Colocalization of the IL-2Rα chimera, with or without the C-terminal tail of LYSET, with GRASP55 in WT and *VPS35* KO SKMEL30 cells. **C** Pearson's coefficient analysis of (**B**) between IL-2Ra and the Golgi marker GRASP55. Each dot represents one cell. Data is presented as mean values +/- standard deviation. Statistical analysis was performed using one-way ANOVA with Dunnett's multiple comparison test. **D** Immunostaining images of LYSET(WT), LYSET(V97A, Y99A), LYSET(V108A, I109A, C110A), and LYSET(V97A, Y99A, V108A,

I109A, C110A) fused to a 10xGCN4 tag in SKMEL30 cells after 18 h of induction with 100 ng/ml doxycycline. **E** Pearson's coefficient analysis of (**D**) between LYSET-10xGCN and the Golgi marker GM130 or the lysosomal marker LAMP2. Each dot represents one cell. Data is presented as mean values +/- standard deviation. Statistical analysis was performed using one-way ANOVA with Dunnett's multiple comparison test, comparing the Pearson Coefficient of C-terminal LYSET mutants to WT. ns no statistical significance. **F** Schematic summarizing the LYSET C-terminal residues critical for retromer binding. Scale bar: 10 μm. Source data are provided as a Source Data file.

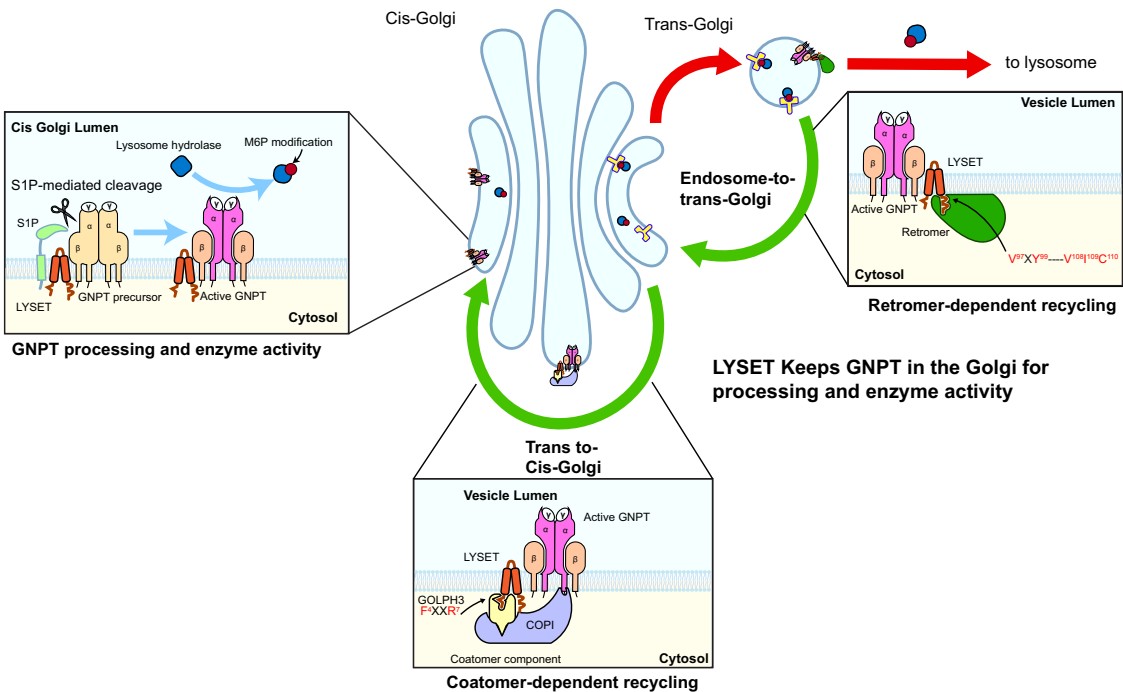

**Fig. 9 | A model summarizing the role of LYSET in the processing, enzymatic activity, and localization of GNPT.** GNPT is kept in the Golgi, where it is responsible for tagging lysosomal enzymes with an M6P tag, via interaction with both LYSET and the Coatomer complex. LYSET itself is maintained in the Golgi by interaction with both GOLPH3, a coatomer adapter protein, and the retromer.

under normal conditions, a fraction of the LYSET−GNPTαβ complex escapes the Golgi and traffics to endo-lysosomes, but this population is retrieved by the retromer complex, which recognizes the C-terminal motifs of LYSET. Thus, LYSET plays an active role in Golgi retention, bridging COPI-mediated retrograde trafficking and retromer-mediated retrieval, ensuring proper localization and activity of the GNPTαβ complex.

## Discussion

### LYSET is critical for GNPT activity at the physiological level

For many years, the M6P pathway was considered a well-understood mechanism for lysosome biogenesis. The study uncovering the role of COPI in the Golgi recycling of GNPTαβ and the cryo-EM work elucidating its structure seemed to close a final chapter in the field[14,21,33,34]. However, the recent discovery of LYSET and its indispensable role in the M6P pathway has reignited interest in this field of exploration. The phenotype of LYSET ablation closely mimics the loss of GNPT, resulting in Mucolipidosis type V. While there is consensus that LYSET plays a critical role in the M6P pathway, the specific mechanism of this regulation has been a subject of debate. In this study, we revisited the intricate details of how LYSET modulates the M6P pathway.

Our findings highlight the significance of LYSET in facilitating the efficient cleavage of newly synthesized GNPTαβ (Fig. 2A). Even following cleavage, LYSET remains crucial for its activity

when GNPT is expressed at endogenous levels or mildly over-expressed (Fig. 2G).

This regulation might operate on two levels. First, LYSET may act as a mediator between S1P and GNPT to enhance cleavage efficiency. In support of this model, we previously observed that LYSET can pull down both GNPT and S1P[5]. Without LYSET, cleavage and activation become inefficient. Second, the LYSET-GNPT interaction retains GNPT within the Golgi apparatus. In the absence of LYSET, the uncleaved GNPT rapidly exits the Golgi, reducing the precursor concentration and diminishing S1P cleavage efficiency. However, overexpression of GNPT or S1P could increase the likelihood of interaction between GNPT and S1P, thereby bypassing the need for LYSET for cleavage.

### LYSET preserves GNPT at the Golgi

Our investigation reaffirmed the vital role of LYSET in maintaining GNPT at the cis-Golgi, consistent with the initial suggestions by Richards et al. and Pechincha et al.[6,7]. Using magnetic beads, we purified lysosomes and observed that, in the presence of LYSET, ~10% of the activated GNPT was already localized to the lysosome in WT cells (Fig. 3D, E). In contrast, minimal uncleaved GNPT reached the lysosome. Upon *LYSET* deletion, a significant increase of both uncleaved and cleaved GNPT in the lysosome was observed. Notably, the majority of these mislocalized enzymes underwent proteolytic cleavage, resulting in smaller degradation products. These results also indicate that the lysosomes still retain partial proteolytic function, implying

that there are other M6P-independent pathways to traffic lysosomal enzymes.

Our attempts to detect GNPTαβ mislocalization through immunostaining were unsuccessful. While endogenous GNPTαβ was readily detected in WT SKMEL30 cells, *LYSET* deletion consistently abolished most of the Golgi signal without a corresponding shift to the lysosome, probably because the antigen regions were digested by the lysosome. In contrast, Richards et al. observed striking relocalization of endogenous GNPTαβ to the lysosome in HAP1 cells following LYSET deletion, with BafA1 further stabilizing GNPTαβ on the lysosome. Notably, treating cells with BafA1 also affects the Golgi exit of GNPTαβ[6,7,21].

## LYSET interacts with GOLPH3 and Retromer to maintain GNPT at the Golgi

How does LYSET retain GNPT at the Golgi? Pechincha et al. proposed that GNPT is unstable due to its relatively hydrophilic transmembrane domain. The interaction between LYSET and GNPT within the membrane is thought to shield these hydrophilic residues from exposure to the lipid bilayer. Supporting this hypothesis, our alanine scanning results demonstrated that the two transmembrane helices of LYSET are critical for its function[35] (Fig. 5C).

Interestingly, our analysis also revealed that the cytosolic regions adjacent to the transmembrane helices play a significant role in LYSET function. Further investigation showed that the N-terminal $F^4XXR^7$ motif of LYSET interacts with the COPI adaptor GOLPH3, a key factor in maintaining the LYSET-GNPT complex at the Golgi. While our paper was under revision, an independent study confirmed that GOLPH3 and its homolog GOLPH3L are crucial for retaining the LYSET-GNPT complex at the Golgi[36].

However, additional mechanisms likely contribute to the Golgi retention of LYSET. Depleting GOLPH3 or mutating the N-terminal tail to alanine did not completely disrupt the Golgi localization of overexpressed LYSET (Fig. S4D, E and Fig. S5A, B), indicating that Golgi retention is not solely dependent on this region.

Our data also revealed a third layer of complexity to ensure GNPT retention. Specifically, the retromer complex plays a critical role in recycling LYSET and GNPT back to the Golgi from post-Golgi compartments. Deleting *VPS35*, a key retromer component, resulted in the mislocalization and degradation of both LYSET and GNPT. Importantly, overexpression of LYSET partially rescued GNPT stability in *VPS35* KO cells, suggesting that LYSET mislocalization is the primary cause of GNPT instability and mislocalization in these cells (Fig. 7D, E).

We show that the C-terminal cytosolic region of LYSET contains enough information to confer Golgi localization mediated by retromer-dependent recycling. The retromer likely binds to two hydrophobic regions in the C-terminal of LYSET ($V^{97}XY^{99}$---$V^{108}IC^{110}$). Mutating these residues results in the redistribution of LYSET from the Golgi to the lysosome, much like when deleting *VPS35*. However, directly demonstrating the physical interaction has proven challenging. Pull-down assays failed to detect an interaction between WT LYSET and the retromer complex, despite VPS35 deletion experiments showing it is critical for LYSET localization and GNPT stability. This suggests that the interaction between LYSET and the retromer is either transient and weak or mediated by an additional cytosolic adaptor, similar to GOLPH3.

In conclusion, our findings indicate that the trafficking of GNPT within the cell relies on multiple coordinated mechanisms. These include LYSET-mediated stabilization at the Golgi via transmembrane and cytosolic interactions, GOLPH3-dependent retention, and retromer-mediated retrieval from post-Golgi compartments. Having two distinct methods (i.e., the GNPT-coatomer interaction and the LYSET-GOLPH3 interaction) for recycling GNPT at the Golgi and a third to retrieve any escaped enzyme ensures the robustness of the M6P pathway, minimizing the risk of its compromise.

# Methods

## Mammalian cell culture

Cell lines used in this study are listed in Supplemental Table S1. HEK293T (CRL-3216) and HeLa (CCL-2) were purchased from ATCC, and SKMEL30 (SK1980-526) was purchased from Memorial Sloan Kettering Cancer Center. The U2OS cell line was a kind gift from Prof J. Tan. Cells were cultured in DMEM (Invitrogen) containing 10% Fetal Bovine Serum (Thermo Fisher), and 20% serum for SKMEL30 cells, 1% penicillin and streptomycin (Invitrogen), and 1 μg/ml plasmocin (Invivogen) at 37 °C, 5% $CO_2$. All cells were tested negative for mycoplasma using the Mycoalert™ mycoplasma detection kit (Lonza).

## Plasmids

Plasmids used in this study are listed in Supplemental Table S2. The CDS of human LYSET was purchased from the DNASU plasmid Repository (Arizona State University).

## Transient transfection

Cells were cultured in DMEM containing 10% (20% for SKMEL30 cells) serum media to 40-50% confluency before transfection. Cells were transfected with individual overexpression plasmids (2 μg DNA for a 3.5 cm dish) using either Lipofectamine 2000 (Invitrogen) or jetOPTIMUS transfection reagent from Polyplus (Sartorius) according to the manufacturer's protocol. Cells in 6-well plates were harvested 48 h post-transfection and lysed in 150 μl of buffer A (25 mM Tris-Cl, pH 7.2, 150 mM NaCl, 1% Triton-X 100 and protease inhibitor cocktail).

## Generation of lentiviral stable cell lines

HEK293T cells were transfected with transfer plasmid, psPAX2 (Addgene 12260), and pMD2.G (Addgene 12259) at a 3.5:3.5:1 ratio using Lipofectamine 2000 according to the manufacturer's instructions. 72 h after transfection, the supernatant was collected and passed through a 0.45 μm filter. HEK293T, SKMEL30, or HeLa cells were seeded in 6 cm dishes and infected with the collected supernatant. After 24 h of infection, the cells were kept in selection (1 μg/ml for puromycin, and 10 μg/ml for blasticidin) for at least 10 days before subsequent analysis.

## Generation of CRISPR-Cas9 KO and KI cell lines

*LYSET* and *GNPTAB* knockout cells were generated as described in Ran et al.[37]. In brief, sgRNA guides were ligated into pspCas9 (BB)−2A-Puro (Addgene, 48139) or Lenti-multi-CRISPR (Addgene 85402) plasmids. For single colony isolation, cells were transfected with CRISPR-Cas9 knockout plasmids using Lipofectamine 2000 according to the manufacturer's instructions. 24 h after transfection, cells were treated with 1 μg/ml puromycin (Invitrogen) for 2-3 days. Cells were then diluted into 96-well plates to a final concentration of 0.5 cell per well. The colonies were further screened and verified via western blot and/or sequencing analysis to confirm clean knockout colonies (Fig. S6).

The generation of *GNPTAB* knock-in (KI) cell line was described in Zhang et al[5]. A 300 bp homology arms (upstream and downstream from the stop codon) were amplified from the genomic DNA. The 3HA coding sequence was inserted in between the homology arms by overlapping extension. The resulting DNA fragment was ligated into the pGEM-T Easy vector and transfected (4 μg) into HEK293T cells together with CRISPR-Cas9 plasmid (2 μg). 24 h after transfection, cells were treated with 1 μg/ml puromycin (Invitrogen) for 2−3 days. Cells were then diluted into 96-well plates to a final concentration of 0.5 cell per well. The single colonies were screened by PCR using a 3HA internal forward primer and a reverse primer located 600 bp downstream of the stop codon. The candidate KI colonies were further verified by western blot and sequencing analysis.

The following reported sgRNAs and shRNAs were used in this study:

*LYSET* sgRNA1: 5′ – TGTCCACACCCAAAAAGGCA – 3′,

*GNPTAB* sgRNA1: 5′ – ACTCATTGCGATCTATCGAG – 3′,
*GNPTAB* sgRNA2 (KI): 5′ – CTTCTATACTCTGATTCGAT – 3′,
*VPS35* sgRNA1: 5′ –CTCAAGGGATGTTGCACACC– 3′
*GOLPH3* sgRNA1: 5′ – AGGGCGACTCCAAGGAAACG – 3′,
*GOLPH3L* sgRNA2: 5′ – GGATATCCGCCTTACTCTTA – 3′,
*GOLPH3* shRNA1: 5′ – GAGAAACAGAACTTCCTACTT – 3′

## Sample preparation and western blotting

Cells were collected in ice-cold 1X PBS, pelleted at 2700 g for 1 min, and lysed in lysis buffer (20 mM Tris pH 8.0, 150 mM NaCl, 1% Triton) containing protease inhibitor cocktail (Bimake) at 4 °C for 20 min. Cell lysates were centrifuged at 18,000 g for 15 min at 4 °C. The protein concentration of the supernatant was further measured by Bradford assay (Bio-rad) and normalized. After adding 2X urea sample buffer (150 mM Tris pH 6.8, 6 M Urea, 6% SDS, 40% glycerol, 100 mM DTT, 0.1% Bromophenol blue), samples were heated at 65 °C for 8 min before further analysis.

Normally, 30 µg of each lysate was loaded and separated on SDS-PAGE gels. Note that for the LYSET blot in Fig. S2, only 1/20 of the samples were loaded in the LYSET overexpression lanes. Protein samples were transferred to a nitrocellulose membrane for Western blot analysis. After incubation with primary and secondary antibodies, membranes were scanned using the Odyssey CLx imaging system (LI-COR) or developed with CL-XPosure film (Thermo Scientific).

The following primary antibodies were used for western blotting in this study: rabbit anti-GFP (1:3000, TP401, Torrey Pines Biolabs), mouse anti-actin (1:5000, 66009-1-Ig, Proteintech), mouse anti-GAPDH (1:2000, 60004-1-1 g, Proteintech), rabbit anti-CTSD (1:1000, 21327-1-AP, Proteintech), rabbit anti-LC3 (1:2000, 14600-1-AP, Proteintech), rabbit anti-HRD1 (1:4000, 13473-1-AP, Proteintech), rabbit anti-Golgin160 (1:1000, 21193-1-AP, Proteintech), mouse anti-HA (1:500, 16B12, BioLegend), mouse anti-CTSC (1:500, D-6, Santa Cruz Biotechnology), mouse anti-VPS35 (1:500, sc-374372, Santa Cruz Biotechnology), rabbit anti-LAPTM4A (1:1000, HPA068554-1, MilliporeSigma), rabbit anti-LYSET (1:1000, HPA-48559, Millipore-Sigma), mouse anti-V5 (1:3000, 46-0705, Invitrogen), rabbit anti-δ-COP (PA5-21484, Invitrogen), rabbit anti-VPS26 (1:1000, ab23892, Abcam), rabbit anti-GOLPH3 (1:2000, AB236296, Abcam), rabbit anti-β-COP (ab2899, Abcam), anti-GCN4 [IPI-C11L34] (1:1000, 218104-rAb, Addgene), mouse anti-LAMP2 (1:1000 H4B4-c, DSHB). Rabbit serum containing antibodies against the α-subunit of GNPTαβ was kindly provided by William Canfield, and the IgG fraction was purified using CaptivA Protein A affinity resin. The generation of single-chain antibodies against M6P was described in Zhang et al.[5,38].

The following secondary antibodies were used in this study: goat anti-mouse IRDye 680LT (926-68020), goat anti-mouse IRDye 800CW (926-32210), goat anti-rabbit IRDye 680LT (926-68021), goat anti-rabbit IRDye 800CW (926-32211). These secondary antibodies were purchased from LI-COR Biosciences and used at 1:10,000 dilution.

To detect LYSET, M6P (scFv M6P), GNPTαβ−3HA KI, or the uninduced 10xGCN4, the anti-protein A HRP (PA00-03, Rockland, for LYSET and M6P) or mouse HRP (115-035-046, Jackson labs, for GNPTαβ −3HA KI) secondary antibodies were used at 1:10,000 dilution. The signal was detected with the Pierce ECL kit (Thermo Scientific). All unprocessed blots are included in the Source Data file.

## Membrane isolation

The membrane isolation protocol was adapted from Shao and Espenshade[39], with some modifications. Essentially, cells with 70-80% confluency from a 10 cm dish were collected in ice-cold 1X PBS, and pelleted at 2700 g for 1 min. The pelleted cells were resuspended in 1 ml ice-cold membrane isolation buffer (1 mM EDTA and 1 mM EGTA in 1X PBS, with protease inhibitor) and homogenized. The homogenate was centrifuged at 900 g for 5 min at 4 °C, and the supernatant was transferred to a new tube and centrifuged at 20,000 g for 20 min at 4 °C to collect membranes. After centrifugation, the membrane pellet was further dissolved in lysis buffer (20 mM Tris pH 8.0, 150 mM NaCl, and 1% Triton) containing 1X protease inhibitor cocktail (Biomake) at 4 °C for 20 min. The undissolved membranes were removed by another round of centrifugation at 20,000 g for 15 min at 4 °C, and the protein concentration from the supernatant was measured by Bradford assay and normalized. Samples were incubated with 2X urea sample buffer samples at 65 °C for 8 min before western blot analysis.

## Phosphotransferase activity assays

The protocol for measuring the GlcNAc-phosphotransferase activity was adapted from van Meel et al.[13], with some modifications. Essentially, the cells were transfected at 50–60% confluency in six-well plates with a total of 2 µg plasmid DNA (pcDNA3.1 vector, *GNPTAB* cDNA, and *LYSET* cDNA in the indicated combinations) and 3 µL jetOPTIMUS transfection reagent according to the manufacturer's protocol. Two days post-transfection, the cells were harvested and lysed in 150 µl of buffer A. Whole cell lysate (50 µg) was assayed in a final volume of 50 µl in buffer containing 50 mM Tris (pH 7.4), 10 mM MgCl$_2$, 10 mM MnCl$_2$, 2 mg/ml BSA, 2 mM ATP, 75 µM UDP-GlcNAc, and 1 µCi UDP-[$^3$H] GlcNAc, with 100 mM α-methylmannoside (α-MM) as acceptor. The reactions were performed at 37 °C for 1 h, quenched with 950 µl of 5 mM EDTA (pH 8.0), and the samples applied to a QAE-Sephadex column (1 ml of packed beads equilibrated with 2 mM Tris, pH 8.0). The columns were washed twice, first with 3 ml of 2 mM Tris (pH 8.0), then with 2 ml of the same buffer. Elution was performed twice, first with 3 ml of 2 mM Tris (pH 8.0) containing 30 mM NaCl, then once again with 2 ml of the same buffer. 8 ml of EcoLite scintillation fluid (MP Biomedicals) was then added to the vials, and the radioactivity in the collected fractions was measured using a Beckman LS6500 scintillation counter. All CPM values that were obtained after subtracting buffer-only background were converted to pmol of GlcNAc-phosphate transferred to α-MM per hour per mg total protein lysate (pmol/hr/mg).

## Lysosomal enzyme assay

Soluble bovine cation-independent mannose 6-phosphate (CI-MPR), purified from fetal bovine serum[40]. was conjugated to CNBr-activated Sepharose 4B (Millipore-Sigma) as per the manufacturer's instructions. For lysosomal enzyme assays, 600 µg of whole cell lysates prepared from either WT HEK293T cells, or from untransfected or 48 h transfected LYSET knockout, GNPTαβ knockout, or double knockout cells were incubated with equivalent amounts of the CI-MPR affinity beads for 2 h at 4 °C, washed twice with cold Tris-buffered saline containing 1% Tx-100 (pH 7.4) (TBST), then assayed for bound lysosomal enzymes. Briefly, β-hexosaminidase (β-Hex), α-galactosidase (α-Gal) and β-galactosidase (β-Gal) were assayed in a final volume of 50 µl with 5 mM 4-methylumbelliferyl(MU)-N-acetyl-β-d-glucosaminide (Sigma-Aldrich), 5 mM 4-MU-α-d-galactopyranoside and 5 mM 4-MU-β-d-galactopyranoside (Calbiochem, San Diego, CA), respectively, in 50 mM citrate buffer containing 0.5% Triton X-100 (pH 4.5) at 37 °C. Reactions were stopped after 1 h by addition of 950 µl of 0.4 M glycine-NaOH (pH 8.0), samples were vortexed and spun down briefly, and the supernatant transferred to a round-bottomed glass tube. The fluorescence was then measured using a TURNER Model 450 Fluorometer (Barnstead Thermolyne Corporation, Dubuque, Iowa), with excitation and emission wave lengths of 360 and 450 nm, respectively.

## GST pull-down assays

In order to assess the binding of the LYSET N-tail to GOLPH3, GST-GOLPH3 was first expressed and purified from E. coli BL21 Codon-Plus (RIL) cells (Agilent). For binding assays, 100 µg of the purified GST-GOLPH3 fusion protein was first immobilized on 50 µl of packed glutathione-agarose beads at room temperature for 1 h in TBST. The beads were washed once with cold TBST, then 300 µl of whole cell

lysates prepared from cells transfected with the various GFP chimeras were added, and the beads tumbled for 2 h at 4 °C. The beads were washed 3 times with cold TBST, and the pellets were resuspended in sample buffer and heated at 100 °C for 10 min before SDS-gel loading.

## Secretion analysis

Cells were cultured to reach 70-80% confluency to collect secreted proteins from the media. They were washed with serum-free DMEM three times and incubated with serum-free DMEM for 16 h. The conditioned media were collected and centrifuged at 500 g for 5 min to remove cell debris, filtered with a 0.45 μm filter, and concentrated to ~100 μl using 10 kDa cutoff Amicon Centrifugal filters (Millipore-Sigma). The protein concentration from the concentrated media was measured by Bradford assay and normalized to its cell lysate. Samples were then mixed with 2X Urea buffer and heated at 65 °C for 8 min before further analysis. For the induced secretion assay in Fig. S1B, the cells were incubated with serum-free DMEM containing 10 mM $NH_4Cl$ for 16 h before collection and further processing.

## In-vitro deglycosylation

The deglycosylation assay was performed as described in Venkatar-angan et al.[41], with some modifications. For the whole cell lysate, cells were rinsed once with 1X ice-cold PBS and then harvested in 1 ml of PBS, split into three equal parts, and centrifuged at 1000 g, 1 min, 4 °C. For deglycosylation assay from the membrane fraction, the pelleted cells were resuspended in 1 ml ice-cold membrane isolation buffer (1 mM EDTA and 1 mM EGTA in 1X PBS, with protease inhibitor) and homogenized. The homogenate was centrifuged at 900 g for 5 min at 4 °C, and the supernatant was split into three equal parts, transferred to a new tube, and centrifuged at 20,000 g for 20 min at 4 °C to collect membranes. Two of the three pellets were resuspended in 40 μl of lysis buffer (20 mM Tris pH 8.0, 150 mM NaCl, 1% Triton, 1x PIC), while the third pellet was resuspended in 1X glyco-buffer 2 (50 mM sodium phosphate pH 7.5, 1% NP-40, 1X PIC). All pellets were allowed to lyse for 20 min at 4 °C. Lysates were subsequently centrifuged at 18,000 g, 15 min, 4 °C, and the protein concentration of the resulting supernatants was measured using Bradford assay. All supernatants were normalized to equal protein concentration. One lysate in the lysis buffer was used as an untreated control. To the second lysate in lysis buffer, 3.9 μl of 10X Denaturation buffer (5% SDS, 400 mM DTT) was added to 35 μl of lysate and boiled at 98 °C for 10 min. The lysate was then allowed to cool to room temperature for 5 min, and a further 4.7 μl of 10X glyco-buffer 3 (500 mM sodium acetate pH 6) and 2 μl of Endo H enzyme were added and incubated at 37 °C overnight. To the lysate in glyco-buffer 2, 10 μl of glyco-buffer 2 was added to 35 μl of the lysate, and 2 μl of PNGase F was further added to it and incubated at 37 °C overnight. After incubation, all lysates were mixed with an equal volume of 2X urea sample buffer, heated at 65 °C for 10 min before further analysis.

## Fluorescence protease protection (FPP) assay

HeLa cells were transfected with GFP tagged construct for 48 h and grown to a density

of about $1-5 \times 10^6$ cells/ml (~80% confluency for a 6 cm dish). The cells were resuspended, and 10 μL were loaded onto a cell microfluidic plate (Millipore-Sigma, CellASIC ONIX M04S-03)

through the cell loading sequence for adherence overnight in 37 °C, 5% CO2 incubator[42]. Cells

were first washed with KHM buffer (110 mM potassium acetate, 20 mM HEPES, 2 mM MgCl2)

for 2 min at 6 psi (41.4 kPa), then treated with 20 μM of digitonin for 3 min at 6 psi, and

with 50 μg/ml proteinase K at 6 psi. Images were captured on the DeltaVision imaging system (GE Healthcare Life Sciences) at intervals of 30 seconds.

## Immunostaining, microscopy, and image processing

Immunostaining was performed as described in Zhang et al.[38], with some modifications. Cells grown on 1.5 circular glass coverslips were washed with ice-cold 1X PBS and fixed in 4% paraformaldehyde for 10 min at room temperature[38]. Cells were permeabilized with 0.3% Triton in PBS for 15 min. The samples were blocked in 3% BSA for 30 min at room temperature, followed by incubating with primary and secondary antibodies. The cell nucleus was stained using Hoechst (1:1000, Invitrogen). Coverslips were mounted in Fluoromount-G (SouthernBiotech) and cured for 24 h before imaging.

The following primary antibodies were used for immunostaining in this study: rabbit anti-GNPT α subunit (1:100, homemade), rabbit anti-LYSET (1:100, HPA-48559, Millipore-Sigma), mouse anti-GM130 (1:100, 610822, BD Biosciences), and anti-GCN4 [IPI-C11L34] (1:100, 218104-rAb, Addgene), mouse anti-IL-2Ra (1:100, 05-170, Millipore-Sigma).

The following secondary antibodies were used at 1:100: FITC goat anti-rabbit (111-095-003, Jackson ImmunoReseach) and TRITC goat anti-mouse (115-025-003, Jackson ImmunoReseach), sheep anti-mouse (NA931, Cytiva) and donkey anti-rabbit (NA934, Cytiva).

Microscopy was performed with a DeltaVision system (GE Healthcare Life Sciences) or a Dragonfly Confocal Microscope System (Andor Technology). The DeltaVision microscope was equipped with a scientific CMOS camera and an Olympus UPLXAP0100X objective. The following filter sets were used: FITC (excitation, 475/28; emission, 525/48), TRITC (excitation 542/27; emission 594/45), and DAPI (excitation 390/18; emission 435/48). Image acquisition and deconvolution were performed with the SOFTWORX program. For spinning disk confocal microscopy, the following filter sets were used: FITC (excitation, 475/28; emission, 525/48), TRITC (excitation 542/27; emission 594/45), and DAPI (excitation 390/18; emission 435/48). Image acquisition and deconvolution were performed with the Fusion program. Images were further cropped or adjusted using ImageJ (National Institutes of Health). The Pearson coefficient analysis was performed using the FIJI ImageJ JACOP BIOP program.

## Magnetic isolation of lysosomes

HEK293T cells were seeded in 150 mm dishes containing DMEM with 10% FBS. Once they reached 40% confluency, cells were incubated in DMEM containing 10% FBS, 10 mM HEPES, 10% DexoMAG40 for 24 hours. Following this, cells were incubated in DMEM with 10% FBS for another 24 hours to allow the endocytosis of the magnetic nano-particles. Following the incubation, cells were rinsed twice and then harvested in 1X ice-cold PBS, using a cell scraper, and spun down at 60 g, 10 min, 4 °C. The supernatant was discarded, and the pellet was resuspended in 3 ml of homogenization buffer (HB: 5 mM Tris, 250 mM Sucrose, 1 mM EGTA in mass-spectrometry grade water (Thermo-Fisher), pH 7.4. HB was supplemented with protease inhibitor cocktail (Roche) immediately before use and degassed. The cell suspension was homogenized in a Dounce homogenizer for 30 strokes and then passed through a 22 G needle eight times. The homogenate was then centrifuged at 800 g, 10 min, 4 °C. The supernatant was collected, and 80 μl was taken as the whole cell lysate (WCL), mixed with equal volume of 2X urea buffer (150 mM Tris pH 6.8, 6 M Urea, 6% SDS, 40% glycerol, 100 mM DTT, 0.1% Bromophenol blue) and heated at 65 °C for 10 min. The remaining supernatant was allowed to flow freely through the magnetic LS column (130-042-401, Miltenyi Biotech). The column was pre-equilibrated with HB and attached to a platform that provided the magnetic field (130-091-051, Miltenyi Biotech). The column was then washed with 20 mL of HB using gravity. The columns were then detached from the magnetic field, and the lysosomes were eluted by flushing 1 ml of HB through the column using a syringe. The eluted lysosomes were further concentrated by spinning the eluate at 14,000 rpm, 1 h, 4 °C. The pelleted lysosome fraction was resuspended in 80 μl lysis buffer (20 mM Tris pH 8.0, 150 mM NaCl, 1% Triton, 1X

PIC) and heated at 65 °C for 10 min with an equal volume of 2X urea buffer.

## Quantification and statistical analysis

For the quantification of western blots, the band intensity was quantified using Image Studio software version 6.0 (LI-COR). For film, band intensity was calculated using ImageJ. Graphs were generated using Prism version 10.6.1 (GraphPad). Statistical analysis was performed with the two-tailed unpaired t-test with Welch's correction or one-way ANOVA with Tukey's or Dunnett's multiple comparison test. Error bars represent the standard deviation. Representative images and blots were independently repeated at least two times with similar results.

## Reporting summary

Further information on research design is available in the Nature Portfolio Reporting Summary linked to this article.

## Data availability

Data supporting this study is provided within the paper and supplementary files. Source data are provided with this paper.

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

## Acknowledgements

We thank the Li and Kornfeld laboratory members for their helpful discussion and technical support. This research is supported by the Protein Folding and Diseases Initiative, a MICHR Pathway Pilot grant, and NIH grants R01HD109346 to M. Li and R01CA008759 to S. Kornfeld.

## Author contributions

Conceptualization: X.Y., B.D., D.H, V.V., S.K. and M.L.; methodology: X.Y., V.V., D.H, B.D., B.J., L.C. and W.Z.; investigation: X.Y., B.D., D.H., V.V., Z.D, W.Z., L.C., B.J., L.Y., J.L. and B.Z.; writing & editing: V.V., B.D., D.H., X.Y. and M.L.; funding acquisition: M.L. and S.K.; resources & supervision: M.L and S.K.; X.Y., BD., D.H. and V.V. contributed equally, and all four have the right to list themselves first in bibliographic documents.

## Competing interests

S.K. was co-founder of M6P Therapeutics and held stock options in the company. The rest of the authors declare no competing interests.

## Dedication

We wish to dedicate this paper to the memory of Stuart Kornfeld (1936-2025) who devoted his life to advancing our understanding of the mannose-6-phosphate pathway.
