## [Transparent Peer Review file · Nature Communications]

Molecular Insights into the Regulation of GNPT $\alpha\beta$ by LYSET

Corresponding Author: Dr Ming li

Version 0:

Reviewer comments:

Reviewer #1

(Remarks to the Author)

The manuscript by Yang, et al. Kornfeld and Li describe some details on how the recently identified TMEM251- also termed LYSET controls the GlyNAc-1-phosphotransferase (GNPT) which is required for mannose 6 phosphorylation of many lysosomal hydrolases. LYSET/TMEM251 was discovered and characterized in two recent publications in Science (PMID: 36074821, 36074822) and NCOMM (PMID: 36096887). In the current study some confirmatory experiments are presented illustrating the essential role of TMEM251/LYSET for the M6PR-dependent delivery of acid hydrolases to the lysosomal compartment. It is confirmed (as outlined in detail below) that a lack of TMEM251/LYSET causes GNPT to be localized to lysosomes where it gets degraded most likely by residual active lysosomal proteases. Topology studies of TMEM251/LYSET suggest that both polypeptide termini are facing the cytosol. Two binding partners of TMEM251/LYSET were characterized, Golph3 and retromer, that may be contributing to Golgi localization of TMEM251/LYSET.

As such the study appears too preliminary and is in large parts not presenting novel information. Especially in the first part (Fig 1-4; results sections (lines 107 to 264) and statements in the introduction (lines 97-99)) of the study there is a general lack of conceptual advance. In its current version – with the focus in topology determination, alanine scan and binding partner description- I would see the manuscript much better suited for a specialized biochemical journal. The analysis of the TMEM251/LYSET topology, the reported alanine scan, potential interaction with GOLPH3 and retromer are new. However, it remains open what are the exact and specific roles of the newly described binding partners of TMEM251/LYSET.

Examples for a lack of novelty:

- o Loss of GNPTAB (Figs 2 and S2) was reported before (both Science papers)
- o Interaction of TMEM251 and GNPT was demonstrated before (both Science papers and NCOMM paper)
- o Mislocalization and lysosomal degradation of GNPT in TMEM251 KO cells was reported before
- o Impaired M6P tagging (Fig. 1E) in TMEM251 or GNPTAB KO was demonstrated in both Science papers before
- o Hypersecretion of proforms of cathepsin and lack of mature forms intracellularly was reported before in TMEM251 KO cells (Fig. S1B, C), accumulation of LC3 was previously reported (in the other two studies but also already by this group)

Concerns:

Figure 1 represents in part already knowledge presented earlier and relies on over-expression in HEK cells since apparently GNPT ptase activity was too low to be measured. Why were no experiments designed in a more suitable cell line allowing to endogenously monitor the consequences of a loss of TMEM251/LYSET or/and GNPT? The conclusion from the presented experiment that both TMEM251/LYSET and GNPT are crucial for the M6P pathway are not new. There is a strong claim that TMEM251 and GNPTAB steady state Golgi localization is achieved by vesicular recycling, but evidence is merely indirect (COP, Golph3, Retromer interactions).

Ptase activity assays (Fig 1B): Immunoblot panels for these transfected cells need to be included to (i) confirm TMEM251 KO and (ii) be able to judge whether GNPT expression levels correlate with detected activity; this similarly applies to Fig. 1C; in this case, however, data in Fig. 1 D appear to be from a similar experimental setup; it would be good to have panels 1C and 1D from one experiment, but they should then be included in one joint subpanel; this similarly applies to panels 3B and 3C; control immunoblots are also missing for Fig. S3 and S4 - this actually appears to be a more general issue; Ptase activity assays should be supplemented with immunoblot panels showing KO and overexpression controls.

The authors observed that high-level ectopic expression of GNPTAB leads to substantial Ptase activity also in TMEM251 KO cells (Fig. 1). What could be the underlying mechanism? Is the retention mechanism the authors described before (direct interaction of GNPTAB with COPI) sufficient at such high GNPTAB levels? Or is overexpressed GNPTAB still lost from the Golgi due to absence of TMEM251 and simply constantly replenished due to high overexpression reaching sufficient steady state Golgi levels? This should be addressed experimentally. Also, can overexpression of GNPTAB rescue the TMEM251 phenotype (e.g. cathepsin maturation defects)?

The lack of novelty holds - in large part - also true for the reported interdependent protein stability of TMEM251 and GNPT and the observation that TMEM251 is required for the processing efficiency and therefore enzymatic activity of GNPTab. The conclusions and interpretations of the data in Fig.2/3 have also been reported before. In Fig.2B it is reported that after HA-detection there should be an accumulation of a higher molecular weight fragment on the TMEM251KO cells. However, such a fragment cannot be seen.

Lysosomal degradation of GNPTab (and the explanation why GNPTab is largely invisible in TMEM251 KO cells) has also been previously demonstrated, i.e. the experiments here do not add much to the previous knowledge.

As an actual first new part of this study - in terms of a gain in knowledge - the determination of the discussed topology of TMEM251/LYSET is presented. With state-of-the-art technology and artificial introduction of a new glycosylation site (Fig.5C) and proteinase K protection assays (Fig.5E) it could be revealed that TMEM251 contains two transmembrane helices with both termini facing the cytosol. Studying 31 alanine mutants of TMEM251 and rescue of cathepsin D maturation in TMEM251 KO cells revealed ten mutants with an apparent loss of TMEM251 function. The two transmembrane regions and amino acids close to them appear especially critical for TMEM251 function. How the identified regions modulate GNPT binding and TMEM251 functions remain -however- unsolved.

This alanine scan is a comprehensive approach to identify sequences critical to TMEM251 function, but requires further controls. In Figure 6A abundance of mature cathepsin D is used as only (indirect!) readout for assessing the capacity of TMEM251 variants to rescue the TMEM251 KO phenotype in 293T cells. TMEM251 mutant expression levels need to be shown (and possibly be included in the quantification) here along with cathepsin D and actin levels, as it obviously matters a) whether TMEM251 mutants are expressed at all (an unstable TMEM251 mutant could be informative re: critical structural motifs) and b) whether their potency to rescue the KO phenotype is really different and not merely due to different expression levels. A selection of mutants found to display impaired TMEM251 function was subsequently tested in SKMEL30 cells (with notable differences in expression among the mutants!), but using a different readout (ability to rescue GNPTAB levels). Readouts in both cell lines should be the same (i.e. also GNPTAB levels need to be shown in 293T cells and cathepsin D levels in SKMEL30 cells) and SKMEL30 data should be similarly quantified to make any robust and cell line-independent conclusions. This is in particular required as the authors observe marked differences for e.g. 67-71A in both systems. In this particular case, it would also be interesting if localization differs between 293T and SKMEL30 cells.

In their study, the authors exclusively use TMEM251 isoform 2 (i.e. a short isoform with only 7 cytoplasmic amino acids preceding TMD1 (according to topology model 2)). The authors only briefly comment on this - "the more common short isoform of TMEM251" (line 301). It is indeed true that most cell lines appear to predominantly express this isoform at the endogenous level - but the SKMEL30 cells used quite extensively in this study are a notable exception that have a robust/dominant endogenous expression of isoform 1 (see Fig. 8A in this manuscript, but also data in Richards et al. and Pechincha et al.). Also, both the long and the short isoform were previously shown to be fully capable of rescuing a TMEM251 deficiency phenotype, i.e. are fully functional. That in mind, the authors should also check whether the critical motifs identified in their alanine scan of the short isoform also impair function of the long isoform and in particular their experiments with the Golgi localization reporter (Fig. 7) should be repeated with the longer N-terminus of the long TMEM251 isoform.

The N-terminal region of TMEM251 contained sufficient information to retain a GFP-plasma membrane reporter in the Golgi. Since the N-terminal sequence met the criteria for binding of a COP1 adaptor, Golp3 its interaction with the N-terminus of TMEM251 was demonstrated in a GST pull down approach. However, direct binding of Golp3 in the streptavidin-binding assay to the N-terminus of TMEM251 could not be shown. Importantly, no experiments are presented to show the importance of Golp3 (e.g. in KO cells) for lysosomal sorting and GNPT stability and function.

GFP reporter constructs like the one used in Figure 7 are well established tools to study the determinants of Golgi localization of type II membrane proteins. Indeed, the observations made here suggest that the first seven amino acids of TMEM251 could dictate a Golgi localization and potentially also an interaction with GOLPH3. However, this system is highly artificial for many reasons and the findings from these experiments should be taken with a grain of salt. TMEM251 is not a type II membrane protein and thus cannot be simply reduced to only its first 7 amino acids - in particular when the alanine scan data show that the 2-7A mutant still exhibits Golgi localization and instead mutation of other motifs impair Golgi localization. If this region is important for Golgi localization of TMEM251, shouldn't this mutant be mislocalized/lost from the Golgi apparatus? N-termini of type II Golgi membrane proteins are frequently rather short and Arg-rich - the capacity of TMEM251 aa 2-7 to direct the reporter to the Golgi may thus simply be a coincidence. Also, these experiments need to be further substantiated by constructs with TMEM251's longer N-terminus and additional mutants as well as full-length TMEM251.

VPS35 deletion (retromer member) did not affect TMEM251 expression levels but slightly reduced the level of GNPT as has been reported earlier. TMEM251 localization was apparently changed although the punctate Golgi-like structures remain unexplained. A fraction of TMEM251 could be found stabilized in lysosomes suggesting a retromer involvement in recycling

of missorted TMEM251.

The imaging data depicting the mislocalization of TMEM251 and GNPTAB in retromer-deficient cells needs to be assessed more quantitatively. There still appears to be a substantial portion of TMEM251 in the Golgi and also the single-dot Golgi staining of GNPTAB is not as evident as in Fig. 4. The authors also write that “the TMEM251-GNPT complex may exit the Golgi and mislocalize to endo-lysosomes” (line 412/413), but co-staining of TMEM251 and GNPTAB in retromer-deficient cells would be required to substantiate that indeed the TMEM251-GNPT complex is mislocalized (and not its two components independently). Also, the authors do see a destabilization/loss of GNPTAB under these conditions, but not of TMEM251. Is GNPTAB more prone to lysosomal degradation than TMEM251?

Quantification and statistical evaluation of multiple independent experiments and biological replicates are missing for most of the (immunoblot and microscopy-based) experiments.

Minor comments:

Different amounts of plasmid DNA (encoding in particular GNPTAB or TMEM251) were transfected transiently for numerous data panels to emphasize dose-dependent effects (e.g. of TMEM251 on GNPTAB stability). The figure legends should state that these absolute DNA amounts given in the figure panels apply to six-well culture dishes.

Figure legends are rather cryptic in general and lack important information here and there (e.g. in Fig. 4E and 4G inhibitor concentrations should be provided in the figure panels or the figure legends)

The authors write that all genome editing cell clones/lines (KO and KI) were validated/confirmed by Sanger sequencing. These data should be made available in the supplementary figures

Fig 2B: From the context it can be assumed that GNPT $\alpha\beta$ and GNPT β were detected with an anti-HA antibody in these KI cells, but that needs to be stated in the legend

Suppl. Fig. 2B – TMEM251 panel, between lane 3 and 4: Cutting/Splicing? Or why does TMEM251 in GNPTAB KOs exhibit a different running behavior compared to parental cells (cf to Fig.1E)

Contradictory statements: line 158 (section heading; “Tmem251 is required for stability of GNPT $\alpha\beta$) vs. line 189 to 191 (“In the absence of TMEM251, [...] the uncleaved GNPT $\alpha\beta$ level remains relatively unchanged”); in particular the latter statement does not appear to adequately reflect observations in e.g. Fig. 2C and Fig. S2B)

Line 244: WT cells should be TMEM251 WT cells (lanes 4 and 5 are GNPT KI cells, so no parental cells)

Lines 260 to 262 – reference to Fig. 4G missing in main text

Fig. 2E: Dot-like Golgi (-proximal) localization of GNPTAB upon TMEM251 deletion in SKMEL30 cells: This observation (in SKMEL30 cells) stands in contrast to observations by Richards et al. showing dot-like, but not really Golgi-proximal GNPTAB staining in TMEM251 KO 293FT cells. What is the nature of this unique dot-like structures? What happens if cells are treated with protease inhibitors, does GNPTAB then also accumulate in lysosomal structures (in light of the lysosomal enrichment data presented)?

Could GNPTAB mutants (K4Q, S14Y) shown previously to impair binding of GNPTAB to COPI also impair binding to TMEM251? (previously looked at?)

Line 384: “Deletion of VPS35 resulted in a significant reduction in endogenous GNPTab protein levels in both cell lines” – this statement is not overtly supported by the data presented. In SKMEL30 there is only a modest (at best) reduction and this would need to be properly quantified in light of this statement; the 293T data appear to be supporting this statement better, but quantification would be required here as well to exclude transfer/ECL issues

Fig. 5C – The E51N mutant generated to test the topology model appears to be not glycosylated quantitatively. Is this an issue with heavy overexpression or could this indicate the simultaneous presence of topologically different variants of TMEM251?

When transiently transfected, low-level overexpression of GNPTAB is insufficient to compensate for the lack of TMEM251 and a drastic difference in Ptase activity is observed between WT and TMEM251 KO cells (Fig. 1B, lane 5 vs 6). When stably expressing low-level GNPTAB-V5 variants (Fig. 3D), no notable difference in GNPTAB-V5 levels is evident between TMEM251 WT and TMEM251 KO cells (despite the latter clearly having a phenotype). How can that be explained?

Reviewer #2

(Remarks to the Author)

In this research, the authors conducted a thorough analysis of the role of TMEM251/LYSET protein in the processing, stability, and localization of GlcNAc-1-phosphotransferase 35 (GNPT), a key enzyme in the M6P pathway. TMEM251 is essential for human health and is a potential drug target for treating cancers and viral infections. While two studies in 2022 linked TMEM251 to the M6P pathway, its specific role remained unclear. The authors used gene editing, microscopy,

mutagenesis analysis, and in vitro methods to show that TMEM251 is crucial for the processing, activity, and maintenance of GNPT in the Golgi apparatus. They also conducted biochemical analysis and alanine mutagenesis scanning to determine the membrane topology of TMEM251 and identified critical residues in its cytosolic region that affect protein function and localization. The findings confirm the importance of TMEM251 and GNPT in the M6P pathway at the physiological level and support the hypothesis that GNPT's steady localization in the Golgi is maintained through vesicular recycling mediated by TMEM251, COPI, Golp3, and retromer. This work is suitable for publication in this journal, although several critical questions remain unanswered. Addressing these questions could further elevate the significance of the findings: What is the stoichiometry of the GNPT-TMEM251 complex? Is this a stable complex, or is it regulated temporally and spatially? Are other Golgi enzymes regulated by TMEM251? What is the molecular mechanism by which TMEM251 influences GNPT processing?

Specific comments and suggestions for improvement:

The study predominantly utilized knockout cells and overexpression conditions. This might lead to secondary defects, compensatory responses, and other artifacts. Implementing acute protein downregulation techniques and focusing on endogenous proteins—perhaps tagged using gene-editing methods—would significantly enhance the robustness of this study.

In Figure 1E, it is crucial to display a complete GNPTAB blot to confirm the effect of TMEM251 knockout on the processing of the endogenous enzyme in HEK cells. Additionally, clarification on the role and relevance of HRD1 in this context would be helpful.

TMEM251 KO cells showed mislocalization of GNPT, forming a "single-dot-like structure" at the Golgi rather than mislocalizing to lysosomes. It's crucial to investigate the nature of this structure. Immuno-electron microscopy (immune-EM) could be a suitable further exploration method.

The observation that in TMEM251 KO cells treated with BafA1, stabilized GNPT remains in the Golgi instead of moving to the lysosome is significant. The authors suggest that "BafA1 stabilizes GNPT by impeding its Golgi exit rather than inhibiting lysosomal degradation." This novel finding warrants further investigation to understand the underlying mechanisms.

The effects of GOLPH3 depletion (in the context of GOLPH3/GOLPH3L double KO) on the M6P pathway and the localization of TMEM251 and GNPT should be explored. This could provide insights into the regulatory mechanisms involving GOLPH3 in this pathway.

Reviewer #3

(Remarks to the Author)

The study by Yang, Doray and colleagues analyzes the role of TMEM251 in the activity of the GNPT enzyme which constitutes the fulcrum of the M6P pathway, and which is essential for the function of lysosomal enzymes. The most innovative part of the work concerns the role of TMEM251 in the GNPT traffic mechanisms. This is a work of considerable interest which, however, in its current form presents some important gaps of both an interpretative and experimental nature. The work is very complex and, as such, intrinsically difficult to analyze, but it is also written in a poorly organized way so that the logical connections between the parts are even more difficult to follow

Below we will try to analyze these weaknesses and where possible indicate how to overcome them.

Major points:

The authors describe the effects of TMEM251 on the stability of the GNPT enzyme and give evidence that TMEM251 is essential for the activating GNPT cleavage and is also essential for the stability of the cleaved portion of GNPT, in the sense that the levels of this protein decrease in the absence of TMEM251. In this regard the authors note that GNPT is intrinsically unstable due to the composition of its transmembrane domains, but in the following experiments the authors demonstrate that in the absence of TMEM251 the uncleaved GNPT is degraded in lysosomes. This is probably the reason for the instability of GNTP. The flow here is difficult to understand and the authors should logically connect the two data sets.

The authors show that TMEM251 is necessary for GNPT activity even after cleavage has occurred ie, after the activation step has been carried out. This is attributed to the physical interaction of TMEM251 with GNPT, which would form a stable complex which would have the effect of stabilizing the active conformation of the enzyme.

In the next part of the work, however, the authors demonstrate that TMEM251 is essential to prevent lysosomal degradation of both the cleaved and uncleaved enzyme. From what we understand, it is therefore probable that the property of TMEM251 to maintain the activity of the cleaved enzyme is simply due to the fact that it localizes GNPT at the Golgi preventing its lysosomal degradation. This possibility does not seem to be grasped by the authors. It is an ambiguity that the authors must resolve either with textual changes and/or with new experiments.

The authors perform an elegant series of experiments that leads them to the conclusion that TMEM251 interacts with Golp3 in the trans Golgi to maintain the localization of the TMEM251/GNPT complex in the cis Golgi and for the prevention of lysosomal degradation of GNPT.

A perplexing point here is the observation that mutants in the cytosolic tail of TMEM251 that abolish the binding between the tail and Golp3 do not change the localization of TMEM251 in the Golgi, in other words, that the interaction between TMEM251 and Golp3 is not essential for the retention of TMEM251 and GNPT in the Golgi. The authors suggest that there

may be additional regions beyond the cytosolic tail that maintain TMEM251 in the Golgi. This is possible, but the finding raises concerns that should be addressed with textual changes or new experiments.

The authors show evidence that the retention of GNPT in the Golgi requires both the binding of GNPT itself to COP1 via the motif previously identified by Doray and Kornfeld, and the binding between TMEM251 and Golph3, as shown in this study. Then, in the last paragraph, the authors propose the existence of a further mechanism for the retention of GNPT in the Golgi which would be based on the activity of the retromer, the complex that transports proteins from the endo-lysosomal system to the Golgi, which thus be responsible for maintaining the localization and functionality of GNPT/ TMEM251. This last conclusion is consistent with the data, which however are simply based on depletion and overexpression experiments of a protein crucial for the function of the retromer, VPS35.

Based on these collective evidence, the authors suggest that a triple recycling/retention mechanism is necessary for the maintenance of GNPT/ TMEM251 in the Golgi and present a model in figure 9.

The model shows a single retrotransport step from the trans to the cis Golgi. This is difficult to understand based on current knowledge and also inconsistent with the data shown in the manuscript and with the localization of the adaptor GOLPH3 within Golgi. The overall model could be modified by proposing a mechanisms of Golgi retention in three steps: first step of retrotransport from Golgi trans to medial cisternae due to the binding between TMEM251 and Golph3 which would then be followed by a second retrograde step dependent on the binding between the cytosolic tail of GNPT and COP1. The third step, mediated by the retromer, could recycle the fraction of GNPT / TMEM251 complex that escaped from the first two recycling mechanisms, from the endo-lysosomes to the trans Golgi. Here, GOLPH3 first and COP1 then could bring the complex back to the cis Golgi. Organized in this way the model could be coherent with the majority of the data and highlight the novelty of the paper.

Overall, we believe that the information brought by this study potentially represents a notable progress in an area of research which is currently under strong development and is of great importance, namely, the mechanisms of selective retrotransport of Golgi enzymes by specialized adaptors, with regulatory effects on glycosylation. However, the authors need to address the points we have raised with textual changes and new experiments.

Minor points:

- 1) One hypothesis put forward by the authors suggests that TMEM251 may serve as a bridging factor, linking GNPT with the S1P protease necessary for GNPT activation. It would be important to investigate the localization and levels of S1P in TMEM251 knockout cells.
- 2) While GOLPH3 is suggested as a mechanism for retaining the TMEM251-GNPT complex at the Golgi, its involvement remains unproven in GOLPH3 knockdown (KD) or knockout (KO) cells. The authors establish that the N-terminal tail of TMEM251 contains sufficient information for GOLPH3-dependent Golgi localization. They should define the motif. However, this is the sole evidence implicating GOLPH3 in TMEM251 retention at the Golgi, and using flow cytometry could quantitatively clarify this aspect.
- 3) Despite an interaction between the N-terminal tail and GOLPH3, the TMEM251 2-7A mutant, expected to lack GOLPH3 interaction, still localizes to the Golgi. The authors propose that additional regions beyond the N-terminal tail are responsible for retaining TMEM251 in the Golgi. However, the TMEM251 2-7A mutant fails to restore GNPT levels in TMEM251 KO cells despite its Golgi localization. The authors should discuss this discrepancy or address it with new experiments; perhaps the TMEM251 2-7A mutant is present in the Golgi but not in its precise subcellular location.
- 4) The authors suggest that there is a fraction of GNPT /TMEM251 which undergoes recycling back to the Golgi through the retromer machinery. They support this with findings that VPS35 KO results in TMEM251 and GNPT appearing in lysosomal puncta, where they degrade due to defective retromer function. However, a discrepancy arises in VPS35 KO, where TMEM251 remains at lysosomes but is not degraded, while GNPT is degraded. The authors should discuss this discrepancy or address it with new experiments. This might be due to the loss of proteases that degrade TMEM251 but not those that target GNPT.
- 5) The authors hypothesize that the TMEM251 cytosolic C-terminal tail, especially residues 93-96, might interact with the retromer for recycling back from endo-lysosomes to the Golgi, although co-immunoprecipitation failed to detect such interaction. Investigating the localization of the 93-96A mutant in lysosomes could further corroborate this hypothesis.
- 6) A topological issue arises from proposing the TMEM251-GNPT complex being recycled by GOLPH3 from trans Golgi to cis in Fig9. GOLPH3 is indeed localized at the trans-Golgi network (TGN). The authors could introduce the 3 steps model in the figure 9, which is more coherent with data they are showing.

Reviewer #4

(Remarks to the Author)

Version 1:

Reviewer comments:

Reviewer #1

(Remarks to the Author)

Reviewer Comments (Reviewer#1):

Thanks for revising the Yang et al. manuscript and trying to address my and the comments of the other reviewers. As outlined in more detail below and despite the attempts to argue that there is an apparent controversy in literature which has to be solved and that findings need to be repeated, I still remain reserved related to the conceptual advance and novelty presented in this manuscript. A large part of the previous manuscript, yet still a substantial part of the revised manuscript (Figs. 1 to 3) is dedicated to resolving the finer details of the LYSET-GNPT interplay. I have previously voiced that this particular part (on its own) does not represent a significant conceptual advance and do have to maintain this view. As suggested in my previous review I find the manuscript much more suited for publication in a specialized biochemical journal adding some extra information to the field. However, in a journal like NCOMM I would have expected a significant body of novel and unexplored findings which is -according to my opinion- not the case in this study.

The authors have now, though, included a paragraph (lines 148ff) that clarifies the situation a bit better and differentiates between their own previously reported data/model (“processing model”) and the data presented by two other independent papers (“trafficking model”). Indeed, resolving this controversy is of interest to the field of researchers interested in LYSET/GNPT biology, but in this particular case it also needs to be noted that the newly presented data rather argue in favour of the trafficking model proposed by others previously and do not strongly support the initial model proposed by the authors (Zhang et al., 2022, Nat Commun.). Indeed, the authors should more strongly voice to what extent their own data now support the model reported before by the other groups, e.g. in lines 193-195, line 217 and line 227/228. While the authors raise a point that some discrepancies exist among the three studies on the function of LYSET, it would be helpful to clarify that the primary point of contention lies with their own publication (Zhang et al., 2022, Nat Commun.), in which the role of LYSET was interpreted as the sole activator of MBTPS1-mediated cleavage of GNPTAB—an interpretation that remains under debate. The perceived “controversy” between the two back-to-back Science papers is relatively minor and may largely stem from methodological differences, such as the use of different cell lines, endogenous expression versus overexpression systems, and antibodies that detect either the alpha or beta subunits of GNPTAB. Other variables, such as subunit stability and processing of the precursor protein, may also contribute. Notably, the immunofluorescence data in Richards et al. (Fig. S14) are based on endogenous protein detection using an antibody targeting the alpha subunit. Following 24-hour Bafilomycin A1 treatment, a modest increase in Golgi-localized signal was observed. The lysosomal staining might reflect a pool of slowly degraded alpha subunit that had accumulated before Golgi exit was inhibited. Similarly, in Figure 3E, the 19-hour S1P inhibitor treatment showed only minor effects, likely due to the high stability of the endogenous protein. Since Pechincha et al. studied only newly synthesized, overexpressed protein, the perceived discrepancies may be less significant than suggested. As the authors themselves acknowledge, the precursor protein may exhibit different stability in the presence of lysosomal proteases. To conclusively demonstrate that the precursor is primarily degraded in lysosomes, experiments using lysosomal protease inhibitors would be necessary. Finally, as far as I can judge the claim that Pechincha et al. and Richards et al. exclusively showed lysosomal degradation of the cleaved GNPTAB subunits is not accurate. While Richards et al. predominantly detect the endogenous alpha subunit at steady state, they do not assert that cleavage is a prerequisite for degradation. The slightly higher molecular weight species observed in LYSET knockout cells may also include the precursor. Moreover, Pechincha et al. provide clear evidence of an increase in cleavage fragments upon treatment with lysosomal protease inhibitors, strongly indicating that the precursor itself undergoes lysosomal degradation.

(Introduction): I agree that LYSET seems to have multiple functions in regulating GNPTAB localization and therefore function. However, the authors neglect here that a stabilizing role of LYSET for the TMD1 of GNPTAB has been clearly shown. The authors themselves have (re-) demonstrated this in their recent MBoC publication (Doray et al. 2025). Showing the interaction of LYSET with the Golgi-adaptor GOLPH3, however, again lacks novelty, since this was shown by Brauer et al. before (EMBO J 2024) which the authors only briefly mention in their discussion.

Figure 1:

The new Figure 1 includes very few novel information but is primarily a reconfirmation of data published by Richards et al, Pechincha et al. and Zhang et al. The whole figure including panel H serving as a control for their newly generated antibody would be better suited for the supplement. In Figure S1, again, the authors present no novel data but replicate published results that never presented any controversy and could be explained both by LYSET-mediated activation, retention or stabilization of GNPTAB.

Figure 2:

Panel A: Quantification is missing. Since the overall expression of GNPTAB is reduced in LYSET KO, it is hard to judge whether there is a slight delay in MBTPS1- mediated processing.

Fig2B: A western blot should be added, otherwise it is impossible to distinguish between altered expression and altered activity. If the activity data is from three independent biological replicates, why is the standard deviation not shown? Single

data points including error bars should be presented.

Fig. 2E The authors have to show the absence of LYSET and the purportedly mild overexpression. The conclusion of the panel is potentially important since it would be the only piece of evidence that the phosphotransferase activity itself (at least of the mutants) might be affected by LYSET expression, so all the controls have to be shown. If LYSET is overexpressed anyway, there is no reason not to use a tagged version. At the same time, LYSET detection would be critical to e.g. take expression levels of LYSET into consideration for evaluating the various mutants generated (Fig. 5). This control is essentially missing. Considering the newly included findings, it would also be important to check levels of endogenous LYSET in GOLPH3/GOLPH3L-deficient cells

Figure 3:

Fig. 3D Is the whole cell lysate shown or a WCL depleted from lysosomes? How do the authors explain the low LAMP2 signal and the many (unspecific?) bands?

Figure 4:

Fig. 4E The labelling of the panels is misleading, please clearly indicate that all the proteinase K-treated conditions are also treated with Digitonin for 3 minutes. Scale bars are missing.

If 4F is really the quantification of the images shown in E, the timepoints of addition of digitonin and proteinase K are incorrectly labelled.

Figure 5:

Fig.5A/B How do the authors explain that some mutations in LYSET have completely different functions in 293T cells than in SK-MEL30 cells (e.g. 67-71A, which shows abnormal CTSD processing/localization, but WT GNPTAB levels)? For 87-91A or 93-96A this is less pronounced, but there is way more GNPTAB than in 82-86A but similar CTSD signals.

Fig. 5C: other topology predictors show a slightly different topology with R7 directly adjacent to the membrane (as in Fig. 8F), please indicate which tool was used and harmonize the different depictions.

R7W does not decrease LYSET levels (confirming the data in Richards et al. 2022), but decreases GNPTAB expression (as also shown in Richards et al.) The data in Fig.6F suggests that R7 is important for GOLPH3 binding so the mutant would be expected to destabilize LYSET levels. How do the authors explain this discrepancy?

In the initial version of the manuscript, the authors presented data demonstrating that the N-terminal cytosolic amino acids of the short LYSET isoform (i) can mediate binding to GST-GOLPH3 in vitro and (ii) enforce GOLPH3-dependent Golgi retention when added to N-terminally to an artificial type II membrane protein reporter. Overcoming the limitations of the reporter data which I pointed out earlier, the authors have now included experiments arguing in favour of a GOLPH3-dependent Golgi localization of even full-length LYSET, showing mislocalization of an ectopically expressed full-length LYSET in GOLPH3/GOLPH3L KO U2OS cells and mislocalized endogenous GNPT in GOLPH3 knockdown cells (Fig. S5). I suggest that these data are all included in one of the manuscript's main figures, although, as pointed out earlier, this finding is not novel since Brauer et al. (2024) have shown this before.

An additional concern is that the authors do not show mislocalization of endogenous LYSET. Also, as a consequence of LYSET mislocalization, impaired cellular M6P tagging and maturation of lysosomal enzymes can be expected (see also ref #36) and experiments in GOLPH3 KO HeLa and U2OS cells available to the laboratory (see Figure 6B, ref #21 and Fig. S5A) should be used to more clearly demonstrate this (instead of Fig. S5E/F).

In addition, through comprehensive mutagenesis, the authors determined that F4 and R7 present in the LYSET N-terminus are critical to Golgi localization of the type II membrane protein reporter, but – strongly supporting that these residues are indeed relevant here – also of full-length LYSET ectopically expressed in 293T and SKMEL30 cells (Fig. S4). Again, I feel that this dataset should be included in a main figure. In addition, it would be interesting if there is an additive effect of the two mutants and whether any of the mutants shows impaired binding to GST-GOLPH3 in vitro. I also still feel that it needs to be explored whether the F4 and R7 mutants exhibit mislocalization when in the context of the long LYSET isoform or whether the substantially longer N-terminus can somehow overrule those mutations (see Brauer et al. 2024).

Figure 7:

What's the difference between the endogenously tagged GNPTAB expression in Fig. 7A and B? Why is there so much background signal in the WCL of C? Why is there a precursor band detected in the 3HA-K1 (-) cells? Shouldn't the LYSET WCL levels be decreased if there's more LYSET mislocalized to lysosomes?

Figure 8:

Fig.8B the IF data are of rather bad quality and the GRASP55 signal seems to be -in part- extremely overexposed. Please select better quality images.

Fig. 8D, IF of poor quality, especially the WT LAMP2 staining. Please include higher quality (resolution) images. LYSET-GCN stainings seem to be overexposed which is precluding proper colocalization analysis (especially the lower panel of LAMP2 co-stainings).

The authors postulate a mechanism of LYSET binding both GOLPH3 and VPS35. Please show the expression of a LYSET mutant both deficient in LYSET and VPS35 binding.

Brauer et al. (see above) show that LYSET interacts with GOLPH3 and GOLPH3L. LYSET is mislocalized and is destabilized in double KO cells. The authors rather suggest VPS35 as being important for LYSET Golgi-localization. Are these cell-type-specific differences?

Figure 9:

The authors did not present data that convincingly shows two independent steps of coatamer-dependent recycling, although it's conceivable that there may be redundant mechanisms. Neither the figure legend nor the main text do explain the two distinct "coatamer-dependent recycling" boxes included in this illustration. This needs to be explained. Do the authors propose different modes of retrieving in the early and the late Golgi, with the latter being COPI-independent but GOLPH3-dependent? (with a label "coatamer component" still in the box?)

Again, the authors postulate a role of LYSET in GNPTAB cleavage, which is not unambiguously shown. The slightly increased processing might well be explained by a stabilisation of GNPTAB in the cis-Golgi apparatus and therefore longer dwell-time in the proximity of MBTPS1. In addition, the authors have demonstrated themselves in Doray et al. that the stabilising mutant of GNPTAB is efficiently cleaved by MBTPS1 in LYSET KO cells and stabilized upon overexpression of LYSET, so it is unclear to the reviewer why they deliberately omit this function in their summary figure.

Additional issues:

As a side note I could not get access (despite asking twice at the editorial board) to a version of the revised manuscript containing the marked changes which would be required to fully appreciate the authors changes and responses to my review.

Page 15 of the rebuttal letter: the authors claim that they cannot do costainings due to antibody issues, but they have endogenously tagged GNPTAB-HA cells which they could use.

Line 323ff. re: GOLPH3 recognition motif: The authors directly compare the FXXR motif found to be important experimentally with a previously suggested LXXR GOLPH3 recognition motif. However, as evident from the data presented by Welch et al. this motif is not necessarily present in all GOLPH3/3L clients and a polybasic stretch directly preceding the transmembrane domain of a Golgi type II protein may be a more accurate distinctive feature of a Golgi client. In fact, immediately after making this statement, the authors introduce GALNT2 as a prototypic GOLPH3 client which is used as their positive control – which completely lacks the LxxR motif (MRRRSR). This should be rephrased accordingly.

The authors now refer to TMEM251 as LYSET in the revised manuscript and it is good to further establish a uniform naming of the gene/protein in the literature. Yet, this makes it hard for a reader to understand labelling in Fig. 6H, in which "251N peptide" certainly refers to an N-terminal TMEM251 peptide. The authors should consider rephrasing the labelling of this panel to make this more obvious. Also, the exact sequence of the biotinylated peptide used should be provided.

Line 458: Brauer et al showed that GOLPH3 and GOLPH3L are redundant in maintaining LYSET and GNPTAB in the cis-Golgi. Were both deleted by the authors? Does the siRNA the authors used target both isoforms? Is the antibody used specific for GOLPH3 or does it equally detect GOLPH3L? Could this explain the compensatory signal in the KD cells?

Handwritten labelling can be found on some immunoblot panels. In particular in case of Fig. 2D it would be good to mask this (as lane numbering is provided below the panels anyway).

In panel 3A it would be good to highlight the degradation products mentioned with arrows.

Panel S3C appears to be very pixelated. Could a higher resolution panel be included in the figure?

Like in Fig. S4A, the box labelled EF1a in Fig. 8A should be additionally garnished with an arrow to clearly demonstrate it is the promoter driving the expression of the fusion construct.

Inconsistent naming of the detection tag used: The tag is mostly referred to as "10x GCN" but as 10x GNC4" (Fig. S4D) (with the latter naming corresponding to the one used by Tanenbaum et al. in the original manuscript reporting the tag)

How strong a reduction in GOLPH3 was achieved following knockdown in SKMEL30 cells (Fig. S5C) – such a control staining (GOLPH3 co-staining or immunoblot) is missing.

The disease mucopolipidosis type 5 does not exist. LYSET deficiency is officially called dysostosis multiplex, Ain-Naz type. Ain et al. 2021 is incorrectly cited by the authors.

Reviewer #2

(Remarks to the Author)

The revised manuscript is significantly improved, both in terms of new experimental data and clarity of writing. The authors

have adequately addressed all of my previous comments.

Reviewer #4

(Remarks to the Author)

Version 2:

Reviewer comments:

Reviewer #1

(Remarks to the Author)

As clearly and in detail stated in my previous reviews my enthusiasm about this manuscript remains dampened. It seems -in contrast to this reviewer's view- that the editor is in favor of promoting this manuscript despite a number of experimental suggestions were not addressed and only briefly discussed.

Some additional remarks:

"We are unsure how to answer this criticism because we did not acknowledge that "the precursor protein may exhibit different stability in the presence of lysosomal proteases." Please indicate the lines where we made such a statement. Instead, we found that LYSET is resistant to lysosomal degradation."

I referred in a long paragraph to the stability of GNPTAB and the authors' comment related to the stability of LYSET is misplaced and does not fit to my remark which should be thoroughly read and responded to.

"We thank the reviewer for pointing this out. This result strongly supports our conclusion that LYSET facilitates S1P-mediated GNPTAB cleavage. All three papers confirmed that the mRNA level of GNPTAB is not affected in LYSET KO cells. With the degradation of the precursor, one should expect to see reduced processing."

The authors again misinterpret the conclusion. While both Richards et al. and Pechincha et al. focus on the degradation of the cleaved subunits, they do not exclude that the precursor can equally be lysosomally degraded. Pechincha et al., however, demonstrate that the kinetics of GNPTAB cleavage are not altered in the absence of LYSET. Of course, the cleavage of the precursor is reduced if it is degraded. This is a trivial conclusion which does not support the authors' interpretation that LYSET facilitates the S1P-mediated cleavage. The authors seem uncertain to confirm their originally proposed role for LYSET as a cleavage-activating factor.

"WCL stands for whole cell lysate, which is the starting material used for lysosome purification. There might be two reasons for the many bands in LAMP2 blots: 1) LAMP2 is a heavily glycosylated protein. It always migrates as a smear or several bands. 2) It is not uncommon for antibodies to recognize additional, non-specific bands. In the "LYS" fraction, however, the two non-specific bands, were removed by lysosome purification, confirming lysosome purification can remove non-specific bands from the WCL."

There are several excellent LAMP2 antibodies (e.g. from DSHB) that detect no unspecific bands, therefore the reviewer is surprised by the poor quality of the western blot.

Version 3:

Reviewer comments:

Reviewer #1

(Remarks to the Author)

The manuscript by Yang and colleagues describes the recently identified LYSET protein as a two-pass Golgi membrane protein that stabilizes GNPT, promotes its S1P-dependent cleavage, and prevents its degradation by protecting it from mislocalization to lysosomes. The authors suggest that LYSET achieves this function by interacting with GOLPH3 and the retromer—through distinct LYSET motifs—to retain the LYSET–GNPT complex in the Golgi, thereby ensuring proper operation of the M6P pathway.

As stated multiple times in my earlier reviews, I acknowledge and appreciate certain new findings and contributions to the field. However, I find that the overall scientific advancement and impact compared to previously published studies is not as substantial as claimed. A journal such as Communications Biology or a comparable publication would likely be more appropriate for this work.

Upon re-examining the manuscript, I also noted that, despite my previous comments, some data presentations still rely on only two measurements (e.g., Fig. 2E and the activity determinations in Supplementary Fig. S1). Moreover, no error bars or

standard deviations (STDEVA) are provided.

Reviewer 1

The manuscript by Yang, et al. Kornfeld and Li describe some details on how the recently identified TMEM251- also termed TMEM251 controls the GlyNAc-1-phosphotransferase (GNPT) which is required for mannose 6 phosphorylation of many lysosomal hydrolases. TMEM251/TMEM251 was discovered and characterized in two recent publications in Science (PMID: 36074821, 36074822) and NCOMM (PMID: 36096887). In the current study some confirmatory experiments are presented illustrating the essential role of TMEM251/TMEM251 for the M6PR-dependent delivery of acid hydrolases to the lysosomal compartment. It is confirmed (as outlined in detail below) that a lack of TMEM251/TMEM251 causes GNPT to be localized to lysosomes where it gets degraded most likely by residual active lysosomal proteases. Topology studies of TMEM251/TMEM251 suggest that both polypeptide termini are facing the cytosol. Two binding partners of TMEM251/TMEM251 were characterized, Golph3 and retromer, that may be contributing to Golgi localization of TMEM251/TMEM251.

As such the study appears too preliminary and is in large parts not presenting novel information. Especially in the first part (Fig 1-4; results sections (lines 107 to 264) and statements in the introduction (lines 97-99)) of the study there is a general lack of conceptual advance. In its current version – with the focus in topology determination, alanine scan and binding partner description- I would see the manuscript much better suited for a specialized biochemical journal. The analysis of the TMEM251/TMEM251 topology, the reported alanine scan, potential interaction with GOLPH3 and retromer are new. However, it remains open what are the exact and specific roles of the newly described binding partners of TMEM251/TMEM251.

We must respectfully disagree with this reviewer on the novelty and conceptual advance. LYSET/TMEM251 was reported in 2022, by three independent groups (Pechincha et al., 2022a; Richards et al., 2022; Zhang et al., 2022). All three agreed that TMEM251 plays a critical role in the M6P pathway by regulating GNPT, but the underlying mechanism is under debate. In addition, there were inconsistencies in the data between the two Science papers (Pechincha et al., 2022a; Richards et al., 2022). It is imperative that this controversy be resolved before the field can move forward, which is what the first part of our study has accomplished. More importantly, our paper reveals that both GOLPH3 and retromer are crucial to the TMEM251 Golgi localization, serving as the underlying mechanism for maintaining GNPT at the Golgi, which is a novel and mechanistic finding.

Examples for a lack of novelty:

Although we appreciate constructive criticism and aim to incorporate all valid suggestions in order to strengthen the manuscript and contribute good science to the field, Reviewer 1's comments make it difficult to do so. The reviewer has overlooked the careful work we have invested in resolving the current controversy and elucidating the detailed mechanism of TMEM251-GNPT $\alpha\beta$ regulation (**Fig. 1-4**). Additionally, the amount of work suggested by Reviewer 1 for the subsequent figures (**Fig. 5-8**) is unfeasible and would not provide additional information to this study. Nevertheless, we have made a huge effort, spending more than a year, to address their concerns and provide valid reasons refuting their claims as outlined below.

o Loss of GNPTAB (Figs 2 and S2) was reported before (both Science papers)

Indeed, both Science papers showed lysosomal degradation of GNPT $\alpha\beta$ in the absence of TMEM251, but their model for how GNPT $\alpha\beta$ degrades is not entirely correct. Specifically, they claimed that GNPT $\alpha\beta$ processing at the Golgi is unaffected by pulse-chase experiments. After processing, only the mature subunits (both α and β) will mislocalize to the lysosome and degrade.

In contrast, by pulse-chase experiment, we showed that the processing efficiency of GNPT $\alpha\beta$ is low in *TMEM251 KO* cells (Fig. 3A, now **Fig. 2A**), and both processed and unprocessed GNPT $\alpha\beta$ are mislocalized to the lysosome for degradation (Fig. 4D-E, now **Fig. 3D-E**). Therefore, the current manuscripts highlight that both the processing and localization of GNPT $\alpha\beta$ are affected in *TMEM251 KO* cells.

o Interaction of TMEM251 and GNPT was demonstrated before (both Science papers and NCOMM paper)

Indeed, all three groups showed this interaction before; thus, we **did not** show the interaction again in the previous submission. Therefore, we believe Reviewer 1's comment is unfounded.

o Mislocalization and lysosomal degradation of GNPT in TMEM251 KO cells was reported before

This is a repetitive comment and has been addressed above.

o Impaired M6P tagging (Fig. 1E) in TMEM251 or GNPTAB KO was demonstrated in both Science papers before.

We have moved it to Supplementary **Figure 1A**. Please also see the following response

o Hypersecretion of proforms of cathepsin and lack of mature forms intracellularly was reported before in TMEM251 KO cells (Fig. S1B, C), accumulation of LC3 was previously reported (in the other two studies but also already by this group)

Indeed, **Fig. 1E** and **S1B-C** have been shown before. Since the purpose of **Fig. 1** was to resolve the controversy related to TMEM251, it was necessary to first verify some of the key results from previous studies. However, we must emphasize that **Fig. 1** was not a simple repeat of published results. Using biochemistry, we measured the enzymatic activities of PTase as well as the direct M6P phosphorylation of several representative lysosomal enzymes (Previous Fig. 1B-D, and S1A), which are new and independently verify the importance of TMEM251. In addition, we included the phenotype of *TMEM251 GNPTAB* double KO, which was also new data. The purpose of **Fig. 1** was to familiarize the reader with GNPT $\alpha\beta$ and TMEM251's role in the M6P pathway and show that, at physiologically relevant levels, both proteins are required. In summary, the majority of **Fig. 1** presents unpublished data, and we disagree with Reviewer 1's conclusion that Fig. 1 lacks novelty. However, to satisfy the reviewer, we have removed most of Fig. 1 and moved Fig. 1E to the supplementary (now **Fig. S1A**).

Concerns:

Figure 1 represents in part already knowledge presented earlier and relies on over-expression in HEK cells since apparently GNPT ptase activity was too low to be measured. Why were no experiments designed in a more suitable cell line allowing to endogenously monitor the consequences of a loss of TMEM251/TMEM251 or/and GNPT? The conclusion from the presented experiment that both TMEM251/TMEM251 and GNPT are crucial for the M6P pathway are not new.

We have addressed Reviewer 1's concerns about the apparent lack of novelty in **Fig. 1** above.

Reviewer 1 misconstrued the findings of **Fig. 1**. We highlight that TMEM251 is required for GNPT activity at closer to endogenous levels (~0.4 μg cDNA), but that overexpression (0.8-2 μg cDNA) can largely bypass this requirement. By using overexpression, we could control the amount of GNPT/TMEM251 expressed in the cell and thus show the dose-dependent increase in activity that occurs. No other labs, including ours, have shown this before.

We have a paper in press in STAR Protocols (Doray et al., 2025) on how to measure the endogenous activity of GNPT in SKMEL30 cells. There, we show that GNPT activity is reduced in both *GNPTAB* KO and *TMEM251* KO cell lines. Additionally, we demonstrate that the endogenous activity of GNPT in SKMEL30 cells is approximately eight times higher compared to HEK293T cells.

There is a strong claim that TMEM251 and GNPTAB steady-state Golgi localization is achieved by vesicular recycling, but evidence is merely indirect (COP, Golph3, Retromer interactions).

We completely disagree with this statement. Our conclusion regarding vesicular recycling is supported by multiple experiments, and the evidence we provided is strong and compelling.

- We showed that the N-terminal tail of TMEM251 contains sufficient information to maintain this construct in the Golgi (Fig. 7A-B, now **Fig. 6A-C**) using a well-established SITM-GFP fusion strategy in the field, as acknowledged by this reviewer in a later comment (Liu et al., 2018; Welch et al., 2021).
- We showed the N-terminal tail of TMEM251 is maintained in the Golgi in a GOLPH3-dependent manner using immunofluorescent imaging (Fig. 7E, now **Fig. 6A-C**) and that the N-terminal tail directly interacts with GOLPH3 using an *in vitro* binding assay (Fig. 7D, now **Fig. 6G**)
- We demonstrated that the Retromer complex is important for maintaining TMEM251 and GNPT $\alpha\beta$ protein levels in the Golgi using western blot (**Fig. 7A-B**), lysosome purification (**Fig. 7C**), and immunofluorescent imaging (Fig. 8C, now **Fig. 7D-E**).

However, to satisfy the reviewer, we have since provided additional evidence to further support the vesicular recycling of TMEM251.

- We determined the critical residues (F⁴XXR⁷) in the TMEM251 N-terminal for GOLPH3-mediated Golgi retention (**Fig. 6D-F, Fig. S4A-E**).
- We provided further evidence that TMEM251 is maintained in the Golgi via GOLPH3 by investigating the colocalization between the Golgi and TMEM251/GNPT $\alpha\beta$ in U2OS and SKMEL30 cell lines lacking GOLPH3 (**Fig. S5A-D**).
- We showed that lysosome enzyme maturation is affected in HeLa GOLPH3 KD cells (**Fig. S5E-F**)
- Using another well-established IL-2R α chimera strategy, we showed that the C-terminal of TMEM251 contains enough information to confer Golgi localization and that this localization is dependent on VPS35 (**Fig. 8A-C**).
- Finally, we narrowed down the likely binding sequence of the retromer in the C-terminal of TMEM251 (V⁹⁷XY⁹⁹---V¹⁰⁸IC¹¹⁰) (**Fig. 8D-F**)

In summary, our vesicular recycling model provides a well-thought-out explanation for why TMEM251 is necessary to maintain GNPT in the Golgi. We demonstrate that, unlike what was claimed by Pechincha et al., TMEM251 does more than passively shield the charged residue in the transmembrane region of the GNPT enzyme.

Ptase activity assays (Fig 1B): Immunoblot panels for these transfected cells need to be included to (i) confirm TMEM251 KO and (ii) be able to judge whether GNPT expression levels correlate with detected activity; this similarly applies to Fig. 1C; in this case, however, data in Fig. 1D appear to be from a similar experimental setup; it would be good to have panels 1C and 1D from one experiment, but they should then be included in one joint subpanel; this similarly applies to panels 3B and 3C; control immunoblots are also missing for Fig. S3 and S4 - this actually appears to be a more general issue; Ptase activity assays should be supplemented with immunoblot panels showing KO and overexpression controls.

To improve the flow of the paper we have removed or redistributed most of the data from Fig. 1B and Fig. S3.

Both Fig. 1C-D (Previous Figures 1C, D, see below), as well as Fig. 3B-C (now **Fig. 2C-D** in the revised manuscript) were from the same experiment. We have included a sentence in the figure legends of Fig. 3B-C (now **Fig. 2C-D**) to make this more obvious. Thank you!

The activity data from **Fig. 2B** is from three independent transfections (i.e., biological replicates), while the data from **Fig. S2A-B** is from two independent transfections.

Previous Figure 1C, D – TMEM 251 is essential for the Ptase activity of GNPT α β at physiological levels, but a high-level expression of GNPT α β can overcome the requirement. (C) In TMEM251 KO cells, the enzyme activities of GNPT α β and β Hexosaminidase can be restored by either gradually increasing the transfected GNPT α β cDNA or co-transfecting TMEM251. Values shown are the average of two assays from two independent transfections. (D) Western blot showing the expression levels of GNPT α β and its processing efficiency under the same experimental setup as (C).

The authors observed that high-level ectopic expression of GNPTAB leads to substantial Ptase activity also in TMEM251 KO cells (Fig. 1). What could be the underlying

mechanism? Is the retention mechanism the authors described before (direct interaction of GNPTAB with COPI) sufficient at such high GNPTAB levels? Or is overexpressed GNPTAB still lost from the Golgi due to absence of TMEM251 and simply constantly replenished due to high overexpression reaching sufficient steady state Golgi levels? This should be addressed experimentally.

This is an excellent question, but it was already answered in Fig. 7F (See Previous Figure 7F below) and lines 360-373 of our initial submission. In that figure, we showed that overexpressed WT GNPT $\alpha\beta$ has a high PTase activity (1997 pmol/hr/mg protein) in *TMEM251* KO cells. In contrast, the overexpressed K4Q mutant, which nearly abolishes the interaction between GNPT and COPI, had very little PTase activity (164 pmol/hr/mg protein, a 92% reduction). This result indicated that overexpressed GNPT $\alpha\beta$ still depends on its interaction with COPI to be maintained in the Golgi. Importantly, when *TMEM251* is present, we observed stimulation of PTase activity even for K4Q mutant (716 pmol/hr/mg protein, 4.4-fold, Previous Fig. 7F). Our data indicated that COPI and *TMEM251* work independently and synergistically to maintain GNPT at the Golgi. For better cohesion, we have since removed Fig. 7F, as well as most of Fig. 1.

Previous Figure 7F –Comparing PTase activities between WT GNPT $\alpha\beta$ and K4Q/ S15Y mutants, with or without *TMEM251*. Values shown are the average of two assays from two independent transfections.

*Also, can overexpression of GNPTAB rescue the *TMEM251* phenotype (e.g. cathepsin maturation defects)?*

Yes, we showed in Fig. 1C (See Previous Figure 1C above) that overexpression of GNPT $\alpha\beta$ restores β -Hex activity in a dose-dependent manner. In addition, the PTase

activity is also increasing, bypassing the requirement of TMEM251. As mentioned before, we have removed the majority of the original Fig. 1.

The lack of novelty holds - in large part – also true for the reported interdependent protein stability of TMEM251 and GNPT and the observation that TMEM251 is required for the processing efficiency and therefore enzymatic activity of GNPT. The conclusions and interpretations of the data in Fig.2/3 have also been reported before.

As we emphasized above, one important aim of this paper was to resolve the controversy. The major argument was whether the GNPT $\alpha\beta$ cleavage (or processing) was affected by TMEM251 deletion. In the previous NC paper, we proposed that GNPT $\alpha\beta$ processing was **defective** without TMEM251. However, the other two Science papers stated that the processing was **normal**, and after processing, the cleaved subunits would mislocalize to the lysosome and degrade. Using a pulse chase experiment, they observed no difference in GNPT $\alpha\beta$ processing after knocking out TMEM251 (Pechincha et al., 2022a).

To resolve the controversy, it is necessary to re-examine the processing and localization of **endogenous** GNPT $\alpha\beta$ again, which is why we generated Fig. 2 (now **Fig. 1**). As one can tell, the processed β - (Fig. 2B, now **Fig. 1A**) and α -subunits (Fig. 2C-D, now **Fig. 1F**) were largely abolished, supporting that the processing might be defective. Interestingly, we did not observe the mislocalization of GNPT $\alpha\beta$ by immunostaining. Instead, we observed only a reduction in endogenous GNPT $\alpha\beta$ at the Golgi (Fig. 2E, now **Fig. 1H**). This is very different from what was reported by Richards et al. (Fig. 4A and S14G of their paper) and similar to the results obtained by Pechincha et al. (Fig. 6F of their paper). Again, when there is a controversy in the literature, it is necessary to carefully repeat and report the findings, instead of stating “it is not novel.” *In an era when more than 50% of biomedical research has reproducibility issues (Baker, 2016; Errington et al., 2021), we sincerely hope this reviewer and the editor do not punish careful reevaluation of controversial literature.*

We also performed a pulse-chase experiment to directly examine the processing of newly synthesized GNPT $\alpha\beta$. As shown in Fig. 3A (now **Fig. 2A**), the processing of GNPT $\alpha\beta$ is markedly reduced after knocking out TMEM251. This is inconsistent with Fig. S10 in the Science paper published by Pechincha et al. The field deserves to know the inconsistency.

Fig. 3B-C (now **Fig. 2C-D**) is novel information. In Fig. 3B-3C (now **Fig. 2C-D**), we demonstrated that overexpression of TMEM251 can stimulate the processing of GNPT $\alpha\beta$ and enhance its PTase activity in a dose-dependent manner. Fig. 3D (now **Fig. 2E**) demonstrated that even after the processing, TMEM251 is still required for PTase activity. We are not aware that any group has done these sets of experiments before.

In Fig. 2B it is reported that after HA-detection there should be an accumulation of a higher molecular weight fragment on the TMEM251KO cells. However, such a fragment cannot be seen.

This statement is incorrect. A clear upper band is visible in Fig. 2B (now **Fig. 1A**) just above the FL GNPT. Similarly, an upper band is also visible in Fig. 3A (now **Fig. 2A**) in the *TMEM251* KO cell line. For clarity, we have added an arrow indicating the upper band in **Fig. 1A** and **Fig. 2A** and mentioned it in the figure legends.

Lysosomal degradation of GNPTab (and the explanation why GNPTab is largely invisible in TMEM251 KO cells) has also been previously demonstrated, i.e. the experiments here do not add much to the previous knowledge.

Although both Science papers concluded that GNPT $\alpha\beta$ mislocalizes to the lysosome for degradation after knocking out *TMEM251*, we missed this important conclusion in our initial NC paper. *It was therefore necessary to verify this result independently (Fig. 4A, now Fig. 3A).*

Therefore, because we never observed the mislocalization of GNPT $\alpha\beta$ to the lysosome by immunostaining (Fig. 2E, now **Fig. 1H** vs. Fig. 4A of Richards et al.), we relied on the biochemical purification of lysosomes to test the mislocalization (**Fig. 3D**). In fact, we repeated the GNPT $\alpha\beta$ immunostaining in HAP1 cells, as reported by Richards, but were unable to detect any GNPT $\alpha\beta$ signal.

Additionally, both Science papers showed that BafA1, but not MG132, can stabilize GNPT $\alpha\beta$. However, it is essential to point out that, even though the two science papers are consistent in their conclusion, their BafA1 treatment results were contradictory and confusing. Richards et al. showed that the BafA1-stabilized GNPT $\alpha\beta$ is localized to the lysosome (Fig. S14F-G). In striking contrast, Pechincha et al. showed that BafA1-stabilized GNPT $\alpha\beta$ is actually localized to the Golgi (Fig. S10E).

In Fig. 4D (now **Fig. 3D**) we observed a striking accumulation of **both unprocessed and mature** GNPT $\alpha\beta$ subunits and their degradation products in the lysosome. It was assumed that only processed mature subunits would mislocalize. Our data in Fig. 4D-E (now **Fig. 3D-E**) clarifies that this is NOT the case because, in the presence of a Site-1-protease inhibitor, unprocessed GNPT $\alpha\beta$ still mislocalizes to the lysosome.

Lastly, our immunostaining of BafA1-treated cells in Fig. 4G (now **Fig. 3G**) demonstrated that GNPT $\alpha\beta$ is stabilized at the Golgi, supporting the observation by Pechincha et al.

Thus, Fig. 4 (now **Fig. 3**) not only clarifies the previous controversy that arose between our previous Nature Comm. and the two Science papers but also clarifies the discrepancy between the two Science papers.

As an actual first new part of this study - in terms of a gain in knowledge - the determination of the discussed topology of TMEM251/TMEM251 is presented. With state-of-the-art technology and artificial introduction of a new glycosylation site (Fig.5C) and proteinase K protection assays (Fig.5E) it could be revealed that TMEM251 contains two transmembrane helices with both termini facing the cytosol. Studying 31 alanine mutants of TMEM251 and rescue of cathepsin D maturation in TMEM251 KO cells revealed ten mutants with an apparent loss of TMEM251 function. The two transmembrane regions and amino acids close to them appear especially critical for TMEM251 function.

We thank Reviewer #1 for the positive comments.

How the identified regions modulate GNPT binding and TMEM251 functions remain - however- unsolved.

This is true; however, determining the membrane topology and critical regions within TMEM251 are important first steps. Figuring out the molecular basis for all 10 critical regions will take years to complete and is beyond the scope of this paper. In our current manuscript, we focus on the molecular basis for maintaining GNPT $\alpha\beta$ in the Golgi.

This alanine scan is a comprehensive approach to identify sequences critical to TMEM251 function, but requires further controls. In Figure 6A abundance of mature cathepsin D is used as only (indirect!) readout for assessing the capacity of TMEM251 variants to rescue the TMEM251 KO phenotype in 293T cells. TMEM251 mutant expression levels need to be shown (and possibly be included in the quantification) here along with cathepsin D and actin levels, as it obviously matters a) whether TMEM251 mutants are expressed at all (an unstable TMEM251 mutant could be informative re: critical structural motifs) and b) whether their potency to rescue the KO phenotype is really different and not merely due to different expression levels.

We thank Reviewer #1 for these thoughtful suggestions. The goal of the alanine scanning experiment was to identify functionally important regions of TMEM251, using CTSD maturation as an indirect readout of TMEM251 activity. Unfortunately, due to the lack of a reliable TMEM251 antibody, we are currently unable to determine TMEM251 protein expression levels as suggested. Unfortunately, the new batch of Sigma TMEM251 antibody (HPA048559) stopped working.

A selection of mutants found to display impaired TMEM251 function was subsequently tested in SKMEL30 cells (with notable differences in expression among the mutants!), but using a different readout (ability to rescue GNPTAB levels). Readouts in both cell lines

should be the same (i.e. also GNPTAB levels need to be shown in 293T cells and cathepsin D levels in SKMEL30 cells) and SKMEL30 data should be similarly quantified to make any robust and cell line-independent conclusions. This is in particular required as the authors observe marked differences for e.g. 67-71A in both systems. In this particular case, it would also be interesting if localization differs between 293T and SKMEL30 cells.

We agree that, in theory, measuring GNPT $\alpha\beta$ levels in HEK293T cells would provide a more direct readout. However, this approach is technically challenging given the number of mutants analyzed. Each sample requires at least one 10-cm dish for total membrane isolation, and performing this for all 31 cell lines with a minimum of three biological replicates would require at least 93 membrane preparations, which is not feasible.

Instead, we utilized the SKMEL30 cell line, which expresses higher endogenous levels of both TMEM251 and GNPT $\alpha\beta$, to focus on a subset of critical mutants for further validation. Importantly, the results from Fig. 6A (now **Fig. 5A**) and Fig. 6C (now **Fig. 5B**) are largely consistent: mutants that showed impaired CTSD maturation in Fig. 5A also exhibited reduced GNPT $\alpha\beta$ levels in **Fig. 5B**, with the exception of 67–71A.

In conclusion, we respectfully argue that performing the full experimental set proposed by Reviewer 1 (i.e., two independent readouts across two cell lines) would not yield additional mechanistic insights beyond what is already presented. We believe our efforts are better directed toward strengthening other aspects of the study.

In their study, the authors exclusively use TMEM251 isoform 2 (i.e. a short isoform with only 7 cytoplasmic amino acids preceding TMD1 (according to topology model 2)). The authors only briefly comment on this – “the more common short isoform of TMEM251” (line 301). It is indeed true that most cell lines appear to predominantly express this isoform at the endogenous level – but the SKMEL30 cells used quite extensively in this study are a notable exception that have a robust/dominant endogenous expression of isoform 1 (see Fig. 8A in this manuscript, but also data in Richards et al. and Pechincha et al.). Also, both the long and the short isoform were previously shown to be fully capable of rescuing a TMEM251 deficiency phenotype, i.e. are fully functional. That in mind, the authors should also check whether the critical motifs identified in their alanine scan of the short isoform also impair function of the long isoform and in particular their experiments with the Golgi localization reporter (Fig. 7) should be repeated with the longer N-terminus of the long TMEM251 isoform.

As acknowledged by Reviewer 1, we previously demonstrated in our previous Nature Communications paper that the short isoform of TMEM251 is functionally equivalent to the long isoform. Specifically, we showed that the short isoform alone is sufficient to rescue the TMEM251 knockout phenotype. Repeating all experiments with the long

isoform would not only require substantial time and resources, but more importantly, would not yield additional scientific insight.

We would also like to respectfully point out a contradiction in Reviewer 1's attitude. While the reviewer criticizes Figures 1–4 as lacking novelty, despite our careful efforts to resolve ambiguities in the literature, they also recommend repeating numerous experiments using the long isoform, which we believe would not enhance the mechanistic understanding of TMEM251 function. Given the demonstrated functional redundancy between the isoforms, we feel that repeating the entire dataset with the long isoform is unnecessary.

The N-terminal region of TMEM251 contained sufficient information to retain a GFP-plasma membrane reporter in the Golgi. Since the N-terminal sequence met the criteria for binding of a COP1 adaptor, Golph3 its interaction with the N-terminus of TMEM251 was demonstrated in a GST pull down approach. However, direct binding of Golph3 in the streptavidin-binding assay to the N-terminus of TMEM251 could not be shown. Importantly, no experiments are presented to show the importance of Golph3 (e.g. in KO cells) for lysosomal sorting and GNPT stability and function. GFP reporter constructs like the one used in Figure 7 are well established tools to study the determinants of Golgi localization of type II membrane proteins. Indeed, the observations made here suggest that the first seven amino acids of TMEM251 could dictate a Golgi localization and potentially also an interaction with GOLPH3. However, this system is highly artificial for many reasons and the findings from these experiments should be taken with a grain of salt. TMEM251 is not a type II membrane protein and thus cannot be simply reduced to only its first 7 amino acids - in particular when the alanine scan data show that the 2-7A mutant still exhibits Golgi localization and instead mutation of other motifs impair Golgi localization. If this region is important for Golgi localization of TMEM251, shouldn't this mutant be mislocalized/lost from the Golgi apparatus? N-termini of type II Golgi membrane proteins are frequently rather short and Arg-rich – the capacity of TMEM251 aa 2-7 to direct the reporter to the Golgi may thus simply a coincidence. Also, these experiments need to be further substantiated by constructs with TMEM251's longer N-terminus and additional mutants as well as full-length TMEM251.

Thank you for the suggestions.

As requested by the reviewer, we examined the effect of GOLPH3 depletion in HeLa cells. Knockdown of GOLPH3 initially led to a reduction in lysosomal enzyme maturation (**Fig. S5E–F**). However, we observed a strong compensatory response over time, with GNPT $\alpha\beta$ protein levels increasing after 14 days (see Supporting Figure 1 below).

In addition, GOLPH3 depletion resulted in reduced colocalization between the Golgi and both TMEM251-10xGCN4 and endogenous GNPT $\alpha\beta$ (Fig. S5A–D). Notably, GOLPH3 appears to be essential for SKMEL30 cell viability, as its knockdown led to a marked decrease in cell survival.

Supporting Figure 1. Western blots show that GNPT $\alpha\beta$ levels initially decrease but subsequently increase in GOLPH3 knockdown HeLa cells.

Regarding Reviewer 1's additional concerns:

We focused on the N-terminal region of TMEM251 because of its critical functional importance, as highlighted by the R7W mutation (R45W in the long isoform), which is a known disease-causing mutation in humans. Our findings indicate that the N-terminal tail mediates interaction with GOLPH3, which is essential for TMEM251 retention in the Golgi.

To further support the significance of the TMEM251 N-terminus, we now provide additional evidence:

- The R7W mutation abolishes the interaction between TMEM251 and GOLPH3 *in vitro* (Supporting Fig. 2A).
- The R7W mutation disrupts Golgi localization of the 251^N-SITM-GFP chimera (Supporting Fig. 2B).
- The R7W mutant undergoes faster degradation than the wild-type protein (Supporting Fig. 2C).

These findings reinforce the functional relevance of the TMEM251 N-terminal tail in Golgi localization and stability.

Supporting Figure 2, R7W affects TMEM251 Golgi retention. (A) Pull-down assay showing the interaction between the 251^{N(WT)}-SITM-GFP and 251^{N(R7W)}-SITM-GFP chimeras and GST-GOLPH3. (B) Immunostaining images and quantification showing the localization of SI^N-SITM-GFP, 251^{N(WT)}-SITM-GFP, and 251^{N(R7W)}-SITM-GFP in WT HeLa cells. (C) A cycloheximide chase assay showing the degradation of WT and R7W TMEM251 over 2 hours. Scale bar: 10 μ m.

Regarding the observation that 2–7A and R7W mutants still localize to the Golgi in Fig. 6B (now Fig. S3A), two factors may contribute to this result:

- 1) Overexpression of TMEM251 mutants was required for localization experiments due to the low sensitivity of the available TMEM251 antibody (i.e., HPA048559, which is now discontinued). As shown in Fig. 8C (now Fig. 7E–F), overexpressed TMEM251 can bypass the requirement for retromer to restore Golgi localization. Additionally, despite being a disease-associated allele in humans, overexpressed R7W mutant can even complement TMEM251 knockout phenotypes (see Supporting Fig. 3 below), further indicating that overexpression may compensate for functional defects.

- 2) The C-terminal region of TMEM251 also contributes to its Golgi localization via retromer-dependent recycling (**Fig. 8A-C**), which may partially compensate for defects in the N-terminal domain.

Supporting Figure 3. Western blots showing that overexpression of either *TMEM251^{WT}* or *TMEM251^{R7W}* in *TMEM251* knockout cells can rescue CTSD maturation and restore degradation of lysosomal cargoes, including LC3-II and LAPT4A.

To investigate the motif within the N-terminal tail of TMEM251 responsible for Golgi localization, we generated individual alanine substitutions at residues 2–7 in the short N-terminal tail of the 251N-SITM-GFP construct. This analysis identified F⁴XXR⁷ as a critical motif required for GOLPH3-mediated Golgi localization (**Fig. 6D–F**).

To assess the functional significance of these residues, we performed rescue experiments in HEK293T *TMEM251* knockout cells using the TMEM251-10xGCN construct, either wild-type or carrying F4A, R5A, or R7A mutations. Under leaky expression from a Tet-on promoter to approximate endogenous levels, F4A and R7A mutants failed to rescue *TMEM251* knockout phenotypes, whereas R5A retained partial function (**Fig. S5A–C**).

Lastly, we evaluated the Golgi localization of these mutants in SKMEL30 cells. Upon moderate induction with 100 ng/μl doxycycline for 18 hours, both F4A and R7A mutants exhibited reduced Golgi localization compared to WT and R5A constructs (**Fig. S5D–E**).

VPS35 deletion (retromer member) did not affect TMEM251 expression levels but slightly reduced the level of GNPT as been reported earlier.

This statement is incorrect. Several members of our team have studied GNPT for decades, and to our knowledge, there are no published reports demonstrating that GNPT levels are reduced following *VPS35* knockout.

TMEM251 localization was apparently changed although the punctate Golgi-like structures remain unexplained. A fraction of TMEM251 could be found stabilized in lysosomes suggesting a retromer involvement in recycling of missorted TMEM251.

The imaging data depicting the mislocalization of TMEM251 and GNPTAB in retromer-deficient cells needs to be assessed more quantitatively.

As suggested by the reviewer, we quantified the mislocalization of TMEM251 and GNPT $\alpha\beta$ by calculating the Pearson correlation coefficient (Fig. 8C, now **Fig. 7D–E**). Additionally, we found that the punctate signal observed near the Golgi in GNPT $\alpha\beta$ -deficient cells is a staining artifact resulting from non-specific binding by the antibodies used in this study (**Fig. 1H**).

There still appears to be a substantial portion of TMEM251 in the Golgi and also the single-dot Golgi staining of GNPTAB is not as evident as in Fig. 4.

We did not claim that VPS35 deletion phenocopies TMEM251 knockout. However, we did observe a partial but significant reduction of TMEM251 from the Golgi, along with a corresponding decrease in GNPT $\alpha\beta$ signals in the Golgi (**Fig. 7D–E**).

In our proposed model (**Fig. 9**), COPI, GOLPH3, and the retromer complex all contribute to the proper Golgi localization of TMEM251 and GNPT $\alpha\beta$. Consistent with this model, deletion of VPS35 would be expected to result in a partial mislocalization of both proteins.

The authors also write that “the TMEM251-GNPT complex may exit the Golgi and mislocalize to endo-lysosomes” (line 412/413), but co-staining of TMEM251 and GNPTAB in retromer-deficient cells would be required to substantiate that indeed the TMEM251-GNPT complex is mislocalized (and not its two components independently).

Both antibodies were raised in rabbits, which prevents us from performing any co-immunostaining experiments. However, in purified lysosomes from WT cells, we observed a small but significant portion of both GNPT $\alpha\beta$ and TMEM251 (**Fig. 3D–E, 7C**), supporting our statement.

Also, the authors do see a destabilization/loss of GNPTAB under these conditions, but not of TMEM251. Is GNPTAB more prone to lysosomal degradation than TMEM251?

Yes, we agree with this interpretation. Uncleaved GNPT $\alpha\beta$ is a 143 kDa protein with the majority of its mass located in the lumen, making it susceptible to degradation by lysosomal proteases when mislocalized. In contrast, TMEM251 has only a short luminal

domain of approximately 40 amino acids, which likely limits its exposure to luminal proteolysis.

Quantification and statistical evaluation of multiple independent experiments and biological replicates are missing for most of the (immunoblot and microscopy-based) experiments.

We have quantified and performed statistical analysis as requested (see **Figs. 1, 2, 5, 6, 7, 8, S3, S4, and S5**).

Minor comments:

Different amounts of plasmid DNA (encoding in particular GNPTAB or TMEM251) were transfected transiently for numerous data panels to emphasize dose-dependent effects (e.g. of TMEM251 on GNPTAB stability). The figure legends should state that these absolute DNA amounts given in the figure panels apply to six-well culture dishes.

This information was stated in the Materials and Methods section of the manuscript.

Figure legends are rather cryptic in general and lack important information here and there (e.g. in Fig. 4E and 4G inhibitor concentrations should be provided in the figure panels or the figure legends)

We have added additional information to the figure legends, including concentrations and treatment time for the inhibitors used.

The authors write that all genome editing cell clones/lines (KO and KI) were validated/confirmed by Sanger sequencing. These data should be made available in the supplementary figures

We verified our cells using either Western blot or sequencing (**Fig. S6**). We also added the cell line information in the Materials and Methods section.

Fig 2B: From the context it can be assumed that GNPT $\alpha\beta$ and GNPT β were detected with an anti-HA antibody in these KI cells, but that needs to be stated in the legend

This was already indicated in the figure panel, but we also updated the legend to make it more straightforward.

Suppl. Fig. S2B – TMEM251 panel, between lane 3 and 4: Cutting/Splicing? Or why does

TMEM251 in GNPTAB KO cells exhibit a different running behavior compared to parental cells (cf to Fig. 1E)

The TMEM251 overexpression lysates were diluted to maintain them within the sensitivity range of the TMEM251 antibody. We have added an explanation in the figure legend (now **Fig. 1D**).

Contradictory statements: line 158 (section heading; “Tmem251 is required for stability of GNPT $\alpha\beta$) vs. line 189 to 191 (“In the absence of TMEM251, [...] the uncleaved GNPT $\alpha\beta$ level remains relatively unchanged”); in particular the latter statement does not appear to adequately reflect observations in e.g. Fig. 2C and Fig. S2B)

Thank you for pointing this out. We have revised the wording and quantified both the total GNPT $\alpha\beta$ and α subunit levels to provide a more numerical description (now **Fig. 1D-G**).

Line 244: WT cells should be TMEM251 WT cells (lanes 4 and 5 are GNPT KI cells, so no parental cells)

TMEM251 WT cells are a confusing nomenclature. We believe that leaving it as WT cells is sufficient to make the point. For further clarity, lane numbers are indicated in the text.

Lines 260 to 262 – reference to Fig. 4G missing in main text

Thank you! We included it in the text of this submission (now **Fig. 3G**).

Fig. 2E: Dot-like Golgi (-proximal) localization of GNPTAB upon TMEM251 deletion in SKMEL30 cells: This observation (in SKMEL30 cells) stands in contrast to observations by Richards et al. showing dot-like, but not really Golgi-proximal GNPTAB staining in TMEM251 KO 293FT cells. What is the nature of this unique dot-like structures? What happens if cells are treated with protease inhibitors, does GNPTAB then also accumulate in lysosomal structures (in light of the lysosomal enrichment data presented)?

We acknowledge the differences between our imaging data and that reported by Richards et al. These discrepancies may be due, in part, to differences in the epitopes recognized by the GNPT α antibodies used in the two studies. In our hands, we were unable to detect a lysosomal pool of GNPT $\alpha\beta$ in *TMEM251* knockout cells by immunostaining (**Fig. 1H**). To confirm GNPT $\alpha\beta$ mislocalization to the lysosome, we found it necessary to perform lysosome purification, which allowed us to detect this population biochemically (**Fig. 3D-E**).

As we noted in our study, BafA1 appears to stabilize GNPT $\alpha\beta$ NOT by preventing lysosomal degradation, but by inhibiting its exit from the Golgi (**Fig. 3G**). This mechanism was also independently observed by Pechincha et al. We are unsure how Richards et al. observed stabilization of GNPT $\alpha\beta$ at the lysosome following BafA1 treatment, as it is inconsistent with both our data and that of others.

Regarding the punctate "dot" signal, we concluded that it is an artifact generated by the GNPT $\alpha\beta$ antibody, as it persists even in *GNPTAB* knockout cells (**Fig. 1H**).

Could GNPTAB mutants (K4Q, S14Y) shown previously to impair binding of GNPTAB to COPI also impair binding to TMEM251? (previously looked at?)

This question was originally addressed in Fig. 7F (now removed for improved flow, see Previous Fig. 7F above). The K4Q and S14Y mutants clearly retain partial binding to TMEM251. This is supported by the observation that co-overexpression of TMEM251 with these mutants enhances phosphotransferase activity, suggesting a residual physical interaction. Although we have removed Fig. 7F to streamline the manuscript, the underlying data support the conclusion that these mutants are still capable of interacting with TMEM251 to some extent.

Line 384: "Deletion of VPS35 resulted in a significant reduction in endogenous GNPTab protein levels in both cell lines" – this statement is not overtly supported by the data presented. In SKMEL30 there is only a modest (at best) reduction and this would need to be properly quantified in light of this statement; the 293T data appear to be supporting this statement better, but quantification would be required here as well to exclude transfer/ECL issues

We have quantified the reduction of GNPT $\alpha\beta$ in the *VPS35* KO cell lines and made corresponding changes in the text (see **Fig. 7A-B**).

Fig. 5C – The E51N mutant generated to test the topology model appears to be not glycosylated quantitatively. Is this an issue with heavy overexpression or could this indicate the simultaneous presence of topologically different variants of TMEM251?

This may be due to overexpression of the E51N mutant. Excessive production of the protein could saturate the glycosylation machinery, resulting in a subset of the protein remaining unglycosylated.

When transiently transfected, low-level overexpression of GNPTAB is insufficient to compensate for the lack of TMEM251 and a drastic difference in Ptase activity is observed

between WT and TMEM251 KO cells (Fig. 1B, lane 5 vs 6). When stably expressing low-level GNPTAB-V5 variants (Fig. 3D), no notable difference in GNPTAB-V5 levels is evident between TMEM251 WT and TMEM251 KO cells (despite the latter clearly having a phenotype). How can that be explained?

TMEM251 regulates GNPT $\alpha\beta$ at two levels: **cleavage** and **enzymatic activity**. In Fig. 3D (now **Fig. 2E**), we use a modified GNPT $\alpha\beta$ construct (GNPT $\alpha\beta$ -Furin-3V5) engineered to be cleaved by Furin instead of S1P, as well as a second construct (GNPT $\alpha\beta$ - Δ 820–928-3V5) lacking the autoinhibitory domain. Both constructs bypass the requirement for S1P-mediated cleavage and, as expected, their expression levels are largely unaffected in *TMEM251* knockout cells. However, despite successful cleavage, these GNPT $\alpha\beta$ variants remain non-functional specifically in the absence of TMEM251. This suggests that TMEM251 exerts an additional layer of regulation by directly modulating GNPT $\alpha\beta$'s phosphorylation enzyme activity, independent of its role in cleavage.

Reviewer 2

In this research, the authors conducted a thorough analysis of the role of TMEM251/TMEM251 protein in the processing, stability, and localization of GlcNAc-1-phosphotransferase 35 (GNPT), a key enzyme in the M6P pathway. TMEM251 is essential for human health and is a potential drug target for treating cancers and viral infections. While two studies in 2022 linked TMEM251 to the M6P pathway, its specific role remained unclear. The authors used gene editing, microscopy, mutagenesis analysis, and in vitro methods to show that TMEM251 is crucial for the processing, activity, and maintenance of GNPT in the Golgi apparatus. They also conducted biochemical analysis and alanine mutagenesis scanning to determine the membrane topology of TMEM251 and identified critical residues in its cytosolic region that affect protein function and localization. The findings confirm the importance of TMEM251 and GNPT in the M6P pathway at the physiological level and support the hypothesis that GNPT's steady localization in the Golgi is maintained through vesicular recycling mediated by TMEM251, COPI, Golph3, and retromer. This work is suitable for publication in this journal, although several critical questions remain unanswered. Addressing these questions could further elevate the significance of the findings: What is the stoichiometry of the GNPT-TMEM251 complex? Is this a stable complex, or is it regulated temporally and spatially? Are other Golgi enzymes regulated by TMEM251? What is the molecular mechanism by which TMEM251 influences GNPT processing?

Specific comments and suggestions for improvement:

The study predominantly utilized knockout cells and overexpression conditions. This might lead to secondary defects, compensatory responses, and other artifacts. Implementing acute protein downregulation techniques and focusing on endogenous proteins—perhaps tagged using gene-editing methods—would significantly enhance the robustness of this study.

Thank you for these thoughtful suggestions. In our experience, CRISPR-mediated gene knockout generally yields cleaner and more consistent results than protein knockdown approaches. To ensure reproducibility, we routinely analyze two or more single-cell clones and test across multiple cell lines. Additionally, rescue experiments are typically performed to confirm the specificity of the observed phenotypes.

In the original Fig. 1 (now removed), we used GNPT $\alpha\beta$ overexpression to demonstrate that the requirement for TMEM251 could be bypassed, but only when GNPT $\alpha\beta$ is expressed at levels significantly above physiological conditions. This supported the idea that TMEM251 is essential for GNPT $\alpha\beta$ function under endogenous expression levels. While we have since removed and redistributed the data from Fig. 1 to improve manuscript flow, the conclusions remain supported.

To focus on endogenous protein behavior, we used SKMEL30 cells, which express higher natural levels of both GNPT $\alpha\beta$ and TMEM251. In HEK293T cells, we employed a GNPT $\alpha\beta$ -3xHA knock-in line that allows us to monitor endogenous levels of both full-length GNPT $\alpha\beta$ and its β subunit via HA antibody.

A key limitation of this study is the lack of a reliable antibody for detecting endogenous TMEM251. Unfortunately, during the course of manuscript revision, the new batch of the TMEM251 antibody (HPA048559 from Sigma) lost sensitivity and could no longer detect the protein. To address this, we developed a Tet-on TMEM251-10xGCN expression system, and used its leaky expression to approximate endogenous TMEM251 levels for our experiments.

In Figure 1E, it is crucial to display a complete GNPTAB blot to confirm the effect of TMEM251 knockout on the processing of the endogenous enzyme in HEK cells. Additionally, clarification on the role and relevance of HRD1 in this context would be helpful.

Fig. 1E is now presented as **Fig. S1A**. In this experiment, HRD1 was used as a loading control for membrane isolation, and GNPT $\alpha\beta$ protein levels were normalized to HRD1, as indicated in the figure legend.

Due to the low endogenous expression of GNPT $\alpha\beta$ in HEK293T cells, detection by immunoblotting is challenging. To improve signal specificity, we cut the membrane at

~100 kDa to concentrate antibody binding on the appropriate region of the blot (now **Fig. S1A**).

To directly address the reviewer's concern, we used a GNPT $\alpha\beta$ -3xHA knock-in HEK293T cell line (Fig. 2B, now **Fig. 1A**), which allowed us to more clearly monitor GNPT $\alpha\beta$ processing in the context of *TMEM251* knockout.

TMEM251 KO cells showed mislocalization of GNPT, forming a "single-dot-like structure" at the Golgi rather than mislocalizing to lysosomes. It's crucial to investigate the nature of this structure. Immuno-electron microscopy (immune-EM) could be a suitable further exploration method.

We now have evidence that the dot-like structure observed near the Golgi is likely due to non-specific antibody binding, as similar puncta are still present in *GNPTAB* knockout cells (**Fig. 1H**).

The observation that in TMEM251 KO cells treated with BafA1, stabilized GNPT remains in the Golgi instead of moving to the lysosome is significant. The authors suggest that "BafA1 stabilizes GNPT by impeding its Golgi exit rather than inhibiting lysosomal degradation." This novel finding warrants further investigation to understand the underlying mechanisms.

This is indeed an interesting observation that warrants further investigation. Importantly, Pechincha et al. also reported that BafA1 can stabilize GNPT $\alpha\beta$ at the Golgi (Pechincha et al., 2022b). However, a detailed exploration of this mechanism is beyond the scope of the current study.

The effects of GOLPH3 depletion (in the context of GOLPH3/GOLPH3L double KO) on the M6P pathway and the localization of TMEM251 and GNPT should be explored. This could provide insights into the regulatory mechanisms involving GOLPH3 in this pathway.

Thank you for this excellent suggestion. We have now included data examining the effects of *GOLPH3* deletion or downregulation on the M6P pathway:

GOLPH3 knockdown in HeLa cells: We knocked down *GOLPH3* in HeLa cells and observed a decrease in lysosomal enzyme maturation (**Fig. S4E–F**). However, over time, we noted a compensatory response in which GNPT $\alpha\beta$ expression was elevated compared to wild-type (See supporting Figure 1 above). A similar compensatory effect was also observed in U2OS cells (data not shown). These findings suggest that such compensatory effects are not limited to CRISPR-mediated knockouts but can also occur in knockdown experiments.

GOLPH3 knockdown in SKMEL30 cells: We generated a *GOLPH3* knockdown SKMEL30 cell line to assess endogenous GNPT $\alpha\beta$ localization. *GOLPH3* depletion resulted in reduced GNPT $\alpha\beta$ localization to the Golgi (**Fig. S5C–D**). Notably, *GOLPH3* appears to be essential in SKMEL30 cells (i.e., knockdown severely impaired cell viability, and complete knockout was lethal).

As there is currently no reliable antibody for immunostaining endogenous TMEM251, we tagged TMEM251 with a 10xGCN4 epitope and expressed it under the leaky expression of a Tet-on promoter. In U2OS cells, we observed a clear reduction in Golgi localization of TMEM251-10xGCN following *GOLPH3/3L* knockout (**Fig. S5A–B**), further supporting the role of *GOLPH3* in TMEM251 retention.

Reviewer 3

The study by Yang, Doray and colleagues analyzes the role of TMEM251 in the activity of the GNPT enzyme which constitutes the fulcrum of the M6P pathway, and which is essential for the function of lysosomal enzymes. The most innovative part of the work concerns the role of TMEM251 in the GNPT traffic mechanisms. This is a work of considerable interest which, however, in its current form presents some important gaps of both an interpretative and experimental nature. The work is very complex and, as such, intrinsically difficult to analyze, but it is also written in a poorly organized way so that the logical connections between the parts are even more difficult to follow

Below we will try to analyze these weaknesses and where possible indicate how to overcome them.

Major points:

The authors describe the effects of TMEM251 on the stability of the GNPT enzyme and give evidence that TMEM251 is essential for the activating GNPT cleavage and is also essential for the stability of the cleaved portion of GNPT, in the sense that the levels of this protein decrease in the absence of TMEM251. In this regard the authors note that GNPT is intrinsically unstable due to the composition of its transmembrane domains, but in the following experiments the authors demonstrate that in the absence of TMEM251 the uncleaved GNPT is degraded in lysosomes. This is probably the reason for the instability of GNTP. The flow here is difficult to understand and the authors should logically connect the two data sets.

Thank you for your comments. We have rewritten a large portion of our manuscript and removed some of the figures to increase the logic flow. In the absence of TMEM251,

GNPT $\alpha\beta$ is indeed unstable and trafficked to the lysosome for degradation. Due to the lack of GNPT $\alpha\beta$ stability in cells lacking TMEM251, GNPT $\alpha\beta$ cleavage and subsequent activity is reduced leading to the observed lysosomal defects.

The authors show that TMEM251 is necessary for GNPT activity even after cleavage has occurred ie, after the activation step has been carried out. This is attributed to the physical interaction of TMEM251 with GNPT, which would form a stable complex which would have the effect of stabilizing the active conformation of the enzyme. In the next part of the work, however, the authors demonstrate that TMEM251 is essential to prevent lysosomal degradation of both the cleaved and uncleaved enzyme. From what we understand, it is therefore probable that the property of TMEM251 to maintain the activity of the cleaved enzyme is simply due to the fact that it localizes GNPT at the Golgi preventing its lysosomal degradation. This possibility does not seem to be grasped by the authors. It is an ambiguity that the authors must resolve either with textual changes and/or with new experiments.

Thank you for pointing this out. As mentioned above, we have revised the text to improve clarity. We propose that TMEM251 functions to retain both the uncleaved and cleaved forms of GNPT $\alpha\beta$ in the Golgi, thereby supporting proper S1P-mediated cleavage and ensuring the enzymatic activity of the cleaved form. In the absence of TMEM251, uncleaved GNPT $\alpha\beta$ may prematurely exit the Golgi before it can be processed by S1P, while the cleaved form is similarly mislocalized before it can perform its function.

The authors perform an elegant series of experiments that leads them to the conclusion that TMEM251 interacts with Golph3 in the trans Golgi to maintain the localization of the TMEM251/GNPT complex in the cis Golgi and for the prevention of lysosomal degradation of GNPT.

A perplexing point here is the observation that mutants in the cytosolic tail of TMEM251 that abolish the binding between the tail and Golph3 do not change the localization of TMEM251 in the Golgi, in other words, that the interaction between TMEM251 and Golph3 is not essential for the retention of TMEM251 and GNPT in the Golgi. The authors suggest that there may be additional regions beyond the cytosolic tail that maintain TMEM251 in the Golgi. This is possible, but the finding raises concerns that should be addressed with textual changes or new experiments.

To address this concern, we have revised the text for greater clarity and conducted additional experiments (see below).

To investigate the role of GOLPH3 in maintaining TMEM251 at the Golgi, we used a Tet-on TMEM251-10xGCN construct in U2OS cells. Under moderate induction, we observed that Golgi localization of TMEM251-10xGCN was reduced in cells lacking GOLPH3/3L compared to wild-type controls (**Fig. S5A–B**). Furthermore, we provide compelling evidence that GOLPH3 mediates this retention through the N-terminal F⁴XXR⁷ motif of TMEM251: mutation of these residues leads to partial loss of TMEM251 from the Golgi (**Fig. 6** and **Fig. S4**).

In addition, depleting GOLPH3 in SKMEL30 cells resulted in a loss of endogenous GNPT $\alpha\beta$ from the Golgi (**Fig. S5C–D**), further supporting GOLPH3's role in stabilizing the GNPT $\alpha\beta$ –TMEM251 complex at the Golgi.

Based on our proposed model (**Fig. 9**), TMEM251 and GNPT $\alpha\beta$ are retained at the Golgi through the combined action of COPI, GOLPH3, and the retromer complex. Therefore, it is not unexpected that only a portion of TMEM251 is lost from the Golgi upon GOLPH3 depletion, as other retention mechanisms remain partially active.

The authors show evidence that the retention of GNPT in the Golgi requires both the binding of GNPT itself to COP1 via the motif previously identified by Doray and Kornfeld, and the binding between TMEM251 and Golph3, as shown in this study. Then, in the last paragraph, the authors propose the existence of a further mechanism for the retention of GNPT in the Golgi which would be based on the activity of the retromer, the complex that transports proteins from the endo-lysosomal system to the Golgi, which thus be responsible for maintaining the localization and functionality of GNPT/ TMEM251. This last conclusion is consistent with the data, which however are simply based on depletion and overexpression experiments of a protein crucial for the function of the retromer, VPS35.

Based on these collective evidence, the authors suggest that a triple recycling/retention mechanism is necessary for the maintenance of GNPT/ TMEM251 in the Golgi and present a model in figure 9.

The model shows a single retrotransport step from the trans to the cis Golgi. This is difficult to understand based on current knowledge and also inconsistent with the data shown in the manuscript and with the localization of the adaptor GOLPH3 within Golgi. The overall model could be modified by proposing a mechanisms of Golgi retention in three steps: first step of retrotransport from Golgi trans to medial cisternae due to the binding between TMEM251 and Golph3 which would then be followed by a second retrograde step dependent on the binding between the cytosolic tail of GNPT and COP1. The third step, mediated by the retromer, could recycle the fraction of GNPT / TMEM251 complex that escaped from the first two recycling mechanisms, from the endo-lysosomes to the trans Golgi. Here, GOLPH3 first and COPI then could bring the complex back to

the cis Golgi. Organized in this way the model could be coherent with the majority of the data and highlight the novelty of the paper.

Thank you for the suggestion. We have revised the model to better reflect the current data.

Overall, we believe that the information brought by this study potentially represents a notable progress in an area of research which is currently under strong development and is of great importance, namely, the mechanisms of selective retrotransport of Golgi enzymes by specialized adaptors, with regulatory effects on glycosylation. However, the authors need to address the points we have raised with textual changes and new experiments.

Thank you for your positive comments. We have addressed the minor concerns as described below.

Minor points:

1) One hypothesis put forward by the authors suggests that TMEM251 may serve as a bridging factor, linking GNPT with the S1P protease necessary for GNPT activation. It would be important to investigate the localization and levels of S1P in TMEM251 knockout cells.

S1P cleaves a range of substrates in the Golgi, including SREBP1, SREBP2, ATF6, and GNPT $\alpha\beta$. Therefore, its localization and stability in the Golgi are likely independent of TMEM251. To confirm this, we performed immunostaining using an S1P-3xHA construct to assess S1P localization in TMEM251 knockout cells (see **Supporting Fig. 5** below). As expected, S1P remained localized to the Golgi in the absence of TMEM251.

Supporting Figure 5. Immunostaining showing the localization of S1P-3xHA and Golgi marker GRASP55 in WT and *TMEM251* KO cells. Scale bar: 10 μ m.

2) While *GOLPH3* is suggested as a mechanism for retaining the *TMEM251*-GNPT complex at the Golgi, its involvement remains unproven in *GOLPH3* knockdown (KD) or knockout (KO) cells. The authors establish that the N-terminal tail of *TMEM251* contains sufficient information for *GOLPH3*-dependent Golgi localization. They should define the motif. However, this is the sole evidence implicating *GOLPH3* in *TMEM251* retention at the Golgi, and using flow cytometry could quantitatively clarify this aspect.

We thank the reviewer for these thoughtful suggestions.

As requested, we identified the N-terminal motif required for *GOLPH3*-dependent Golgi localization of *TMEM251*, specifically the F⁴XXR⁷ sequence. Using the *TMEM251*^N-SI-GFP chimera, we introduced point mutations in the N-terminal tail to determine which residues are essential for Golgi retention (**Fig. 6D–F**). Since the imaging results clearly indicated the critical residues, we did not pursue flow cytometry for further analysis.

To assess the functional importance of these residues, we introduced F4A, R5A, and R7A mutations into our Tet-on *TMEM251*-10xGCN system and evaluated their ability to rescue mature cathepsin D (mCTSD) levels (**Fig. S4A–C**) and restore Golgi localization (**Fig. S4D–E**).

Furthermore, we provided additional evidence for *GOLPH3*-dependent recycling of *TMEM251* by analyzing *TMEM251*-10xGCN localization in U2OS *GOLPH3*/3L double knockout (DKO) cells and endogenous GNPT $\alpha\beta$ localization in SKMEL30 *GOLPH3* knockdown cells (**Fig. S5A–D**). In both cases, Golgi localization of *TMEM251* and

GNPT $\alpha\beta$ was lost upon GOLPH3 depletion. Additionally, *GOLPH3* knockdown in HeLa cells led to reduced lysosomal enzyme maturation (**Fig. S5E–F**). However, over time, we observed a compensatory response, with GNPT $\alpha\beta$ protein levels increasing (see Supporting Fig. 1).

3) Despite an interaction between the N-terminal tail and GOLPH3, the TMEM251 2-7A mutant, expected to lack GOLPH3 interaction, still localizes to the Golgi. The authors propose that additional regions beyond the N-terminal tail are responsible for retaining TMEM251 in the Golgi. However, the TMEM251 2-7A mutant fails to restore GNPT levels in TMEM251 KO cells despite its Golgi localization. The authors should discuss this discrepancy or address it with new experiments; perhaps the TMEM251 2-7A mutant is present in the Golgi but not in its precise subcellular location.

In **Fig. 5A**, GNPT $\alpha\beta$ levels rescued by the various TMEM251 mutants were assessed under Tet-on leaky expression conditions, which approximate endogenous TMEM251 levels. In contrast, for imaging experiments (**Fig. S3A**), the TMEM251 mutants were overexpressed, which may explain the observed Golgi localization of the 2–7A mutant despite its impaired function.

To further investigate this, we generated individual point mutations—F4A, R5A, and R7A—in TMEM251 within the Tet-on expression system (**Fig. S4A**). Under leaky expression, both the F4A and R7A mutants failed to rescue lysosomal enzyme processing (**Fig. S4B–C**). Additionally, when moderately induced with doxycycline, these mutants displayed reduced Golgi colocalization compared to wild-type and R5A (**Fig. S4D–E**). These results demonstrate that disrupting key residues involved in GOLPH3 interaction compromises both the localization and functional activity of TMEM251.

4) The authors suggest that there is a fraction of GNPT /TMEM251 which undergoes recycling back to the Golgi through the retromer machinery. They support this with findings that VPS35 KO results in TMEM251 and GNPT appearing in lysosomal puncta, where they degrade due to defective retromer function. However, a discrepancy arises in VPS35 KO, where TMEM251 remains at lysosomes but is not degraded, while GNPT is degraded. The authors should discuss this discrepancy or address it with new experiments. This might be due to the loss of proteases that degrade TMEM251 but not those that target GNPT.

Thank you. We have addressed this point in the revised text. Full-length GNPT $\alpha\beta$ is a 143 kDa protein with the majority of its mass located in the lumen, rendering it highly susceptible to degradation by lysosomal proteases if mislocalized. In contrast, TMEM251 contains only a short luminal segment of approximately 40 amino acids, making it more resistant to lysosomal degradation.

Consistent with this, we observed partial degradation of TMEM251 in the lysosomal fraction of *VPS35* knockout cells, with a degradation product detected between 10 and 15 kDa (**Fig. 7C**).

5) The authors hypothesize that the TMEM251 cytosolic C-terminal tail, especially residues 93-96, might interact with the retromer for recycling back from endo-lysosomes to the Golgi, although co-immunoprecipitation failed to detect such interaction. Investigating the localization of the 93-96A mutant in lysosomes could further corroborate this hypothesis.

As suggested, we examined the localization of the TMEM251 93–96A mutant and found that it is retained in the ER, suggesting a potential protein folding defect (**Fig. S3A–C**).

To further support our hypothesis that the retromer interacts with the C-terminus of TMEM251, we utilized a C-terminal chimera expressed in both wild-type and *VPS35* knockout SKMEL30 cells. Fusion of the TMEM251 C-terminal tail to the N-terminal portion of IL-2R α was sufficient to confer Golgi localization in wild-type cells (**Fig. 8A–C**). This localization was significantly reduced in *VPS35* knockout cells, indicating that the retromer is required to retain the chimera at the Golgi.

To narrow down the putative retromer binding site, we performed an extensive alanine scan of individual residues in the C-terminal tail of TMEM251-10xGCN. Mutations at V97A, Y99A, V108A, I109A, C110A, and especially their combinations resulted in decreased Golgi signal and increased punctate structures (**Fig. 8D–E**), consistent with impaired Golgi retention and suggesting these residues are important for retromer-mediated recycling.

Although we have not been able to demonstrate a direct interaction between *VPS35* and TMEM251—potentially due to the transient nature of the interaction or the requirement of an adaptor protein—our findings strongly support a functional role for the retromer in recognizing this C-terminal motif.

6) A topological issue arises from proposing the TMEM251-GNPT complex being recycled by GOLPH3 from trans Golgi to cis in Fig9. GOLPH3 is indeed localized at the trans-Golgi network (TGN). The authors could introduce the 3 steps model in the figure 9, which is more coherent with data they are showing.

Thank you. We adjusted the model accordingly.

Reviewer 4

Thank you for taking the time to provide thoughtful and valuable suggestions to improve our manuscript.

References:

- Baker, M. 2016. 1,500 scientists lift the lid on reproducibility. *Nature*. 533:452-454.
- Doray, B., D. Henn, X. Yang, M. Li, and P. Dickson. 2025. Protocol to measure endogenous GlcNAC-1-phosphotransferase activity in SK-MEL-30 cells. *STAR Protocols*.
- Errington, T.M., A. Denis, N. Perfito, E. Iorns, and B.A. Nosek. 2021. Challenges for assessing replicability in preclinical cancer biology. *Elife*. 10.
- Liu, L., B. Doray, and S. Kornfeld. 2018. Recycling of Golgi glycosyltransferases requires direct binding to coatamer. *Proceedings of the National Academy of Sciences*. 115:8984-8989.
- Pechincha, C., S. Groessl, R. Kalis, M. de Almeida, A. Zanotti, M. Wittmann, M. Schneider, R.P. de Campos, S. Rieser, and M. Brandstetter. 2022a. Lysosomal enzyme trafficking factor LYSET enables nutritional usage of extracellular proteins. *Science*. 378:eabn5637.
- Pechincha, C., S. Groessl, R. Kalis, M. de Almeida, A. Zanotti, M. Wittmann, M. Schneider, R.P. de Campos, S. Rieser, M. Brandstetter, A. Schleiffer, K. Muller-Decker, D. Helm, S. Jabs, D. Haselbach, M.K. Lemberg, J. Zuber, and W. Palm. 2022b. Lysosomal enzyme trafficking factor LYSET enables nutritional usage of extracellular proteins. *Science*. 378:eabn5637.
- Richards, C.M., S. Jabs, W. Qiao, L.D. Varanese, M. Schweizer, P.R. Mosen, N.M. Riley, M. Klüssendorf, J.R. Zengel, and R.A. Flynn. 2022. The human disease gene LYSET is essential for lysosomal enzyme transport and viral infection. *Science*. 378:eabn5648.
- Welch, L.G., S.-Y. Peak-Chew, F. Begum, T.J. Stevens, and S. Munro. 2021. GOLPH3 and GOLPH3L are broad-spectrum COPI adaptors for sorting into intra-Golgi transport vesicles. *Journal of Cell Biology*. 220:e202106115.
- Zhang, W., X. Yang, Y. Li, L. Yu, B. Zhang, J. Zhang, W.J. Cho, V. Venkatarangan, L. Chen, and B.B. Burugula. 2022. GCAF (TMEM251) regulates lysosome biogenesis by activating the mannose-6-phosphate pathway. *Nature Communications*. 13:5351.

REVIEWER COMMENTS

Reviewer #1 (Remarks to the Author):

Reviewer Comments (Reviewer#1):

Thanks for revising the Yang et al. manuscript and trying to address my and the comments of the other reviewers. As outlined in more detail below and despite the attempts to argue that there is an apparent controversy in literature which has to be solved and that findings need to be repeated, I still remain reserved related to the conceptual advance and novelty presented in this manuscript. A large part of the previous manuscript, yet still a substantial part of the revised manuscript (Figs. 1 to 3) is dedicated to resolving the finer details of the LYSET-GNPT interplay. I have previously voiced that this particular part (on its own) does not represent a significant conceptual advance and do have to maintain this view. As suggested in my previous review I find the manuscript much more suited for publication in a specialized biochemical journal adding some extra information to the field. However, in a journal like NCOMM I would have expected a significant body of novel and unexplored findings which is -according to my opinion- not the case in this study.

Regarding the “novelty” of our manuscript, we would like to point out that our manuscript not only covers the finer details of LYSET-GNPT interplay, as acknowledged by the reviewer, but also:

- **Confirm the topology of LYSET**
- **Identify the critical residues of LYSET**
- **Confirm GOLPH3’s role in maintaining LYSET in the Golgi,**
- **Identified the recognition motif of GOLPH3 for LYSET,**
- **Determined that the retromer plays a role in maintaining LYSET in the Golgi**
- **Show evidence that the retromer binds to the C-terminal of LYSET.**

We believe that all these discoveries contribute to the novelty of the paper.

The authors have now, though, included a paragraph (lines 148ff) that clarifies the situation a bit better and differentiates between their own previously reported data/model (“processing model”) and the data presented by two other independent papers (“trafficking model”). Indeed, resolving this controversy is of interest to the field of researchers interested in LYSET/GNPT biology, but in this particular case it also needs to be noted that the newly presented data rather argue in favour of the trafficking model proposed by others previously and do not strongly support the initial model proposed by the authors (Zhang et al., 2022, Nat Commun.). Indeed, the authors should more strongly voice to what extent their own data now support the model reported before by the other groups, e.g. in lines 193-195, line 217 and line 227/228. While the authors raise a point that some discrepancies exist among the three studies on the function of LYSET, it would be helpful to clarify that the primary point of contention lies with their own publication (Zhang et al., 2022, Nat Commun.), in which the role of LYSET was interpreted as the sole activator of MBTPS1-mediated cleavage of GNPTAB—an interpretation that remains under debate. The perceived “controversy”

between the two back-to-back Science papers is relatively minor and may largely stem from methodological differences, such as the use of different cell lines, endogenous expression versus overexpression systems, and antibodies that detect either the alpha or beta subunits of GNPTAB. Other variables, such as subunit stability and processing of the precursor protein, may also contribute. Notably, the immunofluorescence data in Richards et al. (Fig. S14) are based on endogenous protein detection using an antibody targeting the alpha subunit. Following 24-hour Bafilomycin A1 treatment, a modest increase in Golgi-localized signal was observed. The lysosomal staining might reflect a pool of slowly degraded alpha subunit that had accumulated before Golgi exit was inhibited. Similarly, in Figure 3E, the 19-hour S1P inhibitor treatment showed only minor effects, likely due to the high stability of the endogenous protein. Since Pechincha et al. studied only newly synthesized, overexpressed protein, the perceived discrepancies may be less significant than suggested. As the authors themselves acknowledge, the precursor protein may exhibit different stability in the presence of lysosomal proteases. To conclusively demonstrate that the precursor is primarily degraded in lysosomes, experiments using lysosomal protease inhibitors would be necessary.

We are unsure how to answer this criticism because we did not acknowledge that “the precursor protein may exhibit different stability in the presence of lysosomal proteases.” Please indicate the lines where we made such a statement. Instead, we found that LYSET is resistant to lysosomal degradation.

Finally, as far as I can judge the claim that Pechincha et al. and Richards et al. exclusively showed lysosomal degradation of the cleaved GNPTAB subunits is not accurate. While Richards et al. predominantly detect the endogenous alpha subunit at steady state, they do not assert that cleavage is a prerequisite for degradation. The slightly higher molecular weight species observed in LYSET knockout cells may also include the precursor.

We are unsure about this criticism as we did not claim that “Pechincha et al. and Richards et al. exclusively showed lysosomal degradation of the cleaved GNPTAB subunits”. We only stated “An alternative model proposed that the processing of GNPT is unaffected following LYSET knockout; instead, the LYSET-GNPT interaction is necessary for retaining cleaved GNPT at the Golgi.” (Lines 86-88)

In Pechincha et al., it was stated, “Radioactive pulse-chase experiments demonstrated that LYSET was dispensable for synthesis and proteolytic processing of ectopically expressed GNPTAB α/β precursor (fig. S10A).

Moreover, Pechincha et al. provide clear evidence of an increase in cleavage fragments upon treatment with lysosomal protease inhibitors, strongly indicating that the precursor itself undergoes lysosomal degradation.

We thank the reviewer for pointing this out. This result strongly supports our conclusion that LYSET facilitates S1P-mediated GNPTAB cleavage. All three papers

confirmed that the mRNA level of GNPTAB is not affected in LYSET KO cells. With the degradation of the precursor, one should expect to see reduced processing.

(Introduction): I agree that LYSET seems to have multiple functions in regulating GNPTAB localization and therefore function. However, the authors neglect here that a stabilizing role of LYSET for the TMD1 of GNPTAB has been clearly shown. The authors themselves have (re-) demonstrated this in their recent MBoC publication (Doray et al. 2025). Showing the interaction of LYSET with the Golgi-adaptor GOLPH3, however, again lacks novelty, since this was shown by Brauer et al. before (EMBO J 2024) which the authors only briefly mention in their discussion.

In our initial submission, we already included results showing the interaction between LYSET and GOLPH3. We have confirmed with the editorial office that we were under the scoop protection policy from NC. Moreover, we identified the recognition site as the FXXR motif using two different methods.

Figure 1:

The new Figure 1 includes very few novel information but is primarily a reconfirmation of data published by Richards et al, Pechincha et al. and Zhang et al. The whole figure including panel H serving as a control for their newly generated antibody would be better suited for the supplement. In Figure S1, again, the authors present no novel data but replicate published results that never presented any controversy and could be explained both by LYSET-mediated activation, retention or stabilization of GNPTAB.

We have addressed this comment in the first round of rebuttal. The reviewer has very detailed knowledge of the field; however, most readers do not have that advantage. It is necessary to familiarize the readers with the importance of LYSET in the M6P pathway. Moreover, the paper consists of a total of nine main figures and six supplementary figures, the majority of which are novel and contribute to the field's understanding of the M6P pathway.

Figure 2:

Panel A: Quantification is missing. Since the overall expression of GNPTAB is reduced in LYSET KO, it is hard to judge whether there is a slight delay in MBTPS1- mediated processing.

We thank the reviewer for this suggestion. We included the quantification for Fig. 2A (See Fig. 2B) to show that the overall processing rate of GNPTAB is reduced in LYSET KO cells.

Fig2B: A western blot should be added, otherwise it is impossible to distinguish between altered expression and altered activity.

We included the Western Blot for Fig. 2B (now Fig. 2C) as the reviewer suggested, thank you.

If the activity data is from three independent biological replicates, why is the standard deviation not shown? Single data points including error bars should be presented.

We are unsure about this criticism because Figure 2B (Now Fig 2C) already shows the standard deviation with the individual data points. In addition, we are deeply saddened to notify this reviewer that our beloved Stuart suddenly passed away on August 17. It will be impossible to ask his group to perform more experiments at this point.

Fig. 2E The authors have to show the absence of LYSET and the purportedly mild overexpression. The conclusion of the panel is potentially important since it would be the only piece of evidence that the phosphotransferase activity itself (at least of the mutants) might be affected by LYSET expression, so all the controls have to be shown. If LYSET is overexpressed anyway, there is no reason not to use a tagged version.

The reviewer misunderstood Figure 2E (Now Fig. 2F). We did not overexpress LYSET. These cell lines were either WT or LYSET KO cell lines. Instead, we mildly OE GNPTAB-820-928deletion-3V5 or GNPTAB-Furin-3V5, the expression levels of which are shown. This experiment is to investigate if LYSET is required for GNPTAB activity after its cleavage.

At the same time, LYSET detection would be critical to e.g. take expression levels of LYSET into consideration for evaluating the various mutants generated (Fig. 5). This control is essentially missing. Considering the newly included findings, it would also be important to check levels of endogenous LYSET in GOLPH3/GOLPH3L-deficient cells

As stated above, we are evaluating the function of different GNPTAB mutants mostly in LYSET KO cells background, and no overexpressed LYSET was used in this assay. So, the expression of endogenous LYSET is not critical for drawing the conclusions. We agree that having a LYSET blot will serve as a nice control and give a complete picture of the whole assay. Unfortunately, as we stated in the last round of revision, the new batch of TMEM251 antibody (HPA048550 from Sigma) lost sensitivity, and we can no longer access the LYSET western blot.

Figure 3:

Fig. 3D Is the whole cell lysate shown or a WCL depleted from lysosomes? How do the authors explain the low LAMP2 signal and the many (unspecific?) bands?

WCL stands for whole cell lysate, which is the starting material used for

lysosome purification. There might be two reasons for the many bands in LAMP2 blots: 1) LAMP2 is a heavily glycosylated protein. It always migrates as a smear or several bands. 2) It is not uncommon for antibodies to recognize additional, non-specific bands. In the “LYS” fraction, however, the two non-specific bands, were removed by lysosome purification, confirming lysosome purification can remove non-specific bands from the WCL.

Figure 4:

Fig. 4E The labelling of the panels is misleading, please clearly indicate that all the proteinase K-treated conditions are also treated with Digitonin for 3 minutes. Scale bars are missing.

We relabeled the panels of Figure 4E so that it is easier to understand and added the scale bar.

If 4F is really the quantification of the images shown in E, the timepoints of addition. of digitonin and proteinase K are incorrectly labelled.

We carefully re-checked the time points in panel F. Digitonin was added at minute 2 and incubated for three minutes (2–5 min), followed by adding proteinase K at minute 5. This timing is consistent with that shown in panel E. The main difference between Figure 4E and 4F, is that the images with just the KHM buffer washing were not included in Figure 4E. However, to make it easier to understand, we slightly changed the labeling on Figure 4F. We also revised the figure legends to provide more detailed information.

Figure 5:

Fig.5A/B How do the authors explain that some mutations in LYSET have completely different functions in 293T cells than in SK-MEL30 cells (e.g. 67-71A, which shows abnormal CTSD processing/localization, but WT GNPTAB levels)? For 87-91A or 93-96A this is less pronounced, but there is way more GNPTAB than in 82-86A but similar CTSD signals.

We have addressed this point in our previously revised manuscript (Line 283, now lines 287-289). These results suggest that mutations in this region do not disrupt LYSET’s role in anchoring GNPTAB, but may instead impair other functions, such as GNPTAB enzymatic activity.

Fig. 5C: other topology predictors show a slightly different topology with R7 directly

adjacent to the membrane (as in Fig. 8F), please indicate which tool was used and harmonize the different depictions.

Thank you for pointing this out. We corrected the topology models in Fig 5C and 8F to be consistent with what was predicted by Topcon. We also mentioned this in the Figure legend.

R7W does not decrease LYSET levels (confirming the data in Richards et al. 2022), but decreases GNPTAB expression (as also shown in Richards et al.) The data in Fig.6F suggests that R7 is important for GOLPH3 binding so the mutant would be expected to destabilize LYSET levels. How do the authors explain this discrepancy?

Although in SKMEL30 cells (Fig. 5B), the overexpressed R7W mutant does not seem to decrease the protein level, we did observe that R7W has a short half-life compared to WT LYSET, which we have previously shown in Supporting Figure 2C of our rebuttal letter. However, we will include the data again. Please see Supporting Figure 1C.

Supporting Figure 1, R7W affects TMEM251 Golgi retention. (A) Pull-down assay showing the interaction between the 251^{N(WT)}-SITM-GFP and 251^{N(R7W)}-SITM-GFP chimeras and GST-

GOLPH3. **(B)** Immunostaining images and quantification showing the localization of SI^N-SITM-GFP, 251^{N(WT)}-SITM-GFP, and 251^{N(R7W)}-SITM-GFP in WT HeLa cells. **(C)** A cycloheximide chase assay showing the degradation of WT and R7W TMEM251 over 2 hours. Scale bar: 10 μ m.

Additionally, we show in Figure S4B, using Tet-on leaky expression to mimic endogenous levels, that mutating R7 not only affects the stability of LYSET, but also subsequent mature CTSD levels. Finally, in Figure 6 A and B of Richards et al. (2022) and Figure 4E of Pechincha et al., (2022), there was clearly less LYSET in the R45W mutant compared to WT, indicating that this mutation is indeed less stable compared to WT. In fact, Pechincha et al stated that “LYSETR45W and LYSEY72X were barely detectable at the protein level, conceivably because they abrogated LYSET function by destabilization and premature truncation of the protein, respectively (Fig. 4E and Fig. S8, B and C).”

In the initial version of the manuscript, the authors presented data demonstrating that the N-terminal cytosolic amino acids of the short LYSET isoform (i) can mediate binding to GST-GOLPH3 in vitro and (ii) enforce GOLPH3-dependent Golgi retention when added to N-terminally to an artificial type II membrane protein reporter. Overcoming the limitations of the reporter data which I pointed out earlier, the authors have now included experiments arguing in favour of a GOLPH3-dependent Golgi localization of even full-length LYSET, showing mislocalization of an ectopically expressed full-length LYSET in GOLPH3/GOLPH3L KO U2OS cells and mislocalized endogenous GNPT in GOLPH3 knockdown cells (Fig. S5). I suggest that these data are all included in one of the manuscript's main figures, although, as pointed out earlier, this finding is not novel since Brauer et al. (2024) have shown this before.

As stated above, we have confirmed with the editorial office that we are under scoop protection. We are pleased that an independent study has largely confirmed our findings about Golph3, which accounts for only one-third of our papers. In addition, we identified the recognition motif to be FxxR.

Thank you for suggesting merging the two figures. We tried; however, we failed to merge them in a logical way. As you can tell, we performed a very extensive analysis here using both Chimera and full-length LYSET. To maintain the flow of the paper, we believe it is better to keep the images as they are.

An additional concern is that the authors do not show mislocalization of endogenous LYSET. Also, as a consequence of LYSET mislocalization, impaired cellular M6P tagging and maturation of lysosomal enzymes can be expected (see also ref #36) and experiments in GOLPH3 KO HeLa and U2OS cells available to the laboratory (see Figure 6B, ref #21 and Fig. S5A) should be used to more clearly demonstrate this (instead of Fig. S5E/F).

First, as we have repeatedly stated, there is no good LYSET antibody available to show the mislocalization of endogenous LYSET. However, we did show mislocalization using both chimera (figure 5B-C) and mildly overexpressed LYSET-GCN4 (figure S5A-B). Second, we have shown that endogenous cathepsin D has maturation defects in Golph3 Knockdown (Figure S5E-F). We are unsure what new information could be drawn by showing more lysosomal enzymes, especially in light of the fact that we have been repeatedly criticized as “not novel” and Brauer et al. have already shown this in their publication. The main conclusion we want to present here is that Golph3 is critical for maintaining LYSET Golgi-localization, and its recognition motif is FXXR. We believe we have provided sufficient evidence to justify them.

In addition, through comprehensive mutagenesis, the authors determined that F4 and R7 present in the LYSET N-terminus are critical to Golgi localization of the type II membrane protein reporter, but – strongly supporting that these residues are indeed relevant here – also of full-length LYSET ectopically expressed in 293T and SKMEL30 cells (Fig. S4). Again, I feel that this dataset should be included in a main figure. In addition, it would be interesting if there is an additive effect of the two mutants and whether any of the mutants shows impaired binding to GST-GOLPH3 in vitro. I also still feel that it needs to be explored whether the F4 and R7 mutants exhibit mislocalization when in the context of the long LYSET isoform or whether the substantially longer N-terminus can somehow overrule those mutations (see Brauer et al. 2024).

We have shown before that long and short isoforms are functionally redundant in our previous Nature Communications publication. Although interesting, repeating all the experiments with the long isoform is not justified, as it does not add new information. In addition, as acknowledged by this reviewer, Brauer et al. have tested the long isoform and reached a similar conclusion to ours.

Regarding combining the figures, please see our response above.

Regarding combining F4A with R7A mutants, we would like to point out that F4A or R7A alone is sufficient to abolish the Golgi localization. It is reasonable to assume that combining the two mutations would NOT do otherwise.

Finally, we showed reduced GST-GOLPH3 binding to the 251(R7W) peptide in our supporting figure 2 of our previous rebuttal letter, and we are unsure why the reviewer would request this data again. However, as stated earlier, we have included the data (please see supporting Figure 1A,B) again for the reviewer.

Figure 7:

What's the difference between the endogenously tagged GNPTAB expression in Fig. 7A

and B? Why is there so much background signal in the WCL of C? Why is there a precursor band detected in the 3HA-KI (-) cells? Shouldn't the LYSET WCL levels be decreased if there's more LYSET mislocalized to lysosomes?

Figure 7B is the quantification of Figure 7A. Thank you for pointing out the variation in background signals of the Anti-HA antibody (16B12, BioLegend). We are unsure about the variation; it could be due to the different lot numbers of the antibody, as we did the two sets of experiments at different times. However, we designed both experiments carefully by including the non-tagged WT controls. With this control, we are confident about our conclusion.

We disagree that “there is a precursor band detected in the 3HA-KI (-) cells”; the right interpretation should be that those bands are background bands.

The observation that LYSET does not reduce after knocking out VPS35 indicates that LYSET is resistant to luminal proteases, even after it mislocalizes to the lysosome. This is different from GNPTAB. The possible reason for the resistance is that LYSET is a small transmembrane protein with only 37 residues exposed to the lumen (Figure 5C), whereas GNPTAB has 1256 amino acids, and the majority of the protein is in the lumen.

Figure 8:

Fig.8B the IF data are of rather bad quality and the GRASP55 signal seems to be -in part- extremely overexposed. Please select better quality images.

Thank you for pointing this out. We have fixed this issue.

Fig. 8D, IF of poor quality, especially the WT LAMP2 staining. Please include higher quality (resolution) images. LYSET-GCN stainings seem to be overexposed which is precluding proper colocalization analysis (especially the lower panel of LAMP2 co-stainings).

We have fixed this issue. Thank you!

The authors postulate a mechanism of LYSET binding both GOLPH3 and VPS35. Please show the expression of a LYSET mutant both deficient in LYSET and VPS35 binding.

We are unsure about the request to show LYSET interacting with LYSET.

Regarding LYSET interaction with VPS35, we have tried very hard to get this to work. But this proves to be technically challenging. However, the mislocalization of endogenous LYSET (and GNPTAB) in VPS35 KO melanoma cells, as well as our extensive analysis using IL-2Ra-LYSETC chimera, together with our mutagenesis scan, provided compelling evidence to demonstrate the importance of retromer in maintaining LYSET at the Golgi.

Brauer et al. (see above) show that LYSET interacts with GOLPH3 and GOLPH3L. LYSET is mislocalized and is destabilized in double KO cells. The authors rather suggest VPS35 as being important for LYSET Golgi-localization. Are these cell-type-specific differences?

Here, we have shown that both GOLPH3 and VPS35 contribute to keeping LYSET in the Golgi, not just VPS35 alone. The GOLPH3 (and likely GOLPH3L) interacts with the N-terminus FXXR motif, whereas retromer likely interacts with the C-terminal V⁹⁷X^{Y99}—V¹⁰⁸I¹⁰⁹C¹¹⁰ motif.

Regarding GOLPH3-LYSET interaction, our conclusions are largely consistent. The minor discrepancy may be due to differences in cell lines and protein expression levels.

Figure 9:

The authors did not present data that convincingly shows two independent steps of coatamer-dependent recycling, although it's conceivable that there may be redundant mechanisms. Neither the figure legend nor the main text do explain the two distinct "coatamer-dependent recycling" boxes included in this illustration. This needs to be explained. Do the authors propose different modes of retrieving in the early and the late Golgi, with the latter being COPI-independent but GOLPH3-dependent? (with a label "coatamer component" still in the box?)

This was suggested by Reviewer #3 from the previous round. The two separate boxes are now merged into one.

Again, the authors postulate a role of LYSET in GNPTAB cleavage, which is not unambiguously shown. The slightly increased processing might well be explained by a stabilisation of GNPTAB in the cis-Golgi apparatus and therefore longer dwell-time in the proximity of MBTPS1. In addition, the authors have demonstrated themselves in Doray et al. that the stabilising mutant of GNPTAB is efficiently cleaved by MBTPS1 in LYSET KO cells and stabilized upon overexpression of LYSET, so it is unclear to the reviewer why they deliberately omit this function in their summary figure.

The reviewer misconstrued the summary figure. We did not deliberately omit this in the summary figure. We believe we have clearly explained both in text and in the figures that LYSET is necessary to keep GNPTAB in the Golgi. Without

LYSET, GNPTAB is sent to the lysosome to be degraded, and without LYSET, to keep GNPTAB in the Golgi S1P S1P-mediated cleavage can no longer occur. Additionally, LYSET is kept in the Golgi via both Golph3 and the retromer complex. In fact, the main idea of the figure is to show that GNPTAB is necessary for M6P modification, LYSET keeps GNPTAB in the Golgi, and that LYSET is kept in the Golgi via multiple mechanisms. Lastly, we have now added a sentence in Figure 9 to explain that LYSET keeps GNPTAB in the Golgi to facilitate its cleavage.

Additional issues:

As a side note I could not get access (despite asking twice at the editorial board) to a version of the revised manuscript containing the marked changes which would be required to fully appreciate the authors changes and responses to my review.

We apologize. The editorial office requested a marked version on July 25th, and we sent it to them on the same day. However, there might have been some miscommunication between the editorial office and the reviewer, as it did not reach the reviewer.

Page 15 of the rebuttal letter: the authors claim that they cannot do costainings due to antibody issues, but they have endogenously tagged GNPTAB-HA cells which they could use.

The endogenous GNPTAB-3HA KI cell line is in a HEK293t background. Although we can detect GNPTAB-3HA via western blot, we were unable to do so using immunofluorescence. This is most likely due to the low expression levels of GNPTAB in Hek293t cells.

Line 323ff. re: GOLPH3 recognition motif: The authors directly compare the FXXR motif found to be important experimentally with a previously suggested LXXR GOLPH3 recognition motif. However, as evident from the data presented by Welch et al. this motif is not necessarily present in all GOLPH3/3L clients and a polybasic stretch directly preceding the transmembrane domain of a Golgi type II protein may be a more accurate distinctive feature of a Golgi client. In fact, immediately after making this statement, the authors introduce GALNT2 as a prototypic GOLPH3 client which is used as their positive control – which completely lacks the LxxR motif (MRRRSR). This should be rephrased accordingly.

We have rephrased this part in the manuscript (lines 327-330). Thank you for the suggestion.

The authors now refer to TMEM251 as LYSET in the revised manuscript and it is good to further establish a uniform naming of the gene/protein in the literature. Yet, this makes it hard for a reader to understand labelling in Fig. 6H, in which “251N peptide” certainly refers to an N-terminal TMEM251 peptide. The authors should consider rephrasing the labelling of this panel to make this more obvious. Also, the exact sequence of the biotinylated peptide used should be provided.

Thank you for pointing this out. We have changed the labelling to “LYSET-N peptide” to make it more consistent with the rest of the manuscript. The exact peptide sequence is MMNFRQRM, which was described in the manuscript (line 337).

Line 458: Brauer et al showed that GOLPH3 and GOLPH3L are redundant in maintaining LYSET and GNPTAB in the cis-Golgi. Were both deleted by the authors? Does the siRNA the authors used target both isoforms? Is the antibody used specific for GOLPH3 or does it equally detect GOLPH3L? Could this explain the compensatory signal in the KD cells?

As shown in Figure S6 F,G we cleanly knocked out GOLPH3 in the U2OS cell line, while two of the three GOLPH3L chromosomes contained mutations. According to Ng MM, et al (2013), HeLa cells primarily express GOLPH3 and not GOLPH3L. Thus, we used a shRNA specific to GOLPH3 to knock down GOLPH3 only. We did, however, as a control, use shRNA targeting both GOLPH3 and GOLPH3L and had similar observations. Thus, we believe it is unlikely that the slight compensatory effect observed is due to GOLPH3L.

Handwritten labelling can be found on some immunoblot panels. In particular in case of Fig. 2D it would be good to mask this (as lane numbering is provided below the panels anyway).

We cropped the images to remove the handwritten labelling.

In panel 3A it would be good to highlight the degradation products mentioned with arrows.

We used a red arrow to highlight the degradation products.

Panel S3C appears to be very pixelated. Could a higher resolution panel be included in the figure?

Thank you for pointing this out. This was most likely a conversion issue. We changed the images so that they are no longer pixilated.

Like in Fig. S4A, the box labelled EF1a in Fig. 8A should be additionally garnished with an arrow to clearly demonstrate it is the promoter driving the expression of the fusion construct.

We added the arrow to clarify the promoter region. Thank you for the suggestion.

Inconsistent naming of the detection tag used: The tag is mostly referred to as “10x GCN” but as 10x GNC4” (Fig. S4D) (with the latter naming corresponding to the one used by Tanenbaum et al. in the original manuscript reporting the tag)

We changed all the labeling to 10xGCN4 to keep it consistent throughout the text.

How strong a reduction in GOLPH3 was achieved following knockdown in SKMEL30 cells (Fig. S5C) – such a control staining (GOLPH3 co-staining or immunoblot) is missing.

We added a western blot to show the reduction of GOLPH3 in SKMEL30 cells (now Fig. S5C)

The disease mucopolipidosis type 5 does not exist. LYSET deficiency is officially called dysostosis multiplex, Ain-Naz type. Ain et al. 2021 is incorrectly cited by the authors.

We previously proposed to call the disease Mucopolipidosis Type V due to its similarities to Mucopolipidosis Type II and III. However, as the reviewer recommended, we have added the name dysostosis multiplex (see lines 316-317).

Reviewer #2 (Remarks to the Author):

The revised manuscript is significantly improved, both in terms of new experimental data and clarity of writing. The authors have adequately addressed all of my previous comments.

We thank the reviewer for all the constructive comments.

Reviewer #4 (Remarks to the Author):

We thank the reviewer for all the constructive comments.

Rebuttal letter

Reviewer #1 (Remarks to the Author):

As clearly and in detail stated in my previous reviews my enthusiasm about this manuscript remains dampened. It seems -in contrast to this reviewer's view- that the editor is in favor of promoting this manuscript despite a number of experimental suggestions were not addressed and only briefly discussed.

We apologize that we could not address all of the Reviewers comments to a satisfactory degree. We tried our best to address all the comments we felt would improve the science of our paper and we thank the reviewer for all the thoughtful critique. We still feel that our manuscript not only confirms previous work, but more importantly provides novel advances to the field including:

- Resolve the topology of LYSET
- Identify the critical residues of LYSET
- Confirm GOLPH3's role in maintaining LYSET in the Golgi,
- Identified the recognition motif of GOLPH3 for LYSET,
- Determined that the retromer plays a role in maintaining LYSET in the Golgi
- Show evidence that the retromer binds to the C-terminal of LYSET.

Some additional remarks:

"We are unsure how to answer this criticism because we did not acknowledge that "the precursor protein may exhibit different stability in the presence of lysosomal proteases." Please indicate the lines where we made such a statement. Instead, we found that LYSET is resistant to lysosomal degradation."

I referred in a long paragraph to the stability of GNPTAB and the authors's comment related to the stability of LYSET is misplaced and does not fit to my remark which should be thoroughly read and responded to.

We re-read the reviewers' comments and hope that the following response is satisfactory. As discussed in our previous rebuttal letter, we did not mention anything related to the stability of the GNPTAB precursor protein in the presence of lysosomal proteases in our manuscript and was thus unsure what the reviewer was referring to. We did, however, mention that LYSET was resistant to lysosomal degradation in our manuscript and therefore included it in our previous reply to the Reviewer.

Please see below a more in-depth response to the reviewers' additional concerns.

In regards to the request by the Reviewer to “*More strongly voice to what extent their own data now support the model reported before by the other groups*” and to “*clarify that the primary point of contention lies with their own publication (Zhang et al., 2022, Nat Commun.), in which the role of LYSET was interpreted as the sole activator of MBTPS1-mediated cleavage of GNPTAB—an interpretation that remains under debate.*”

We believe we have comprehensively discussed the two models and acknowledged that LYSET is necessary to keep GNPT in the Golgi, as proposed in the Trafficking model. Below we have outlined the discussion of the two models and our conclusions:

- In Lines 82-94 we give an introduction to the two proposed models and acknowledge that the processing model does not account for the reduction of total GNPTAB in LYSET KO cells.
- In line 98-99 we summarize that without LYSET, GNPT is mislocalized and degraded by the lysosome.
- In line 108-121 we confirmed that GNPTAB protein levels are reduced in LYSET KO Hek293t cells (Figure 1A-E).
- In line 122-129 we used the SKMEL30 cells to again confirm that GNPTAB levels are reduced in cells lacking LYSET (Figure 1F-H).
- In lines 142-147 we summarized our findings that LYSET is necessary for GNPTAB processing and stability.
- In lines 149-152, we again discussed the two models and acknowledged that the processing model cannot explain the large loss of total GNPTAB.
- The entire section between lines 148 and 197 is dedicated to show that LYSET is necessary to keep GNPTAB in the Golgi and thus facilitates cleavage (Figure 2A-G).
- We finalized the section by summarizing our findings and acknowledging that LYSET is necessary to keep both full length and uncleaved GNPTAB in the Golgi, and thus consequently facilitates S1P-mediated cleavage (lines 195-197).
- Next, we again dedicated a section (lines 198-233) to validate that both full-length and cleaved GNPTAB gets degraded in the lysosome (Figure 3A-G).
- Finally, we discussed the two models again (lines 421-431) clearly stating that without LYSET full length GNPT may rapidly exit Golgi and therefore reduce S1P cleavage efficiency by reducing the precursor concentration.
- We acknowledged that as stated by Richards et al., and Pechincha et al., LYSET maintains GNPT in the Golgi (lines 433-434)

We hope the reviewer takes note of our careful discussion of the literature, our attempts to resolve the controversy and our acknowledgement that LYSET maintains GNPT in the Golgi as proposed previously by Richards et al. and Pechincha et al.

The reviewer also raises the concern that “*The perceived “controversy” between the two back-to-back Science papers is relatively minor and may largely stem from methodological differences, such as the use of different cell lines, endogenous expression*

versus overexpression systems, and antibodies that detect either the alpha or beta subunits of GNPTAB.”

Regarding this, as mentioned in the previous rebuttal letter, we have since changed the wording to more accurately discuss the controversy and the data presented by Richards et al. and Pechincha et al. in our manuscript. Specifically, we refer to the two science papers in the following context:

- As mentioned above, we give an introduction to the two proposed models and acknowledge that the processing model does not account for the reduction of total GNPTAB in LYSET KO cells (Lines 82-94).
- In lines 442-448 we mention that we could not see the mislocalization of GNPTAB to the lysosome via immunostaining as seen by Richards et al., but specifically mention that it could be due to the antigen regions being digested. In this paragraph we mentioned that Richards et al., observed relocalization of GNPTAB to the lysosomes in HAP1 cells (lines 445-448). Please note that we do not dispute the point that GNPTAB is mislocalized to the lysosome, since, as discussed above, we confirmed this via western blot.
- We have, as per the previous rebuttal letter, changed the wording and included Richards et al., when noting the fact that BafA1 stabilizes GNPT in the lysosome (lines 227-229 and 445-448)
- We mentioned that Pechincha et al., proposed that GNPT is unstable due to its relatively hydrophilic transmembrane domain (lines 452-453). Our data support this hypothesis by using alanine scanning to identify multiple critical regions of GNPTAB, including the transmembrane regions (Figure 5C).

We thank the reviewer for all the thoughtful input to help improve the manuscript so that we more accurately express our data and contribute to the field.

“We thank the reviewer for pointing this out. This result strongly supports our conclusion that LYSET facilitates S1P-mediated GNPTAB cleavage. All three papers confirmed that the mRNA level of GNPTAB is not affected in LYSET KO cells. With the degradation of the precursor, one should expect to see reduced processing.”

The authors again misinterpret the conclusion. While both Richards et al. and Pechincha et al. focus on the degradation of the cleaved subunits, they do not exclude that the precursor can equally be lysosomally degraded. Pechincha et al., however, demonstrate that the kinetics of GNPTAB cleavage are not altered in the absence of LYSET. Of course, the cleavage of the precursor is reduced if it is degraded. This is a trivial conclusion which does not support the authors' interpretation that LYSET facilitates the S1P-mediated cleavage. The authors seem uncertain to confirm their originally proposed role for LYSET as a cleavage-activating factor.

To answer the reviewer, we discussed this point in depth in both our manuscript and the previous rebuttal letters. We used the beginning of this manuscript to resolve the controversy between the models proposed by us (Zhang et al.,) and Richards et al. and Pechincha et al. We acknowledge in lines 230-233 that LYSET is necessary to keep both full length and uncleaved GNPTAB in the Golgi, unlike what we initially proposed in Zhang et al. Instead, we suggest that LYSET facilitates S1P-mediated cleavage by maintaining GNPT in the Golgi. Without LYSET, S1P cleavage efficiency is reduced since the GNPTAB precursor concentration is lowered (lines 428-429). In line 425-426, we mention that LYSET can pull down both S1P and GNPT as shown in Zhang et al. which supports the processing model. However, we also mention that without LYSET, the precursor rapidly leaves the Golgi which could also affect S1P cleavage of GNPTAB (lines 427-429). Moreover, our data show that overexpressing either GNPTAB or S1P bypasses the need for LYSET (lines 429-431, Figure 2C-D).

We would like to respectfully disagree with the Reviewer, as all these experiments and discussion do not just lead to a trivial conclusion. It instead allows us to resolve the controversy that arose between the papers. We believe that when differences arise it is in the scientific community's best interest to clarify, and we have done so extensively as discussed above.

“WCL stands for whole cell lysate, which is the starting material used for lysosome purification. There might be two reasons for the many bands in LAMP2 blots: 1) LAMP2 is a heavily glycosylated protein. It always migrates as a smear or several bands. 2) It is not uncommon for antibodies to recognize additional, non-specific bands. In the “LYS” fraction, however, the two non-specific bands, were removed by lysosome purification, confirming lysosome purification can remove non-specific bands from the WCL.”

There are several excellent LAMP2 antibodies (e.g. from DSHB) that detect no unspecific bands, therefore the reviewer is surprised by the poor quality of the western blot.

Thank you for the suggestion. We used the DSHB antibody to probe for LAMP2 in our WCL and Lysosome fractions and subsequently replaced the blots in Figure 3D and Figure 7C.

Response to Reviewer

We would like to thank the reviewer for taking the time to read through and assisting us in improving our manuscript titled “Molecular Insights into the Regulation of GNPTAB by LYSET”.

Please see below for a detailed response to the reviewers’ concerns:

Reviewer #1 (Remarks to the Author):

The manuscript by Yang and colleagues describes the recently identified LYSET protein as a two-pass Golgi membrane protein that stabilizes GNPT, promotes its S1P-dependent cleavage, and prevents its degradation by protecting it from mislocalization to lysosomes. The authors suggest that LYSET achieves this function by interacting with GOLPH3 and the retromer—through distinct LYSET motifs—to retain the LYSET–GNPT complex in the Golgi, thereby ensuring proper operation of the M6P pathway.

As stated, multiple times in my earlier reviews, I acknowledge and appreciate certain new findings and contributions to the field. However, I find that the overall scientific advancement and impact compared to previously published studies is not as substantial as claimed. A journal such as *Communications Biology* or a comparable publication would likely be more appropriate for this work.

We are regretful that we could not convince the Reviewer of the suitability of this manuscript for *Nature Communications* despite our best efforts to address the numerous critiques of the reviewer. Nevertheless, we hope that our manuscript does contribute to furthering the M6P field and we are grateful to the editors for allowing us to share our research.

Upon re-examining the manuscript, I also noted that, despite my previous comments, some data presentations still rely on only two measurements (e.g., Fig. 2E and the activity determinations in Supplementary Fig. S1). Moreover, no error bars or standard deviations (STDEVA) are provided.

As the reviewer requested we have performed an additional repeat for Figure S1C and added error bars to the values.

Unfortunately, due to our collaborator’s passing, with his lab closed down, and the materials no longer available we cannot repeat Figure 2E. However, Figure 2E is independently supported by Figure 2F, which demonstrates a dose-dependent GNPTAB-V5 processing using a different experimental approach. We respectfully argue that this independent validation in Figure 2F provides stronger support than simply adding an additional replicate to Figure 2E.